# Stochastic Shortest Path: Minimax, Parameter-Free and Towards Horizon-Free Regret

**Jean Tarbouriech**[*]
Facebook AI Research & Inria Lille
`jean.tarbouriech@gmail.com`

**Runlong Zhou**[*]
Tsinghua University
`zhourunlongvector@gmail.com`

**Simon S. Du**
University of Washington & Facebook AI Research
`ssdu@cs.washington.edu`

**Matteo Pirotta**
Facebook AI Research Paris
`pirotta@fb.com`

**Michal Valko**
DeepMind Paris
`valkom@deepmind.com`

**Alessandro Lazaric**
Facebook AI Research Paris
`lazaric@fb.com`

## Abstract

We study the problem of learning in the stochastic shortest path (SSP) setting, where an agent seeks to minimize the expected cost accumulated before reaching a goal state. We design a novel model-based algorithm EB-SSP that carefully skews the empirical transitions and perturbs the empirical costs with an exploration bonus to induce an optimistic SSP problem whose associated value iteration scheme is guaranteed to converge. We prove that EB-SSP achieves the minimax regret rate $\widetilde{O}(B_\star\sqrt{SAK})$, where $K$ is the number of episodes, $S$ is the number of states, $A$ is the number of actions, and $B_\star$ bounds the expected cumulative cost of the optimal policy from any state, thus closing the gap with the lower bound. Interestingly, EB-SSP obtains this result while being parameter-free, i.e., it does not require any prior knowledge of $B_\star$, nor of $T_\star$, which bounds the expected time-to-goal of the optimal policy from any state. Furthermore, we illustrate various cases (e.g., positive costs, or general costs when an order-accurate estimate of $T_\star$ is available) where the regret only contains a logarithmic dependence on $T_\star$, thus yielding the first (nearly) horizon-free regret bound beyond the finite-horizon MDP setting.

## 1  Introduction

Stochastic shortest path (SSP) is a goal-oriented reinforcement learning (RL) setting where the agent aims to reach a predefined goal state while minimizing its total expected cost [Bertsekas, 1995]. In particular, the interaction between the agent and the environment ends *only* when (and if) the goal state is reached, so the length of an episode is not predetermined (nor bounded) and it is influenced by the agent's behavior. SSP includes both finite-horizon and discounted Markov Decision Processes (MDPs) as special cases. Moreover, many common RL problems can be cast under the SSP formulation, such as game playing (e.g., Atari games) or navigation (e.g., Mujoco mazes).

We study the online learning problem in the SSP setting (online SSP in short), where both the transition dynamics and the cost function are initially unknown and the agent interacts with the environment through multiple episodes. The learning objective is to achieve a performance as close

---

[*]equal contribution

35th Conference on Neural Information Processing Systems (NeurIPS 2021).

as possible to the optimal policy $\pi^\star$, that is, the agent should achieve low *regret* (i.e., the cumulative difference between the total cost accumulated across episodes by the agent and by the optimal policy). We identify three desirable properties for a learning algorithm in online SSP.

- **Desired property 1: Minimax.** The information-theoretic lower bound on the regret is $\Omega(B_\star\sqrt{SAK})$ [Rosenberg et al., 2020], where $K$ is the number of episodes, $S$ is the number of states, $A$ is the number of actions, and $B_\star$ bounds the total expected cost of the optimal policy starting from any state (assuming for simplicity that $B_\star \geq 1$).

  *An algorithm for online SSP is (nearly) minimax optimal if its regret is bounded by $\widetilde{O}(B_\star\sqrt{SAK})$, up to logarithmic factors and lower-order terms.*

- **Desired property 2: Parameter-free.** Another relevant dimension is the amount of prior knowledge required by the algorithm. While the knowledge of $S$, $A$, and the cost (or reward) range $[0, 1]$ is standard across regret-minimization settings (e.g., finite-horizon, discounted, average-reward), the complexity of learning in SSP problems may be linked to SSP-specific quantities such as $B_\star$ and $T_\star$, which denotes the expected time-to-goal of the optimal policy from any state.

  *An algorithm for online SSP is parameter-free if it relies neither on $T_\star$ nor $B_\star$ prior knowledge.*

- **Desired property 3: Horizon-free.** A core challenge in SSP is to trade off between minimizing costs and quickly reaching the goal state. This is accentuated when the instantaneous costs are small, i.e., when there is a mismatch between $B_\star$ and $T_\star$. Indeed, while $B_\star \leq T_\star$ always holds since the cost range is $[0, 1]$, the gap between the two may be arbitrarily large (see e.g., the simple example of App. A). The lower bound stipulates that the regret does depend on $B_\star$, while the "time horizon" of the problem, i.e., $T_\star$ should a priori not impact the regret, even as a lower-order term.

  *An algorithm for online SSP is (nearly) horizon-free if its regret depends only logarithmically on $T_\star$.*

  Our definition extends the property of so-called horizon-free bounds recently uncovered in finite-horizon MDPs with total reward bounded by 1 [Wang et al., 2020a, Zhang et al., 2021a,b]. These bounds depend only logarithmically on the horizon $H$, which is the number of time steps by which *any* policy terminates. Such notion of horizon would clearly be too strong in the more general class of SSP, where some (even most) policies may never reach the goal, thus having unbounded time horizon. A more adequate notion of horizon in SSP is $T_\star$, which bounds the *expected* time of the *optimal* policy to terminate the episode starting from any state.

Finally, while the previous properties focus on the learning aspects of the algorithm, another important consideration is computational efficiency. It is desirable that a learning algorithm has run-time complexity at most polynomial in $K, S, A, B_\star$, and $T_\star$. All existing algorithms for online SSP, including the one proposed in this paper, meet such requirement.

**Related Work.** Table 1 reviews the existing work on online learning in SSP. The setting was first studied by Tarbouriech et al. [2020a] who gave a parameter-free algorithm with a $\widetilde{O}(K^{3/2})$ regret guarantee. Rosenberg et al. [2020] then improved this result by deriving the first order-optimal algorithm with regret $\widetilde{O}(B_\star^{3/2}S\sqrt{AK})$ in the parameter-free case and $\widetilde{O}(B_\star S\sqrt{AK})$ if $B_\star$ is known (to tune cost perturbation appropriately). Both approaches are model-optimistic,[2] drawing inspiration from the ideas behind the UCRL2 algorithm [Jaksch et al., 2010] for average-reward MDPs.

Concurrently to our work, Cohen et al. [2021] propose an algorithm for online SSP based on a black-box reduction from SSP to finite-horizon MDPs. It successively tackles finite-horizon problems with horizon set to $H = \Omega(T_\star)$ and costs augmented by a terminal cost set to $c_H(s) = \Omega(B_\star\mathbb{I}(s \neq g))$, where $g$ denotes the goal state. This finite-horizon construction guarantees that its optimal policy has a similar value function to the optimal policy in the original SSP instance up to a lower-order bias. Their algorithm comes with a regret bound of $O(B_\star\sqrt{SAKL} + T_\star^4S^2AL^5)$, with $L = \log(KT_\star SA\delta^{-1})$ (with probability at least $1 - \delta$). It achieves a nearly minimax-optimal rate, however it relies on both $T_\star$ and $B_\star$ prior knowledge to tune the horizon and terminal cost in the reduction, respectively.[3]

---

[2]We refer the reader to Neu and Pike-Burke [2020] for details on the differences and interplay between model-optimistic and value-optimistic approaches.

[3]As mentioned by Cohen et al. [2021, Remark 2], in the case of positive costs lower bounded by $c_{\min} > 0$, their knowledge of $T_\star$ can be bypassed by replacing it with the upper bound $T_\star \leq B_\star/c_{\min}$. However, when generalizing from the $c_{\min}$ case to general costs with a perturbation argument, their regret guarantee worsens from $\widetilde{O}(\sqrt{K} + c_{\min}^{-4})$ to $\widetilde{O}(K^{4/5})$, because of the poor additive dependence on $c_{\min}^{-1}$.

| Algorithm | Regret | Minimax | Parameters | Horizon-Free |
|---|---|---|---|---|
| [Tarbouriech et al., 2020a] | $\widetilde{O}_K(K^{2/3})$ | No | None | No |
| [Rosenberg et al., 2020] | $\widetilde{O}\left(B_\star S\sqrt{AK} + T_\star^{3/2}S^2 A\right)$ | No | $B_\star$ | No |
| | $\widetilde{O}\left(B_\star^{3/2} S\sqrt{AK} + T_\star B_\star S^2 A\right)$ | No | None | No |
| [Cohen et al., 2021] *(concurrent work)* | $\widetilde{O}\left(B_\star\sqrt{SAK} + T_\star^4 S^2 A\right)$ | Yes | $B_\star, T_\star$ | No |
| This work | $\widetilde{O}\left(B_\star\sqrt{SAK} + B_\star S^2 A\right)$ | Yes | $B_\star, T_\star$ | **Yes** |
| | $\widetilde{O}\left(B_\star\sqrt{SAK} + B_\star S^2 A + \frac{T_\star}{\text{poly}(K)}\right)$ | Yes | $B_\star$ | No[*] |
| | $\widetilde{O}\left(B_\star\sqrt{SAK} + B_\star^3 S^3 A\right)$ | Yes | $T_\star$ | **Yes** |
| | $\widetilde{O}\left(B_\star\sqrt{SAK} + B_\star^3 S^3 A + \frac{T_\star}{\text{poly}(K)}\right)$ | Yes | **None** | No[*] |
| Lower Bound | $\Omega(B_\star\sqrt{SAK})$ | - | - | - |

Table 1: Regret comparisons of algorithms for online SSP (we assume for simplicity that $B_\star \geq 1$). The notation $\widetilde{O}$ omits logarithmic factors and $\widetilde{O}_K$ only reports the dependence in $K$. **Regret** is the performance metric of Eq. 1. **Minimax**: Whether the regret matches the $\Omega(B_\star\sqrt{SAK})$ lower bound [Rosenberg et al., 2020], up to logarithmic and lower-order terms. **Parameters**: The parameters that the algorithm requires as input: either both $B_\star$ and $T_\star$, or one of them, or none (i.e., parameter-free). **Horizon-Free**: Whether the regret bound depends only logarithmically on $T_\star$. [*]If $K$ is known in advance, the additive term $T_\star/\text{poly}(K)$ has a denominator that is polynomial in $K$, so it becomes negligible for large values of $K$ (if $K$ is unknown, the additive term is $T_\star$). See Sect. 4 for the full statements of our bounds.

Finally, all existing bounds contain lower-order dependencies either on $T_\star$ in the case of general costs, or on $B_\star/c_{\min}$ in the case of positive costs lower bounded by $c_{\min} > 0$ (note that $T_\star \leq B_\star/c_{\min}$, which is one of the reasons why $c_{\min}$ can show up in existing bounds). As such, no existing analysis satisfies horizon-free properties for online SSP.

**Contributions.** We summarize our main contributions as follows (see also Table 1):

- We propose EB-SSP (Exploration Bonus for SSP), a new algorithm for online SSP. It introduces a value-optimistic scheme to efficiently compute optimistic policies for SSP, by both perturbing the empirical costs with an exploration bonus and slightly biasing the empirical transitions towards reaching the goal from *each* state-action pair with positive probability. Under these biased transitions, *all* policies are in fact proper (i.e., they eventually reach the goal with probability 1 starting from any state). We decay the bias over time in a way that it only contributes to a lower-order regret term. See Sect. 3 for an overview of our algorithm and analysis. Note that EB-SSP is *not* based on a model-optimistic approach[2] [Tarbouriech et al., 2020a, Rosenberg et al., 2020], and it does *not* rely on a reduction from SSP to finite-horizon [Cohen et al., 2021] (i.e., we operate at the level of the non-truncated SSP model);

- EB-SSP is the first algorithm to achieve the **minimax** regret rate of $\widetilde{O}(B_\star\sqrt{SAK})$ while simultaneously being **parameter-free**: it does not require to know nor estimate $T_\star$, and it is able to bypass the knowledge of $B_\star$ at the cost of only logarithmic and lower-order contributions to the regret;

- EB-SSP is the first algorithm to achieve **horizon-free** regret for SSP in various cases: i) positive costs, ii) no almost-sure zero-cost cycles, and iii) the general cost case when an order-accurate estimate of $T_\star$ is available (i.e., a value $\overline{T}_\star$ such that $\frac{T_\star}{\upsilon} \leq \overline{T}_\star \leq \lambda T_\star^\zeta$ for some unknown constants $\upsilon, \lambda, \zeta \geq 1$ is available). This property is especially relevant if $T_\star$ is much larger than $B_\star$, which can occur in SSP models with very small instantaneous costs. Moreover, EB-SSP achieves its horizon-free guarantees while maintaining the minimax rate. For instance, under general costs when

relying on $T_\star$ and $B_\star$, its regret is $\widetilde{O}(B_\star\sqrt{SAK}+B_\star S^2 A)$.[4] To the best of our knowledge, EB-SSP yields the first set of (nearly) horizon-free bounds beyond the setting of finite-horizon MDPs.

**Additional Related Work.** *Planning in SSP:* Early work by Bertsekas and Tsitsiklis [1991], followed by [e.g., Bertsekas, 1995, Bonet, 2007, Kolobov et al., 2011, Bertsekas and Yu, 2013, Guillot and Stauffer, 2020], examine the planning problem in SSP, i.e., how to compute an optimal policy when all parameters of the SSP model are known. Under mild assumptions, the optimal policy is deterministic and stationary and can be computed efficiently using standard planning techniques, e.g., value iteration, policy iteration or linear programming.

*Regret minimization in MDPs:* The exploration-exploitation dilemma in tabular MDPs has been extensively studied in finite-horizon [e.g., Azar et al., 2017, Jin et al., 2018, Zanette and Brunskill, 2019, Efroni et al., 2019, Simchowitz and Jamieson, 2019, Zhang et al., 2020, Neu and Pike-Burke, 2020, Xu et al., 2021, Menard et al., 2021] and infinite-horizon [e.g., Jaksch et al., 2010, Bartlett and Tewari, 2012, Fruit et al., 2018, Wang et al., 2020b, Qian et al., 2019, Wei et al., 2020].

*Other SSP-based settings:* SSP with adversarial costs was investigated by Rosenberg and Mansour [2021], Chen et al. [2021], Chen and Luo [2021].[5] Tarbouriech et al. [2021] study the sample complexity of SSP with a generative model, as a standard regret-to-PAC conversion may not hold in SSP (as opposed to finite-horizon). Exploration problems involving multiple goal states (i.e., multi-goal SSP or goal-conditioned RL) were analyzed by Lim and Auer [2012], Tarbouriech et al. [2020b].

## 2 Preliminaries

An SSP problem is an MDP $M := \langle S, A, P, c, s_0, g \rangle$, where $S$ is the finite state space with cardinality $S$, $A$ is the finite action space with cardinality $A$, and $s_0 \in S$ is the initial state. We denote by $g \notin S$ the goal state, and we set $S' := S \cup \{g\}$ (thus $S' := S + 1$). Taking action $a$ in state $s$ incurs a cost drawn i.i.d. from a distribution on $[0, 1]$ with expectation $c(s, a)$, and the next state $s' \in S'$ is selected with probability $P(s'|s, a)$ (where $\sum_{s' \in S'} P(s'|s, a) = 1$). The goal state $g$ is absorbing and zero-cost, i.e., $P(g|g, a) = 1$ and $c(g, a) = 0$ for any action $a$.

For notational convenience, let $P_{s,a} := P(\cdot|s, a)$, $P_{s,a,s'} := P(s'|s, a)$. For any two vectors $X, Y$ of size $S'$, we write their inner product as $XY := \sum_{s \in S'} X(s)Y(s)$, we denote by $X^2$ the vector $[X(1)^2, X(2)^2, \dots, X(S')^2]^\top$, let $\|X\|_\infty := \max_{s \in S'} |X(s)|$, $\|X\|_\infty^{\neq g} := \max_{s \in S} |X(s)|$, and if $X$ is a probability distribution on $S'$, then $\mathbb{V}(X, Y) := \sum_{s \in S'} X(s)Y(s)^2 - (\sum_{s \in S'} X(s)Y(s))^2$.

A stationary and deterministic policy $\pi : S \to A$ is a mapping from state $s$ to action $\pi(s)$. A policy $\pi$ is said to be proper if it reaches the goal with probability 1 when starting from any state in $S$ (otherwise it is improper). We denote by $\Pi_{\text{proper}}$ the set of proper, stationary and deterministic policies. We make the following basic assumption which ensures that the SSP problem is well-posed.

**Assumption 1.** *There exists at least one proper policy, i.e., $\Pi_{proper} \neq \emptyset$.*

The agent's objective is to minimize its expected cumulative cost incurred until the goal is reached. The value function (also called cost-to-go) of a policy $\pi$ and its associated $Q$-function are defined as

$$V^\pi(s) := \lim_{T \to \infty} \mathbb{E}\bigg[\sum_{t=1}^{T} c_t(s_t, \pi(s_t)) \,\big|\, s_1 = s\bigg], \;\; Q^\pi(s, a) := \lim_{T \to \infty} \mathbb{E}\bigg[\sum_{t=1}^{T} c_t(s_t, \pi(s_t)) \,\big|\, s_1 = s, \pi(s_1) = a\bigg],$$

where $c_t \in [0, 1]$ is the (instantaneous) cost incurred at time $t$ at state-action pair $(s_t, \pi(s_t))$, and the expectation is w.r.t. the random sequence of states generated by executing $\pi$ starting from state $s \in S$ (and taking action $a \in A$ in the second case). Note that $V^\pi$ may have unbounded components if $\pi$ never reaches the goal. For a proper policy $\pi$, $V^\pi(s)$ and $Q^\pi(s, a)$ are finite for any $s, a$. By definition of the goal, we set $V^\pi(g) = Q^\pi(g, a) = 0$ for all policies $\pi$ and actions $a$. Finally, we

---

[4]We conjecture the optimal problem-independent regret in SSP to be $\widetilde{O}(B_\star\sqrt{SAK} + B_\star SA)$ (by analogy with the conjecture of Menard et al. [2021] for finite-horizon MDPs), which shows the tightness of our bound up to an $S$ lower-order factor.

[5]A different line of work [e.g. Neu et al., 2010, 2012, Rosenberg and Mansour, 2019a,b, Jin et al., 2020, Jin and Luo, 2020] studies finite-horizon MDPs with adversarial costs (sometimes called online loop-free SSP), where an episode ends after a fixed number of $H$ steps (as opposed to lasting as long as the goal is reached).

denote by $T^\pi(s)$ the expected time that $\pi$ takes to reach $g$ starting at state $s$; in particular, if $\pi$ is proper then $T^\pi(s)$ is finite for all $s$, yet if $\pi$ is improper there must exist at least one $s$ such that $T^\pi(s) = \infty$.

Equipped with Asm. 1 and an additional condition on improper policies defined below, one can derive important properties on the optimal policy $\pi^\star$ that minimizes the value function component-wise.

**Lemma 2** (Bertsekas and Tsitsiklis, 1991; Yu and Bertsekas, 2013). *Suppose that Asm. 1 holds and that for every improper policy $\pi'$ there exists at least one state $s \in \mathcal{S}$ such that $V^{\pi'}(s) = +\infty$. Then the optimal policy $\pi^\star$ is stationary, deterministic, and proper. Moreover, $V^\star = V^{\pi^\star}$ is the unique solution of the optimality equations $V^\star = \mathcal{L}V^\star$ and $V^\star(s) < +\infty$ for any $s \in \mathcal{S}$, where for any vector $V \in \mathbb{R}^S$ the optimal Bellman operator $\mathcal{L}$ is defined as $\mathcal{L}V(s) := \min_{a \in \mathcal{A}}\{c(s,a) + P_{s,a}V\}$. Also, the optimal $Q$-value, denoted by $Q^\star = Q^{\pi^\star}$, is related to the optimal value function as follows: $Q^\star(s,a) = c(s,a) + P_{s,a}V^\star$ and $V^\star(s) = \min_{a \in \mathcal{A}} Q^\star(s,a)$, for all $(s,a) \in \mathcal{S} \times \mathcal{A}$.*

Since we will target the best proper policy, we will handle the second requirement of Lem. 2 as follows [Bertsekas and Yu, 2013, Rosenberg et al., 2020]. First, the requirement is in particular verified if all instantaneous costs are strictly positive. To deal with the case of non-negative costs, we can introduce a small additive perturbation $\eta \in (0, 1]$ to all costs to yield a new (strictly positive) cost function $c_\eta(s,a) = \max\{c(s,a), \eta\}$. In this cost-perturbed MDP, the conditions of Lem. 2 hold so we get an optimal policy $\pi_\eta^\star$ that is stationary, deterministic and proper and has a finite value function $V_\eta^\star$. Taking the limit as $\eta \to 0$, we have that $\pi_\eta^\star \to \pi^\star$ and $V_\eta^\star \to V^{\pi^\star}$, where $\pi^\star$ is the optimal proper policy in the original model that is also stationary and deterministic, and $V^{\pi^\star}$ denotes its value function. This enables to circumvent the second condition of Lem. 2 and only require Asm. 1 to hold.

**Learning formulation.** We consider the learning problem where the agent does not have any prior knowledge of the cost function $c$ or transition function $P$. Each episode starts at the initial state $s_0$ (the extension to any possibly unknown distribution of initial states is straightforward), and ends *only* when the goal state $g$ is reached (note that this may never happen if the agent does not reach the goal). We evaluate the performance of the agent after $K$ episodes by its *regret*, which is defined as

$$R_K := \sum_{k=1}^{K} \sum_{h=1}^{I^k} c_h^k - K \cdot \min_{\pi \in \Pi_{\text{proper}}} V^\pi(s_0), \tag{1}$$

where $I^k$ is the time needed to complete episode $k$ and $c_h^k$ is the cost incurred in the $h$-th step of episode $k$ when visiting $(s_h^k, a_h^k)$. If there exists $k$ such that $I^k$ is infinite, then we define $R_K = \infty$. Throughout we denote the optimal proper policy by $\pi^\star$ and $V^\star(s) := V^{\pi^\star}(s) = \min_{\pi \in \Pi_{\text{proper}}} V^\pi(s)$ and $Q^\star(s,a) := Q^{\pi^\star}(s,a) = \min_{\pi \in \Pi_{\text{proper}}} Q^\pi(s,a)$ for all $(s,a)$. Let $B_\star > 0$ bound the values of $V^\star$, i.e., $B_\star := \max_{s \in \mathcal{S}} V^\star(s)$. Note that $Q^\star(s,a) \le 1 + B_\star$. Also let $T_\star > 0$ bound the expected time-to-goal of the optimal policy, i.e., $T_\star := \max_{s \in \mathcal{S}} T^{\pi^\star}(s)$. We see that $B_\star \le T_\star < +\infty$.

## 3 Main Algorithm

We introduce our algorithm EB-SSP (Exploration Bonus for SSP) in Alg. 1. It takes as input the state-action space $\mathcal{S} \times \mathcal{A}$ and confidence level $\delta \in (0, 1)$. For now it considers that an estimate $B$ such that $B \ge \max\{B_\star, 1\}$ is available, and we later handle the case of unknown $B_\star$ (Sect. 4.2 and App. H). As explained in Sect. 2, the algorithm enforces the conditions of Lem. 2 to hold by adding a small cost perturbation $\eta \in [0, 1]$ (cf. lines 3, 12 in Alg. 1) — either $\eta = 0$ if the agent is aware that all costs are already positive, otherwise a careful choice of $\eta > 0$ is provided in Sect. 4.

Our algorithm builds on a value-optimistic approach by sequentially constructing optimistic lower bounds on the optimal $Q$-function and executing the policy that greedily minimizes them. Similar to the MVP algorithm of Zhang et al. [2021a] designed for finite-horizon RL, we adopt the doubling update framework (first proposed by Jaksch et al. [2010]): whenever the number of visits of a state-action pair is doubled, the algorithm updates the empirical cost and transition probability of this state-action pair, and computes a new optimistic $Q$-estimate and optimistic greedy policy. Note that this slightly differs from MVP which waits for the end of its finite-horizon episode to update the policy. In SSP, however, having this delay may yield linear regret as the episode has the risk of never terminating under the current policy (e.g., if it is improper), which is why we perform the policy update instantaneously when the doubling condition is met.

**Algorithm 1:** Algorithm EB-SSP

1  **Input:** $\mathcal{S}$, $s_0 \in \mathcal{S}$, $g \notin \mathcal{S}$, $\mathcal{A}$, $\delta$.
2  **Input:** an estimate $B$ guaranteeing $B \geq \max\{B_\star, 1\}$ (see Sect. 4.2 and App. H if not available).
3  **Optional input:** cost perturbation $\eta \in [0, 1]$.
4  **Specify:** Trigger set $\mathcal{N} \leftarrow \{2^{j-1} : j = 1, 2, \ldots\}$. Constants $c_1 = 6$, $c_2 = 36$, $c_3 = 2\sqrt{2}$, $c_4 = 2\sqrt{2}$.
5  For $(s, a, s') \in \mathcal{S} \times \mathcal{A} \times \mathcal{S}'$, set $N(s, a) \leftarrow 0$; $n(s, a) \leftarrow 0$; $N(s, a, s') \leftarrow 0$; $\widehat{P}_{s,a,s'} \leftarrow 0$;
     $\theta(s, a) \leftarrow 0$; $\widehat{c}(s, a) \leftarrow 0$; $Q(s, a) \leftarrow 0$; $V(s) \leftarrow 0$.
6  Set initial time step $t \leftarrow 1$ and trigger index $j \leftarrow 0$.
7  **for** episode $k = 1, 2, \ldots$ **do**
8  $\quad$ Set $s_t \leftarrow s_0$
9  $\quad$ **while** $s_t \neq g$ **do**
10 $\quad\quad$ Take action $a_t = \arg\min_{a \in \mathcal{A}} Q(s_t, a)$, incur cost $c_t$ and observe next state $s_{t+1} \sim P(\cdot | s_t, a_t)$.
11 $\quad\quad$ Set $(s, a, s', c) \leftarrow (s_t, a_t, s_{t+1}, \max\{c_t, \eta\})$ and $t \leftarrow t + 1$.
12 $\quad\quad$ Set $N(s, a) \leftarrow N(s, a) + 1$, $\theta(s, a) \leftarrow \theta(s, a) + c$, $N(s, a, s') \leftarrow N(s, a, s') + 1$.
13 $\quad\quad$ **if** $N(s, a) \in \mathcal{N}$ **then**
14 $\quad\quad\quad$ \\ *Update triggered:* `VISGO` *procedure.*
15 $\quad\quad\quad$ Set $\widehat{c}(s, a) \leftarrow \mathbb{I}[N(s, a) \geq 2]\frac{2\theta(s,a)}{N(s,a)} + \mathbb{I}[N(s, a) = 1]\theta(s, a)$ and $\theta(s, a) \leftarrow 0$.
16 $\quad\quad\quad$ For $s' \in \mathcal{S}'$, set $\widehat{P}_{s,a,s'} \leftarrow N(s, a, s')/N(s, a)$, $n(s, a) \leftarrow N(s, a)$, and $\widetilde{P}_{s,a,s'}$ as in Eq. 5.
17 $\quad\quad\quad$ Set $j \leftarrow j + 1$, $\epsilon_{\text{VI}} \leftarrow 2^{-j}/(SA)$ and $i \leftarrow 0$, $V^{(0)} \leftarrow 0$, $V^{(-1)} \leftarrow +\infty$.
18 $\quad\quad\quad$ For all $(s, a) \in \mathcal{S} \times \mathcal{A}$, set $n^+(s, a) \leftarrow \max\{n(s, a), 1\}$ and $\iota_{s,a} \leftarrow \ln\left(\frac{12SAS'[n^+(s,a)]^2}{\delta}\right)$.
19 $\quad\quad\quad$ **while** $\|V^{(i)} - V^{(i-1)}\|_\infty > \epsilon_{\text{VI}}$ **do**
20 $\quad\quad\quad\quad$ For all $(s, a) \in \mathcal{S} \times \mathcal{A}$, set

$$b^{(i+1)}(s, a) \leftarrow b(V^{(i)}, s, a), \qquad \text{\\ see Eq. 6 for bonus expression} \tag{2}$$

$$Q^{(i+1)}(s, a) \leftarrow \max\{\widehat{c}(s, a) + \widetilde{P}_{s,a}V^{(i)} - b^{(i+1)}(s, a), 0\}, \tag{3}$$

$$V^{(i+1)}(s) \leftarrow \min_a Q^{(i+1)}(s, a). \tag{4}$$

21 $\quad\quad\quad\quad$ Set $V^{(i+1)}(g) = 0$ and $i \leftarrow i + 1$.
22 $\quad\quad\quad$ Set $Q \leftarrow Q^{(i)}$, $V \leftarrow V^{(i)}$.

---

The main algorithmic component lies in how to compute the $Q$-values (w.r.t. which the policy is greedy) when a doubling condition is met. To this purpose, we introduce a procedure called `VISGO`, for `Value Iteration with Slight Goal Optimism`. Starting with optimistic values $V^{(0)} = 0$, it iteratively computes $V^{(i+1)} = \widetilde{\mathcal{L}}V^{(i)}$ for a carefully defined operator $\widetilde{\mathcal{L}}$. It ends when a stopping condition is met, specifically once $\|V^{(i+1)} - V^{(i)}\|_\infty \leq \epsilon_{\text{VI}}$ for a precision level $\epsilon_{\text{VI}} > 0$ (specified later), and it outputs the values $V^{(i+1)}$ (and $Q$-values $Q^{(i+1)}$). We now explain how we design $\widetilde{\mathcal{L}}$ and then provide some intuition. Let $\widehat{P}$ and $\widehat{c}$ be the current empirical transition probabilities and costs, and let $n(s, a)$ be the current number of visits to state-action pair $(s, a)$ (and $n^+(s, a) = \max\{n(s, a), 1\}$). We first define transition probabilities $\widetilde{P}$ that are slightly skewed towards the goal w.r.t. $\widehat{P}$, as follows

$$\widetilde{P}_{s,a,s'} := \frac{n(s, a)}{n(s, a) + 1}\widehat{P}_{s,a,s'} + \frac{\mathbb{I}[s' = g]}{n(s, a) + 1}. \tag{5}$$

Given the estimate $B$, specific positive constants $c_1, c_2, c_3, c_4$ and a state-action dependent logarithmic term $\iota_{s,a}$, we then define the exploration bonus function, for any state-action pair $(s, a) \in \mathcal{S} \times \mathcal{A}$ and vector $V \in \mathbb{R}^{S'}$ such that $V(g) = 0$, as follows

$$b(V, s, a) := \max\left\{c_1\sqrt{\frac{\mathbb{V}(\widetilde{P}_{s,a}, V)\iota_{s,a}}{n^+(s, a)}}, c_2\frac{B\iota_{s,a}}{n^+(s, a)}\right\} + c_3\sqrt{\frac{\widehat{c}(s, a)\iota_{s,a}}{n^+(s, a)}} + c_4\frac{B\sqrt{S'\iota_{s,a}}}{n^+(s, a)}. \tag{6}$$

Note that the last term in Eq. 6 accounts for the skewing of $\widetilde{P}$ w.r.t. $\widehat{P}$ (see Lem. 14). Given the transitions $\widetilde{P}$ and exploration bonus $b$, we are ready to define the operator $\widetilde{\mathcal{L}}$ as

$$\widetilde{\mathcal{L}}V(s) := \max\left\{\min_{a \in \mathcal{A}}\{\widehat{c}(s, a) + \widetilde{P}_{s,a}V - b(V, s, a)\}, 0\right\}. \tag{7}$$

We see that $\widetilde{\mathcal{L}}$ promotes optimism in two different ways:

**(i)** On the empirical cost function $\widehat{c}$, via the bonus $b$ (Eq. 6) that intuitively lowers the costs to $\widehat{c} - b$;

**(ii)** On the empirical transition function $\widehat{P}$, via the transitions $\widetilde{P}$ (Eq. 5) that slightly bias $\widehat{P}$ with the addition of a non-zero probability of reaching the goal from *every* state-action pair.

While the first feature **(i)** is standard in finite-horizon approaches, the second **(ii)** is SSP-specific, and is required to cope with the fact that the empirical model $\widehat{P}$ may *not* admit any proper policy, meaning that executing value iteration for SSP on $\widehat{P}$ may diverge. Our simple transition skewing actually guarantees that *all* policies are proper in $\widetilde{P}$, for any fixed and bounded cost function.[6] By decaying the extra goal-reaching probability inversely with $n(s, a)$, we can tightly control the gap between $\widetilde{P}$ and $\widehat{P}$ and ensure that it only accounts for a lower-order regret term (cf. last term of Eq. 6).

Equipped with these two sources of optimism, as long as $B \geq B_\star$, we are able to prove that a VISGO procedure verifies the following two key properties:

**(1)** *Optimism:* VISGO outputs an optimistic estimator of the optimal $Q$-function at each iteration step, i.e., $Q^{(i)}(s, a) \leq Q^\star(s, a), \forall i \geq 0$,

**(2)** *Finite-time near-convergence:* VISGO terminates within a finite number of iteration steps (note that the final iterate $V^{(j)}$ approximates the fixed point of $\widetilde{\mathcal{L}}$ up to an error scaling with $\epsilon_{\mathrm{VI}}$).

To satisfy **(1)**, we derive similarly to MVP [Zhang et al., 2021a] a *monotonicity* property for the operator $\widetilde{\mathcal{L}}$, which is achieved by carefully tuning the constants $c_1, c_2, c_3, c_4$ in the bonus of Eq. 6. On the other hand, the requirement **(2)** is SSP-specific, since it is not needed in finite-horizon where value iteration requires exactly $H$ backward induction steps. *Without* bonuses, the design of $\widetilde{P}$ would have directly entailed that $\widetilde{\mathcal{L}}$ is contractive and convergent [Bertsekas, 1995]. However, our variance-aware exploration bonuses introduce a subtle correlation between value iterates (i.e., $b$ depends on $V$ in Eq. 6), which leads to a cost function that varies across iterates. By directly analyzing $\widetilde{\mathcal{L}}$, we establish that it is contractive with modulus $\rho := 1 - \nu < 1$, where $\nu := \min_{s,a} \widetilde{P}_{s,a,g} > 0$. This *contraction* property guarantees a polynomially bounded number of iterations before terminating, i.e., **(2)**.

**Remark 1** (Computational complexity). Denote by $T$ the accumulated time within the $K$ episodes. By the stopping condition $||V^{(i+1)} - V^{(i)}||_\infty \leq \epsilon_{\mathrm{VI}}$, the choice of $\epsilon_{\mathrm{VI}}$ and the $\rho$-contraction of the operator $\widetilde{\mathcal{L}}$ with $\rho \leq 1 - 1/T$, any VISGO procedure is guaranteed to stop at an iteration $i \leq \log(\max\{B_\star, 1\}/\epsilon_{\mathrm{VI}})/(1-\rho) = O(TSA \log(T \max\{B_\star, 1\}))$. Since there are at most $O(SA \log T)$ VISGO procedures, we see that the total computational complexity of EB-SSP is near-linear in $T$, where $T$ is bounded polynomially w.r.t. $K$ as shown in the various cases of Sect. 4.1 (see App. G for details). Therefore EB-SSP is computationally efficient. Note that its $\mathrm{poly}(K)$ complexity is a limitation shared by all existing parameter-free algorithms in SSP. On the other hand, the algorithm of Cohen et al. [2021] can obtain a $\log(K)$ computational complexity but only with $T_\star$ prior knowledge: without it, using the upper bound $T_\star \leq B_\star/c_{\min}$, where $c_{\min}^{-1}$ becomes $\mathrm{poly}(K)$ when applying the cost perturbation trick, also leads to $\mathrm{poly}(K)$ complexity. It is an interesting open question whether it is possible in SSP to have $\log(K)$ computational complexity while staying parameter-free.

## 4 Main Results

Besides ensuring the computational efficiency of EB-SSP, the properties of VISGO lay the foundations for our regret analysis (App. D) to yield the following general guarantee.

**Theorem 3.** *Assume that $B \geq \max\{B_\star, 1\}$ and that the conditions of Lem. 2 hold. Then with probability at least $1 - \delta$ the regret of EB-SSP (Alg. 1 with $\eta = 0$) can be bounded by*

$$R_K = O\left( \sqrt{(B_\star^2 + B_\star)SAK} \log\left( \frac{\max\{B_\star, 1\}SAT}{\delta} \right) + BS^2 A \log^2\left( \frac{\max\{B_\star, 1\}SAT}{\delta} \right) \right),$$

*with $T$ the accumulated time within the $K$ episodes.*

---

[6] In fact this transition skewing implies that an SSP problem defined on $\widetilde{P}$ is equivalent to a discounted RL problem, with a varying state-action dependent discount factor. Also note that for different albeit mildly related purposes, a perturbation trick is sometimes used in regret minimization for average-reward MDPs [e.g., Fruit et al., 2018, Qian et al., 2019], where a non-zero probability of reaching an arbitrary state at each state-action is added to guarantee that all policies are unichain and that value iteration variants nearly converge in finite-time.

Thm. 3 is an intermediate result for the regret of EB-SSP, as it depends on the *random and possibly unbounded* total number of steps $T$ executed over $K$ episodes, it requires the possibly restrictive second condition of Lem. 2, and it relies on the parameter $B$ being properly tuned. Nonetheless, it already displays interesting properties: **1)** The dependence on $T$ is limited to logarithmic terms; **2)** The parameter $B$ only affects the lower order term, while the main order term naturally scales with the exact range $B_\star$; **3)** Up to dependence on $T$, the main order term displays minimax optimal dependencies on $B_\star$, $S$, $A$, and $K$.

Throughout the rest of the section, we consider for ease of exposition that $B_\star \geq 1$.[7] For simplicity, when tuning the cost perturbations later, we assume as in prior works [e.g., Rosenberg et al., 2020, Chen et al., 2021, Chen and Luo, 2021] that the total number of episodes $K$ is known to the agent (this knowledge can be eliminated with the standard doubling trick).

***Proof idea of Thm. 3.*** We decompose the regret into three parts: $X_1$ (error on the optimistic $V$-values), $X_2$ (Bellman error) and $X_3$ (cost estimation error), and among them the major part is $X_2$. Later, $X_1$ and $X_2$ introduce the intermediate quantities $X_4$ (variance of the optimistic $V$-values) and $X_5$ (variance of the differences $V^\star - V$), which are bounded using the recursion technique generalized from Zhang et al. [2021a], where we normalize the values by $1/B_\star$ to avoid an exponential blow-up in the recursions. At a high-level, the key idea is to calculate errors of different orders, $F(1), F(2), \ldots, F(d), \ldots$ (see Lem. 24 and 25), and recursively bound $F(i)$'s variance by a sublinear function of $F(i+1)$. Throughout the proof, we bound quantities by solving inequalities that contain the unknown quantities on both sides, such as $X_3 \leq \widetilde{O}(\sqrt{X_3 + C_K})$ or $X_2 \leq \widetilde{O}(\sqrt{X_2 + C_K})$, where the random variable $C_K$ denotes the cumulative cost over the $K$ episodes. Indeed, the analysis at each time step $t$ brings out the instantaneous cost $c_t$ and it is important to combine them so that we can make $C_K$ appear explicitly. Ultimately, we obtain a regret bound scaling as $R_K = \widetilde{O}((\sqrt{B_\star} + 1)\sqrt{SAC_K})$. Since the regret in SSP is defined as $R_K = C_K - KV^\star(s_0)$, we obtain a quadratic inequality in $C_K$, which we solve to get the $\widetilde{O}(\sqrt{(B_\star^2 + B_\star)SAK})$ regret bound.

## 4.1 Regret Bounds for $B = B_\star$

First we assume that $B = B_\star$ (i.e., the agent has prior knowledge of $B_\star$) and we instantiate the regret achieved by EB-SSP under various conditions on the SSP model.

☐ ***Positive Costs.*** We first focus on the case of positive costs.

**Assumption 4.** *All costs are lower bounded by a constant $c_{\min} > 0$ which is unknown to the agent.*

Asm. 4 guarantees that the conditions of Lem. 2 hold. Moreover, denoting by $C$ the cumulative cost over $K$ episodes, the total time satisfies $T \leq C/c_{\min}$. By simplifying the bound of Thm. 3 as $C \leq B_\star K + R_K \leq O(B_\star S^2 AK \cdot \sqrt{B_\star TSA/\delta})$, we loosely obtain that $T = O(B_\star^3 S^5 A^3 K^2/(c_{\min}^2 \delta))$.

**Corollary 5.** *Under Asm. 4, running EB-SSP (Alg. 1) with $B = B_\star$ and $\eta = 0$ gives the following regret bound with probability at least $1 - \delta$*

$$R_K = O\left(B_\star \sqrt{SAK} \log\left(\frac{KB_\star SA}{c_{\min}\delta}\right) + B_\star S^2 A \log^2\left(\frac{KB_\star SA}{c_{\min}\delta}\right)\right).$$

The bound of Cor. 5 only depends polynomially on $K, S, A, B_\star$. We note that $T_\star \leq B_\star/c_{\min}$ and that this upper bound only appears in the logarithms. Under positive costs, the regret of EB-SSP is thus (nearly) **minimax** and **horizon-free**. Furthermore, in App. B we introduce an alternative assumption on the SSP problem (which is weaker than Asm. 4) that considers that there are no almost-sure zero-cost cycles. In this case also, the regret of EB-SSP is (nearly) minimax and horizon-free.

☐ ***General Costs and $T_\star$ Unknown.*** Now we handle the case of non-negative costs, with no assumption other than Asm. 1. We use a cost perturbation argument to generalize the results from positive to general costs (similar to Tarbouriech et al. [2020a], Rosenberg et al. [2020]). As reviewed in Sect. 2, this circumvents the second condition of Lem. 2 (which holds in the cost-perturbed MDP) and target the optimal proper policy in the original MDP up to a bias scaling with the cost perturbation. Indeed, running EB-SSP with costs $c_\eta(s, a) \leftarrow \max\{c(s, a), \eta\}$ for $\eta \in (0, 1]$ gives the bound of Cor. 5 with $c_{\min} \leftarrow \eta$, $B_\star \leftarrow B_\star + \eta T_\star$ and an additive bias of $\eta T_\star K$. We then pick $\eta$ to balance these terms.

---

[7]Otherwise, all later bounds hold by replacing $B_\star$ with $\max\{B_\star, 1\}$, except for the $B_\star$ factor in the leading term that becomes $\sqrt{B_\star}$. This matches the lower bound of Cohen et al. [2021] of $\Omega(\sqrt{B_\star SAK})$ for $B_\star < 1$.

**Corollary 6.** *Let* $L := \log\big(KT_\star SA\delta^{-1}\big)$. *Running* EB-SSP *(Alg. 1) with* $B = B_\star$ *and* $\eta = K^{-n}$ *for **any** choice of constant* $n > 1$ *gives the following regret bound with probability at least* $1 - \delta$

$$R_K = O\Big(nB_\star\sqrt{SAK}L \;+\; \frac{T_\star}{K^{n-1}} + \frac{nT_\star\sqrt{SAL}}{K^{n-1/2}} \;+\; n^2 B_\star S^2 A L^2\Big).$$

This bound can be decomposed as (i) a $\sqrt{K}$ leading term and (ii) an additive term that depends on $T_\star$ and vanishes as $K \to +\infty$ (we omit the last term that does not depend polynomially on either $K$ or $T_\star$). Note that the second term (ii) can be made as small as possible by increasing the choice of exponent $n$ in the cost perturbation, at the cost of the multiplicative constant $n$ in (i). Equipped only with Asm. 1, the regret of EB-SSP is thus (nearly) **minimax**, and it may be dubbed as *horizon-vanishing* when $K$ is given in advance, insofar as it contains an additive term that depends on $T_\star$ and that becomes negligible for large values of $K$ (if $K$ is unknown in advance, the application of the doubling trick yields an additive term (ii) scaling as $T_\star$). We now show that the trade-off between (i) and (ii) can be resolved with loose knowledge of $T_\star$ and leads to a horizon-free bound.

□ *General Costs and Order-Accurate Estimate of* $T_\star$ *Available.* We now consider that an order-accurate estimate of $T_\star$ is available. It may be a constant lower-bound approximation away from $T_\star$, or a polynomial upper-bound approximation away from $T_\star$.

**Assumption 7.** *The agent has prior knowledge of a quantity* $\overline{T}_\star$ *that verifies* $\frac{T_\star}{\upsilon} \leq \overline{T}_\star \leq \lambda T_\star^\zeta$ *for some unknown constants* $\upsilon, \lambda, \zeta \geq 1$. *(Note that* $\upsilon = \lambda = \zeta = 1$ *when* $T_\star$ *is known.)*

We now tune the cost perturbation $\eta$ using $\overline{T}_\star$. Specifically, selecting $\eta := (\overline{T}_\star K)^{-1}$ ensures that the bias satisfies $\eta T_\star K \leq \upsilon = O(1)$. We thus obtain the following guarantee (see App. C for the explicit dependencies on the *constant* terms $\upsilon, \lambda, \zeta$ which only appear as multiplicative and additive factors).

**Corollary 8.** *Under Asm. 7, running* EB-SSP *(Alg. 1) with* $B = B_\star$ *and* $\eta = (\overline{T}_\star K)^{-1}$ *gives the following regret bound with probability at least* $1 - \delta$

$$R_K = O\Big( B_\star\sqrt{SAK}\log\Big(\frac{KT_\star SA}{\delta}\Big) + B_\star S^2 A \log^2\Big(\frac{KT_\star SA}{\delta}\Big)\Big).$$

This bound depends polynomially on $K, S, A, B_\star$, and only logarithmically on $T_\star$. Thus under general costs with an order-accurate estimate of $T_\star$, EB-SSP's regret is (nearly) **minimax** and **horizon-free**.

We can compare Cor. 8 with the concurrent result of Cohen et al. [2021]. Their regret bound scales as $O(B_\star\sqrt{SAKL} + T_\star^4 S^2 AL^5)$ with $L = \log(KT_\star SA\delta^{-1})$ under the assumptions of known $T_\star$ and $B_\star$ (or tight upper bounds of them), which imply that the conditions of Cor. 8 hold. The bound of Cor. 8 is strictly tighter, since it always holds that $B_\star \leq T_\star$ and the gap between the two may be arbitrarily large (see e.g., App. A), especially when some instantaneous costs are very small.

### 4.2 Regret Bounds for Unknown $B_\star$ with Parameter-Free EB-SSP

We now introduce a parameter-free version of EB-SSP that bypasses the requirement of $B \geq B_\star$ (line 2 of Alg. 1). Note that the challenge of not knowing the range of the optimal value function does not appear in finite-horizon MDPs, where the bound $H$ (or 1 for Zhang et al. [2021a]) is assumed to be known to the agent. In SSP, if the agent does not have a valid estimate $B \geq B_\star$, then it may design an under-specified exploration bonus which cannot guarantee optimism. The case of unknown $B_\star$ is non-trivial: it appears impossible to properly estimate $B_\star$ (since some states may never be visited) and it is unclear how a standard doubling trick may be used.[8]

Parameter-free EB-SSP initializes a proxy $\widetilde{B} = 1$ and increases it over the learning interaction according to a carefully defined schedule. We need to ensure that the proxy $\widetilde{B}$ does not remain below $B^\star$ for too long, since in this case, the regret may keep growing linearly. Thus, our *first condition* to increase $\widetilde{B}$ is whenever a new episode $k$ begins, specifically we set $\widetilde{B} \leftarrow \max\{\widetilde{B}, \sqrt{k}/(S^{3/2}A^{1/2})\}$, which ensures that $\widetilde{B} \geq B^\star$ for large enough episodes. However, this is not enough: indeed notice that when $\widetilde{B} < B^\star$, the agent may never reach the goal and thus get *stuck* in the episode, so we cannot

---

[8]Note that Qian et al. [2019] raised an open question whether it is possible to design an exploration bonus strategy in a setting where no prior knowledge of the "optimal range" is available. Indeed their approach in average-reward MDPs relies on prior knowledge of an upper bound on the optimal bias span.

exclusively rely on the end of an episode as a trigger for increasing $\widetilde{B}$. Our *second condition* to increase $\widetilde{B}$ is to set $\widetilde{B} \leftarrow 2\widetilde{B}$ whenever the cumulative cost exceeds a carefully defined threshold (that depends on $\widetilde{B}$, $S$, $A$, $\delta$ and the current episode and time indexes $k$ and $t$, which are all computable quantities). Since the regret is upper bounded by the cumulative cost, this second condition prevents the learner from accumulating too large regret when $\widetilde{B} < B^\star$. Finally, we introduce a *third condition* to increase $\widetilde{B}$ in order to ensure the computational efficiency, since VISGO may diverge when $\widetilde{B} < B^\star$ (specifically, we track the range of the value $V^{(i)}$ at each VISGO iteration $i$ and if $\|V^{(i)}\|_\infty > \widetilde{B}$, then we terminate VISGO and increase $\widetilde{B} \leftarrow 2\widetilde{B}$). At a high-level, the analysis of the scheme proceeds as follows: we bound the regret by the cumulative cost when $\widetilde{B} < B^\star$ (first regime), and by the regret bound of Thm. 3 when $\widetilde{B} \geq B^\star$ (second regime). Note that this two-regime decomposition is only implicit (i.e., at the level of analysis), since the agent is unable to know in which regime it is (since $B^\star$ is unknown). The full pseudo-code and analysis of parameter-free EB-SSP is deferred to App. H.

**Theorem 9** (Extension of Theorem 3 to unknown $B_\star$)**.** *Assume the conditions of Lem. 2 hold. Then with probability at least $1 - \delta$ the regret of parameter-free EB-SSP (Alg. 2, App. H) can be bounded by*

$$R_K = O\left( R_K^\star \log\left( \frac{B_\star SAT}{\delta} \right) + B_\star^3 S^3 A \log^3\left( \frac{B_\star SAT}{\delta} \right) \right),$$

*where $T$ is the cumulative time within the $K$ episodes and $R_K^\star$ bounds the regret after $K$ episodes of EB-SSP in the case of known $B_\star$ (i.e., the bound of Thm. 3 with $B = B_\star$).*

Thm. 9 implies that we can remove the condition of $B \geq \max\{B_\star, 1\}$ in Thm. 3, i.e., we make the statement **parameter-free**. Hence, *all* the regret bounds from Sect. 4.1 in the case of known $B_\star$ (i.e., Cor. 5, 6, 8, 11) still hold up to additional logarithmic and lower-order terms when $B_\star$ is unknown.

## 5   Conclusion

We introduced EB-SSP, the first algorithm for online SSP to be *simultaneously* nearly minimax-optimal and parameter-free (i.e., it does not need to know $T_\star$ nor $B_\star$). Also in various cases its regret is nearly horizon-free with only a *logarithmic* dependence on $T_\star$, thus exponentially improving over existing bounds w.r.t. the dependence on $T_\star$, which may be arbitrarily larger than $B_\star$ when instantaneous costs are small. The horizon-free property is perhaps even more meaningful in the goal-oriented setting than in finite-horizon MDPs (with total reward bounded by 1) [e.g., Wang et al., 2020a, Zhang et al., 2021a,b], as we do *not* impose a known constraint on the total cost of a trajectory.

An interesting question raised by our paper is whether it is possible to simultaneously achieve minimax, parameter-free and horizon-free regret for SSP under general costs. Another direction can be to build on our approach (e.g., the VISGO procedure) to derive tight sample complexity bounds in SSP, which as explained by Tarbouriech et al. [2021] do not directly ensue from regret guarantees.

## Acknowledgement

SSD gratefully acknowledges the funding from NSF Award's IIS-2110170 and DMS-2134106.

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
