# Appendix

## Table of Contents

## A   $T_\star$ can be arbitrarily larger than $B_\star$, $S$, $A$

Here we provide a simple illustration that the inequality $B_\star \leq T_\star$ may be arbitrarily loose, which shows that scaling with $T_\star$ can be much worse than scaling with $B_\star$. Recall that $B_\star$ bounds the total expected cost of the optimal policy starting from any state, and $T_\star$ bounds the expected time-to-goal of the optimal policy from any state.

Let us consider an SSP instance whose optimal policy induces the absorbing Markov chain depicted in Fig. 1. It is easy to see that $B_\star = 1$ and that $T_\star = \Omega(S\, p_{\min}^{-1})$. Hence, the gap between $B_\star$ and $T_\star$ can grow arbitrarily large as $p_{\min} \to 0$.

This simple example illustrates the benefit of having a bound that is (nearly) *horizon-free* (cf. desired property 3 in Sect. 1). Indeed, a bound that is not horizon-free scales polynomially with $T_\star$ and thus with $p_{\min}^{-1}$, which may be arbitrarily large if $p_{\min} \to 0$. In contrast, a horizon-free bound only scales logarithmically with $p_{\min}^{-1}$ and can therefore be much tighter.

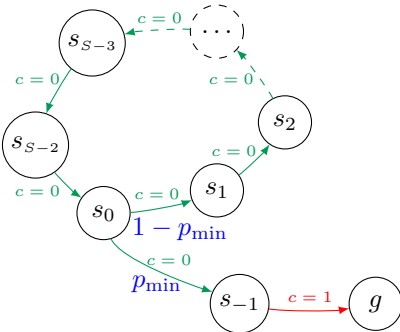

Figure 1: Markov chain of the optimal policy of an SSP instance with $S$ states. Transitions in green incur a cost of 0, while the transition in red leading to the goal state $g$ incurs a cost of 1. All transitions are deterministic, apart from the one starting from $s_0$, which reaches state $s_{-1}$ with probability $p_{\min}$ and state $s_1$ with probability $1 - p_{\min}$, where $p_{\min} > 0$.

## B   An Alternative Assumption on the SSP Problem: No Almost-Sure Zero-Cost Cycles

Here we complement Sect. 4.1 by introducing an alternative assumption on the SSP problem (which is weaker than Asm. 4) and we analyze the regret bound achieved by EB-SSP (under the set-up of Sect. 4.1). We draw inspiration from the common assumption in the deterministic shortest path setting that the transition graph does not possess any cycle of zero costs [Bertsekas, 1991]. In the following we introduce a "stochastic" counterpart of this assumption.

**Assumption 10.** *There exist unknown constants $c^\dagger > 0$ and $q^\dagger > 0$ such that:*

$$\mathbb{P}\left( \bigcap_{s' \in \mathcal{S}} \bigcap_{\omega \in \Omega_{s'}} \Big\{ \sum_{i=1}^{|\omega|} c_i \geq c^\dagger \Big\} \right) \geq q^\dagger,$$

*where for every state $s' \in \mathcal{S}$ we denote by $\Omega_{s'}$ the set of all possible trajectories in the SSP-MDP that start from state $s'$ and end in state $s'$, and we denote by $c_1, \ldots, c_{|\omega|}$ the sequence of costs incurred during a trajectory $\omega$.*

Asm. 10 is strictly weaker than the assumption of positive costs (Asm. 4) and it guarantees that the conditions of Lem. 2 hold. Intuitively, it implies that the agent has a non-zero probability of gradually accumulating some positive cost as its trajectory length increases. In particular, under Asm. 10, any trajectory of length $S + 1$ that does not reach the goal must accumulate costs of at least $c^\dagger$ with probability at least $q^\dagger$.

When $z \geq \ln(T/\delta)/q^\dagger \geq \frac{\ln(T/\delta)}{-\ln(1 - q^\dagger)}$, it is guaranteed that $(1 - q^\dagger)^z \leq \delta/T$. Repeatedly applying this argument means that with probability at least $1 - \delta/T$, for $z \geq \ln(T/\delta)/q^\dagger$ it holds that either $\sum_{i=1}^{z(S+1)} c_i \geq c^\dagger$, or the agent has reached the goal in the trajectory indexed by the time steps $[1, z(S+1)]$. Denote $z_0 := \lceil \ln(T/\delta)/q^\dagger \rceil$. For each episode, divide time steps in it into chunks with length $z_0(S+1)$, with the exception that the last chunk in it may have length less than or equal to $z_0(S+1)$ (just like taking modulo). So in each episode, the agent accumulates cost of at least $c^\dagger$ in each chunk except for the last one, and in the last chunk the agent reaches $g$. If we define $Z$ as the total number of chunks with cost at least $c^\dagger$ in all episodes, then $Z \geq \frac{T - K z_0(S+1)}{z_0(S+1)}$. Thus from $C \geq Z c^\dagger$ we have $T \leq O\left( \frac{S \log(T/\delta)}{q^\dagger} \left( \frac{C}{c^\dagger} + K \right) \right) \leq O(S(T/\delta)^{1/4} CK/(q^\dagger c^\dagger))$, with $C$ the cumulative cost. Using the loose bound $C \leq O(B_\star S^2 AK \cdot \sqrt{B_\star TSA/\delta})$ and isolating $T$ (with the same reasoning as in the case of positive costs in Sect. 4.1) gives that $T \leq O(B_\star^6 S^{14} A^6 K^8 / ((q^\dagger c^\dagger)^4 \delta^3))$ and thus that $\log T = O(\log(KB_\star SA/(c^\dagger q^\dagger \delta)))$. Plugging this in Thm. 3 yields the following.

**Corollary 11.** *Under Asm. 10, running* EB-SSP *(Alg. 1) with $B = B_\star \geq 1$ and $\eta = 0$ gives the following regret bound with probability at least $1 - \delta$*

$$R_K = O\left( B_\star \sqrt{SAK} \log\left( \frac{KB_\star SA}{c^\dagger q^\dagger \delta} \right) + B_\star S^2 A \log^2\left( \frac{KB_\star SA}{c^\dagger q^\dagger \delta} \right) \right).$$

The regret bound of Cor. 11 is (nearly) **minimax** and **horizon-free** (and it can be made **parameter-free** by executing Alg. 2 instead of Alg. 1). The bound depends logarithmically on the inverse of the constants $c^\dagger$, $q^\dagger$. We observe that i) it no longer becomes relevant if one constant is exponentially small, ii) spelling out $c^\dagger$, $q^\dagger$ satisfying Asm. 10 is challenging as they subtly depend on both the cost function and the transition dynamics, although iii) the agent does not need to know nor estimate $c^\dagger$ and $q^\dagger$ to achieve the regret bound of Cor. 11.

## C   Full Statement of Corollary 8

Here we make explicit the *constant* terms $\upsilon, \lambda, \zeta$ in the regret bound of Cor. 8.

Recall that Asm. 7 considers that the agent has prior knowledge of a quantity $\overline{T}_\star$ that verifies $T_\star/\upsilon \leq \overline{T}_\star \leq \lambda T_\star^\zeta$ for some unknown constants $\upsilon, \lambda, \zeta \geq 1$ (note that $\upsilon = \lambda = \zeta = 1$ when $T_\star$ is known). Under Asm. 7, running EB-SSP (Alg. 1) with $B = B_\star$ and $\eta = (\overline{T}_\star K)^{-1}$ gives the following regret bound with probability at least $1 - \delta$

$$R_K = O\left( \left( B_\star + \frac{\nu}{K} \right) \sqrt{SAK} \zeta \log\left( \frac{\lambda K T_\star SA}{\delta} \right) + \left( B_\star + \frac{\nu}{K} \right) S^2 A \zeta^2 \log^2\left( \frac{\lambda K T_\star SA}{\delta} \right) + \nu \right).$$

# D  Proof of Theorem 3

In this section, we present the proof of Thm. 3 (the missing proofs of the intermediate results within the section are deferred to App. E). We recall that throughout App. D we analyze Alg. 1 without cost perturbation (i.e., $\eta = 0$) and we assume that **1)** the estimate verifies $B \geq \max\{B_\star, 1\}$ and **2)** the conditions of Lem. 2 hold.

## D.1  High-Probability Event

**Definition 12** (High-probability event). *We define the event $\mathcal{E} := \mathcal{E}_1 \cap \mathcal{E}_2 \cap \mathcal{E}_3$, where*

$$\mathcal{E}_1 := \left\{ \forall (s,a) \in \mathcal{S} \times \mathcal{A}, \forall n(s,a) \geq 1 : |(\widehat{P}_{s,a} - P_{s,a})V^\star| \leq 2\sqrt{\frac{\mathbb{V}(\widehat{P}_{s,a}, V^\star)\iota_{s,a}}{n(s,a)}} + \frac{14 B_\star \iota_{s,a}}{3n(s,a)} \right\}, \quad (8)$$

$$\mathcal{E}_2 := \left\{ \forall (s,a) \in \mathcal{S} \times \mathcal{A}, \forall n(s,a) \geq 1 : |\widehat{c}(s,a) - c(s,a)| \leq 2\sqrt{\frac{2\widehat{c}(s,a)\iota_{s,a}}{n(s,a)}} + \frac{28\iota_{s,a}}{3n(s,a)} \right\}, \quad (9)$$

$$\mathcal{E}_3 := \left\{ \forall (s,a,s') \in \mathcal{S} \times \mathcal{A} \times \mathcal{S}', \forall n(s,a) \geq 1 : |P_{s,a,s'} - \widehat{P}_{s,a,s'}| \leq \sqrt{\frac{2P_{s,a,s'}\iota_{s,a}}{n(s,a)}} + \frac{\iota_{s,a}}{n(s,a)} \right\}, \quad (10)$$

*where $\iota_{s,a} := \ln\left( \frac{12 S A S' [n^+(s,a)]^2}{\delta} \right)$.*

**Lemma 13.** *It holds that $\mathbb{P}(\mathcal{E}) \geq 1 - \delta$.*

*Proof.* The events $\mathcal{E}_1$ and $\mathcal{E}_2$ hold with probability at least $1 - 2\delta/3$ by the concentration inequality of Lem. 27 and by union bound over all $(s,a) \in \mathcal{S} \times \mathcal{A}$. The event $\mathcal{E}_3$ holds with probability at least $1 - \delta/3$ by Bennett's inequality (Lem. 26, anytime version), by Lem. 33 and by union bound over all $(s,a,s') \in \mathcal{S} \times \mathcal{A} \times \mathcal{S}'$. □

## D.2  Analysis of a VISGO Procedure

A VISGO procedure in Alg. 1 computes iterates of the form $V^{(i+1)} = \widetilde{\mathcal{L}} V^{(i)}$, where $\widetilde{\mathcal{L}}$ is an operator that we define as follows. For any $U \in \mathbb{R}^{S'}$ such that $U(g) = 0$, we set $\widetilde{\mathcal{L}} U(g) := 0$ and for $s \in \mathcal{S}$ we set $\widetilde{\mathcal{L}} U(s) := \min_{a \in \mathcal{A}} \widetilde{\mathcal{L}} U(s,a)$, where

$$\widetilde{\mathcal{L}} U(s,a) := \max\left\{ \widehat{c}(s,a) + \widetilde{P}_{s,a} U - \max\left\{ c_1 \sqrt{\frac{\mathbb{V}(\widetilde{P}_{s,a}, U)\iota_{s,a}}{n^+(s,a)}}, \ c_2 \frac{B \iota_{s,a}}{n^+(s,a)} \right\} \right.$$

$$\left. - c_3 \sqrt{\frac{\widehat{c}(s,a)\iota_{s,a}}{n^+(s,a)}} - c_4 \frac{B\sqrt{S'\iota_{s,a}}}{n^+(s,a)}, \ 0 \right\}. \quad (11)$$

Starting from an optimistic initialization $V^{(0)} = 0$ at each state, we show the following two properties:

- *Optimism:* with high probability, $Q^{(i)}(s,a) \leq Q^\star(s,a), \forall i \geq 0$;

- *Finite-time near-convergence:* Given any error $\epsilon_{\mathrm{VI}} > 0$, the procedure stops at a *finite* iteration $j$ such that $\|V^{(j)} - V^{(j-1)}\|_\infty \leq \epsilon_{\mathrm{VI}}$, which implies that the vector $V^{(j)}$ verifies some fixed point equation for $\widetilde{\mathcal{L}}$ up to an error scaling with $\epsilon_{\mathrm{VI}}$.

### D.2.1  Properties of the slightly skewed transitions $\widetilde{P}$

Lem. 14 shows that the bias introduced by replacing $\widehat{P}_{s,a}$ with $\widetilde{P}_{s,a}$ decays inversely with $n(s,a)$, the number of visits to state-action pair $(s,a)$.

**Lemma 14.** *For any non-negative vector $U \in \mathbb{R}^{S'}$ such that $U(g) = 0$, for any $(s,a) \in \mathcal{S} \times \mathcal{A}$, it holds that*

$$\widetilde{P}_{s,a} U \leq \widehat{P}_{s,a} U \leq \widetilde{P}_{s,a} U + \frac{\|U\|_\infty}{n(s,a)+1}, \qquad \left| \mathbb{V}(\widetilde{P}_{s,a}, U) - \mathbb{V}(\widehat{P}_{s,a}, U) \right| \leq \frac{2\|U\|_\infty^2 S'}{n(s,a)+1}.$$

Denote by $\nu$ the probability of reaching the goal from any state-action pair in $\widetilde{P}$, i.e.,

$$\nu_{s,a} := \widetilde{P}_{s,a,g}, \qquad \nu := \min_{s,a} \nu_{s,a}. \tag{12}$$

By construction of $\widetilde{P}$, the quantity $\nu$ is strictly positive. This immediately implies the following result.

**Lemma 15.** *In the SSP-MDP associated to $\widetilde{P}$ with any bounded cost function, all policies are proper.*

**Remark 2** (Mapping to a discounted problem). In an SSP problem with only proper policies, the (optimal) Bellman operator is usually contractive only w.r.t. a weighted-sup norm [Bertsekas, 1995]. Here, the construction of $\widetilde{P}$ entails that any SSP defined on it with fixed bounded costs has a (optimal) Bellman operator that is a sup-norm contraction. In fact, the SSP problem on $\widetilde{P}$ can be cast as a discounted problem with a (state-action dependent) discount factor $\gamma_{s,a} := 1 - \nu_{s,a} < 1$ (we recall that discounted MDPs are a subclass of SSP-MDPs). Intuitively, at insufficiently visited state-action pairs, the agent behaves optimistically which increases the chance of reaching the goal and terminating the trajectory. Equivalently, we can interpret the agent as being uncertain about its future predictions and it is thus encouraged to act more myopically, which is connected to lowering the discount factor in the discounted RL setting.

### D.2.2 Important auxiliary function $f$ and its properties

Lem. 16 examines an auxiliary function $f$ that plays a key role in the analysis. Indeed, we see that an instantiation of $f$ surfaces in the definition of the operator $\widetilde{\mathcal{L}}$ (Eq. 11). While the first property (monotonicity) is similar to the one required in Zhang et al. [2021a], the third property (contraction) is SSP-specific and is crucial to guarantee the (finite-time) near-convergence of a VISGO procedure.

**Lemma 16.** *Let $\Upsilon := \{v \in \mathbb{R}^{S'} : v \geq 0, v(g) = 0, \|v\|_\infty \leq B\}$. Let $f : \Delta^{S'} \times \Upsilon \times \mathbb{R} \times \mathbb{R} \times \mathbb{R} \to \mathbb{R}$ with $f(p, v, n, B, \iota) := pv - \max\left\{ c_1 \sqrt{\frac{\mathbb{V}(p,v)\iota}{n}}, c_2 \frac{B\iota}{n} \right\}$, with $c_1 = 6$ and $c_2 = 36$ (here taking any pair of constants such that $c_1^2 \leq c_2$ works). Then $f$ satisfies, for all $p \in \Delta^{S'}$, $v \in \Upsilon$ and $n, \iota > 0$,*

1. *$f(p, v, n, B, \iota)$ is non-decreasing in $v(s)$, i.e.,*

$$\forall (v, v') \in \Upsilon^2, \ v \leq v' \implies f(p, v, n, B, \iota) \leq f(p, v', n, B, \iota);$$

2. *$f(p, v, n, B, \iota) \leq pv - \frac{c_1}{2}\sqrt{\frac{\mathbb{V}(p,v)\iota}{n}} - \frac{c_2}{2}\frac{B\iota}{n} \leq pv - 2\sqrt{\frac{\mathbb{V}(p,v)\iota}{n}} - 14\frac{B\iota}{n};$*

3. *If $p(g) > 0$, then $f(p, v, n, B, \iota)$ is $\rho_p$-contractive in $v(s)$, with $\rho_p := 1 - p(g) < 1$, i.e.,*

$$\forall (v, v') \in \Upsilon^2, \ |f(p, v, n, B, \iota) - f(p, v', n, B, \iota)| \leq \rho_p \|v - v'\|_\infty.$$

### D.2.3 Optimism of VISGO

We now show that with the bonus defined in Eq. 2, the $Q$-function is always optimistic with high probability.

**Lemma 17.** *Conditioned on the event $\mathcal{E}$, for any output $Q$ of the VISGO procedure (line 22 of Alg. 1) and for any state-action pair $(s, a) \in \mathcal{S} \times \mathcal{A}$, it holds that*

$$Q(s, a) \leq Q^\star(s, a).$$

*Proof idea.* We prove the result by induction on the inner iterations $i$ of VISGO, i.e., $Q^{(i)}(s, a) \leq Q^\star(s, a)$. We use the update of the $Q$-value (line 3), Lem. 14, the definition of event $\mathcal{E}$ combined with the fact that $B \geq B_\star$, as well as the first two properties of Lem. 16 applied to $f(\widetilde{P}_{s,a}, V^{(i)}, n^+(s, a), B, \iota_{s,a})$. $\qquad\square$

### D.2.4 Finite-time near-convergence of VISGO

***Warm-up: convergence with no bonuses.*** For the sake of discussion, let us first examine an idealized case where $n(s, a) \to +\infty$ for all $(s, a)$, which means $b(s, a) = 0$ for all $(s, a)$. In that case, the

iterates verify $V^{(i+1)} = \widetilde{\mathcal{L}}^\star V^{(i)}$, where $\widetilde{\mathcal{L}}^\star U(s) := \min_a \{c(s,a) + \widetilde{P}_{s,a} U\}, \forall U \in \mathbb{R}^S, s \in \mathcal{S}$. Thus $\widetilde{\mathcal{L}}^\star$ is the optimal Bellman operator of the SSP instance $\widetilde{M}$ with transitions $\widetilde{P}$ and cost function $c$. From Lem. 15, all policies are proper in $\widetilde{M}$. As a result, the operator $\widetilde{\mathcal{L}}^\star$ is contractive (cf. Remark 2) and convergent [Bertsekas, 1995].

***Convergence with bonuses.*** In VISGO, however, we must account for the bonuses $b(s,a)$. Setting aside the truncation of each iterate $V^{(i)}$ (i.e., the lower bounding by 0), we notice that a update for $V^{(i+1)}$ can be interpreted as the (truncated) Bellman operator of an SSP problem with cost function $c(s,a) - b^{(i+1)}(s,a)$. However, $b^{(i+1)}(s,a)$ depends on $V^{(i)}$, the previous iterate. This dependence means that the cost function is no longer fixed and the reasoning from the previous paragraph no longer holds. As a result, we directly analyze the properties of the operator $\widetilde{\mathcal{L}}$ that defines the sequence of iterates $V^{(i+1)} = \widetilde{\mathcal{L}} V^{(i)}$ in VISGO (Eq. 11).

**Lemma 18.** *The sequence $(V^{(i)})_{i \geq 0}$ is non-decreasing. Combining this with the fact that it is upper bounded by $V^\star$ from Lem. 17, the sequence must converge.*

While Lem. 18 states that $\widetilde{\mathcal{L}}$ ultimately converges starting from a vector of zeros, the following result guarantees that it can approximate in finite time its fixed point within any (arbitrarily small) positive component-wise accuracy.

**Lemma 19.** *Denote by $\nu > 0$ the probability of reaching the goal from any state-action pair in $\widetilde{P}$, i.e., $\nu := \min_{s,a} \widetilde{P}_{s,a,g}$. Then $\widetilde{\mathcal{L}}$ is a $\rho$-contractive operator with modulus $\rho := 1 - \nu < 1$.*

*Proof idea.* We can apply the third property (contraction) of Lem. 16 to $f(\widetilde{P}_{s,a}, V^{(i)}, n^+(s,a), B, \iota_{s,a})$, for any state-action pair $(s,a)$. Taking the maximum over $(s,a)$ pairs yields the contraction property of $\widetilde{\mathcal{L}}$. $\qquad\square$

**Remark 3.** Lem. 19 guarantees that $\|V^{(i+1)} - V^{(i)}\|_\infty \leq \epsilon_{\text{VI}}$ for $i \geq \frac{\log(\max\{B_\star, 1\}/\epsilon_{\text{VI}})}{1-\rho}$, which yields the desired property of finite-time near-convergence of VISGO (i.e., it always stops at a finite iteration $i$). Moreover, by definition of $\epsilon_{\text{VI}}$ we have $\log(1/\epsilon_{\text{VI}}) = O(SA \log(T))$, the (possibly loose) lower bound $1 - \rho = \nu \geq \frac{1}{T+1}$, and there are at most $O(SA \log T)$ VISGO procedures in total, thus we see that EB-SSP has a polynomially bounded computational complexity.

### D.3 Interval Decomposition and Notation

**Interval decomposition.** In the analysis we split the time steps into *intervals*. The first interval begins at the first time step, and an interval ends once either (1) the goal state $g$ is reached; (2) or the trigger condition holds (i.e., the visit to a state-action pair is doubled). We see that an update is triggered (line 13 of Alg. 1) whenever condition (2) is met.

**Notation.** We index intervals by $m = 1, 2, \ldots$ and the length of interval $m$ is denoted by $H^m$ (it is bounded almost surely). The trajectory visited in interval $m$ is denoted by $U^m = (s_1^m, a_1^m, \ldots, s_{H^m}^m, a_{H^m}^m, s_{H^m+1}^m)$, where $a_h^m$ is the action taken in state $s_h^m$. The concatenation of the trajectories of the intervals up to and including interval $m$ is denoted by $\overline{U}^m$, i.e., $\overline{U}^m = \bigcup_{m'=1}^m U^{m'}$. Moreover, $c_h^m$ denotes the cost in the $h$-th step of interval $m$. We use the notation $Q^m(s,a), V^m(s)$, $\widehat{P}_{s,a}^m, \widetilde{P}_{s,a}^m$ and $\epsilon_{\text{VI}}^m$ to denote the values (computed in lines 14-22) of $Q(s,a), V(s), \widehat{P}_{s,a}, \widetilde{P}_{s,a}$ and $\epsilon_{\text{VI}}$ in the beginning of interval $m$. Let $n^m(s,a)$ and $\widehat{c}^m(s,a)$ denote the values of $\max\{n(s,a), 1\}$ and $\widehat{c}(s,a)$ used for computing $Q^m(s,a)$. Finally, we set

$$b^m(s,a) := \max\left\{ c_1 \sqrt{\frac{\mathbb{V}(\widetilde{P}_{s,a}, V^m)\iota_{s,a}}{n^m(s,a)}}, \ c_2 \frac{B\iota_{s,a}}{n^m(s,a)} \right\} + c_3 \sqrt{\frac{\widehat{c}^m(s,a)\iota_{s,a}}{n^m(s,a)}} + c_4 \frac{B\sqrt{S'\iota_{s,a}}}{n^m(s,a)}.$$

### D.4 Bounding the Bellman Error

**Lemma 20.** *Conditioned on the event $\mathcal{E}$, for any interval $m$ and state-action pair $(s,a) \in \mathcal{S} \times \mathcal{A}$,*

$$|c(s,a) + P_{s,a} V^m - Q^m(s,a)| \leq \min\{\beta^m(s,a), B_\star + 1\},$$

*where we define*

$$\beta^m(s,a) := 4b^m(s,a) + \sqrt{\frac{2\mathbb{V}(P_{s,a},V^\star)\iota_{s,a}}{n^m(s,a)}} + \sqrt{\frac{2S'\mathbb{V}(P_{s,a},V^\star - V^m)\iota_{s,a}}{n^m(s,a)}}$$
$$+ \frac{3B_\star S'\iota_{s,a}}{n^m(s,a)} + \left(1 + c_1\sqrt{\iota_{s,a}/2}\right)\epsilon^m_{\mathrm{VI}}.$$

*Proof idea.* We use that $V^m$ approximates the fixed point of $\widetilde{\mathcal{L}}$ up to an error scaling with $\epsilon_{\mathrm{VI}}$. We end up decomposing and bounding the difference $P_{s,a}V^m - \widetilde{P}_{s,a}V^m \leq (\widehat{P}_{s,a} - \widetilde{P}_{s,a})V^m + (P_{s,a} - \widehat{P}_{s,a})V^\star + (P_{s,a} - \widehat{P}_{s,a})(V^m - V^\star)$, where the first term is bounded by Lem. 14 and 17, while the second and third terms are bounded using the definition of the event $\mathcal{E}$. $\square$

### D.5 Regret Decomposition

We assume that the event $\mathcal{E}$ defined in Def. 12 holds. In particular it guarantees that Lem. 17 and Lem. 20 hold for all intervals $m$ simultaneously.

We denote by $M$ the total number of intervals in which the first $K$ episodes elapse. For any $M' \leq M$, we denote by $\mathcal{M}_0(M')$ the set of intervals which are among the first $M'$ intervals, and constitute the first intervals in each episode (i.e., either it is the first interval or its previous interval ended in the goal state). We also denote by $K_{M'} := |\mathcal{M}_0(M')|$, $T_{M'} := \sum_{m=1}^{M'} H^m$ and $C_{M'} := \sum_{m=1}^{M'} \sum_{h=1}^{H^m} c_h^m$. Note that $K$ and $T$ are equivalent to $K_M$ and $T_M$, respectively, and $C_{M'}$ is the cumulative cost in the first $M'$ intervals.

Instead of bounding the regret $R_K$ from Eq. 1, we bound $\widetilde{R}_{M'} := C_{M'} - K_{M'}V^\star(s_0)$ for any fixed choice of $M' \leq M$, as done in Rosenberg et al. [2020]. We see that $\widetilde{R}_M = R_K$, the true regret within $K$ episodes. To derive Thm. 3, we will show that $M$ is finite and instantiate $M' = M$. In the following we do the analysis for arbitrary $M' \leq M$ as it will be useful for the parameter-free case studied in App. H (i.e., when no estimate $B \geq B_\star$ is available).

We decompose $\widetilde{R}_{M'}$ as follows

$$\widetilde{R}_{M'} \overset{(i)}{\leq} \sum_{m=1}^{M'}\sum_{h=1}^{H^m} c_h^m - \sum_{m \in \mathcal{M}_0(M')} V^m(s_0),$$

$$\overset{(ii)}{\leq} \sum_{m=1}^{M'}\sum_{h=1}^{H^m} c_h^m + \sum_{m=1}^{M'}\left(\sum_{h=1}^{H^m} V^m(s_{h+1}^m) - V^m(s_h^m)\right) + 2SA\log_2(T_{M'})\max_{1 \leq m \leq M'}\|V^m\|_\infty$$

$$\overset{(iii)}{\leq} \sum_{m=1}^{M'}\sum_{h=1}^{H^m}\left[c_h^m + P_{s_h^m,a_h^m}V^m - V^m(s_h^m)\right] + \sum_{m=1}^{M'}\sum_{h=1}^{H^m}\left[V^m(s_{h+1}^m) - P_{s_h^m,a_h^m}V^m\right]$$
$$+ 2B_\star SA\log_2(T_{M'})$$

$$\overset{(iv)}{\leq} \underbrace{\sum_{m=1}^{M'}\sum_{h=1}^{H^m}\left[V^m(s_{h+1}^m) - P_{s_h^m,a_h^m}V^m\right]}_{:=X_1(M')} + \underbrace{\sum_{m=1}^{M'}\sum_{h=1}^{H^m}\beta^m(s_h^m,a_h^m)}_{:=X_2(M')} + \underbrace{\sum_{m=1}^{M'}\sum_{h=1}^{H^m} c_h^m - c(s_h^m,a_h^m)}_{:=X_3(M')}$$
$$+ 2B_\star SA\log_2(T_{M'}),$$

where (i) uses the optimism property of Lem. 17, (ii) stems from the construction of intervals (Lem. 22), (iii) uses that $\max_{1 \leq m \leq M'}\|V^m\|_\infty \leq B_\star$ (from Lem. 17), and (iv) comes from Lem. 20. We now focus on bounding the terms $X_1(M')$, $X_2(M')$ and $X_3(M')$. To this end, we introduce the following useful quantities

$$X_4(M') := \sum_{m=1}^{M'}\sum_{h=1}^{H^m}\mathbb{V}(P_{s_h^m,a_h^m},V^m), \qquad X_5(M') := \sum_{m=1}^{M'}\sum_{h=1}^{H^m}\mathbb{V}(P_{s_h^m,a_h^m},V^\star - V^m).$$

### D.5.1 The $X_1(M')$ term

$X_1(M')$ could be viewed as a martingale, so by taking $c = \max\{B_\star, 1\}$ in the technical Lem. 30, we have with probability at least $1 - \delta$,

$$|X_1(M')| \leq 2\sqrt{2X_4(M')(\log_2((\max\{B_\star, 1\})^2 T_{M'}) + \ln(2/\delta))}$$
$$+ 5(\max\{B_\star, 1\})(\log_2((\max\{B_\star, 1\})^2 T_{M'}) + \ln(2/\delta)).$$

To bound $X_1(M')$, we only need to bound $X_4(M')$.

### D.5.2 The $X_3(M')$ term

Taking $c = 1$ in the technical Lem. 30, we have

$$\mathbb{P}\left[|X_3(M')| \geq 2\sqrt{2\sum_{m=1}^{M'}\sum_{h=1}^{H^m}\text{Var}(s_h^m, a_h^m)(\log_2(T_{M'}) + \ln(2/\delta)) + 5(\log_2(T_{M'}) + \ln(2/\delta))}\right] \leq \delta,$$

where $\text{Var}(s_t, a_t) := \mathbb{E}[(c_t - c(s_t, a_t))^2]$ ($c_t$ denotes the cost incurred at time step $t$). By Lem. 33,

$$\sum_{m=1}^{M'}\sum_{h=1}^{H^m}\text{Var}(s_h^m, a_h^m) \leq \sum_{m=1}^{M'}\sum_{h=1}^{H^m}c(s_h^m, a_h^m)$$
$$\leq \sum_{m=1}^{M'}\sum_{h=1}^{H^m}(c(s_h^m, a_h^m) - c_h^m) + C_{M'}$$
$$\leq |X_3(M')| + C_{M'}.$$

Therefore we have

$$\mathbb{P}\left[|X_3(M')| \geq 2\sqrt{2(|X_3(M')| + C_{M'})(\log_2(T_{M'}) + \ln(2/\delta)) + 5(\log_2(T_{M'}) + \ln(2/\delta))}\right] \leq \delta,$$

which implies that $|X_3(M')| \leq O\left(\log_2(T_{M'}) + \ln(2/\delta) + \sqrt{C_{M'}(\log_2(T_{M'}) + \ln(2/\delta))}\right)$ with probability at least $1 - \delta$.

### D.5.3 The $X_2(M')$ term

The full proof of the bound on $X_2(M')$ is deferred to App. E.3. Here we provide a brief sketch. First, we bound $\beta^m$ and apply a pigeonhole principle to obtain

$$X_2(M') \leq O\left(\sqrt{SA\log_2(T_{M'})\iota_{M'}X_4(M')} + \sqrt{S^2 A\log_2(T_{M'})\iota_{M'}X_5(M')}\right.$$

$$+ \sqrt{SA\log_2(T_{M'})\iota_{M'}\sum_{m=1}^{M'}\sum_{h=1}^{H^m}\widehat{c}^m(s_h^m, a_h^m)}$$

$$\left. + B_\star S^2 A\log_2(T_{M'}) + BS^{3/2}A\log_2(T_{M'})\iota_{M'} + \sum_{m=1}^{M'}\sum_{h=1}^{H^m}(1 + c_1\sqrt{\iota_{M'}/2})\epsilon_{\text{VI}}^m\right)$$

with the logarithmic term $\iota_{M'} := \ln\left(\frac{12SAS'T_{M'}^2}{\delta}\right)$ which is the upper-bound of $\iota_{s,a}$ when considering only time steps in the first $M'$ intervals. The regret contributions of the estimated costs and the VISGO precision errors are respectively

$$\sum_{m=1}^{M'}\sum_{h=1}^{H^m}\widehat{c}^m(s_h^m, a_h^m) \leq 2SA(\log_2(T_{M'}) + 1) + 2C_{M'},$$

$$\sum_{m=1}^{M'}\sum_{h=1}^{H^m}(1 + c_1\sqrt{\iota_{M'}/2})\epsilon_{\text{VI}}^m = O(SA\log_2(T_{M'})\sqrt{\iota_{M'}}).$$

To bound $X_4(M')$ and $X_5(M')$, we perform a recursion-based analysis on the value functions normalized by $1/B_\star$. We split the analysis on the intervals, and not on the episodes as done in Zhang et al. [2021a]. In Lem. 24 and 25 we establish that with overwhelming probability,

$$X_4(M') \leq O\big(B_\star(C_{M'} + X_2(M')) + (B_\star^2 SA + B_\star)(\log_2(T_{M'}) + \ln(2/\delta)))\big),$$
$$X_5(M') \leq O\big(B_\star^2 SA(\log_2(T_{M'}) + \ln(2/\delta)) + B_\star X_2(M')\big).$$

As a result, we obtain

$$X_2(M') \leq O\Big(\sqrt{SAX_4(M')}\bar\iota_{M'} + \sqrt{S^2AX_5(M')}\bar\iota_{M'}$$
$$+ SA\bar\iota_{M'}^{3/2} + \sqrt{SAC_{M'}}\bar\iota_{M'} + B_\star S^2 A\bar\iota_{M'}^2 + BS^{3/2}A\bar\iota_{M'}^2\Big),$$
$$X_4(M') \leq O\big(B_\star(C_{M'} + X_2(M')) + (B_\star^2 SA + B_\star)\bar\iota_{M'}\big),$$
$$X_5(M') \leq O\big(B_\star^2 SA\bar\iota_{M'} + B_\star X_2(M')\big).$$

with the logarithmic term $\bar\iota_{M'} := \ln\left(\frac{12SAS'T_{M'}^2}{\delta}\right) + \log_2((\max\{B_\star, 1\})^2 T_{M'}) + \ln\left(\frac{2}{\delta}\right)$. Isolating the $X_2(M')$ term finally yields

$$X_2(M') \leq O((\sqrt{B_\star} + 1)\sqrt{SAC_{M'}}\bar\iota_{M'} + BS^2A\bar\iota_{M'}^2).$$

### D.5.4 Putting Everything Together

Ultimately, with probability at least $1 - 6\delta$ we have

$$\widetilde{R}_{M'} \leq X_1(M') + X_2(M') + X_3(M') + 2B_\star SA\log_2(T_{M'})$$
$$\leq O((\sqrt{B_\star} + 1)\sqrt{SAC_{M'}}\bar\iota_{M'} + BS^2A\bar\iota_{M'}^2).$$

Noting that $\widetilde{R}_{M'} = C_{M'} - K_{M'}V^\star(s_0)$, we have

$$C_{M'} \leq K_{M'}V^\star(s_0) + O((\sqrt{B_\star} + 1)\sqrt{SAC_{M'}}\bar\iota_{M'} + BS^2A\bar\iota_{M'}^2),$$
$$C_{M'} \overset{(i)}{\leq} \left(O\big((\sqrt{B_\star} + 1)\sqrt{SA}\bar\iota_{M'}\big) + \sqrt{K_{M'}V^\star(s_0) + O(BS^2A\bar\iota_{M'}^2)}\right)^2$$
$$\leq K_{M'}V^\star(s_0) + O\big((\sqrt{B_\star} + 1)\sqrt{V^\star(s_0)SAK_{M'}}\bar\iota_{M'} + BS^2A\bar\iota_{M'}^2\big)$$
$$\leq K_{M'}V^\star(s_0) + O\big((B_\star + \sqrt{B_\star})\sqrt{SAK_{M'}}\bar\iota_{M'} + BS^2A\bar\iota_{M'}^2\big),$$

where (i) uses Lem. 35, $V^\star(s_0) \leq B_\star$ and $\sqrt{B_\star} + 1 \leq O(\sqrt{B_\star + 1}) \leq O(\sqrt{B})$. Hence

$$\widetilde{R}_{M'} \leq O\big(\sqrt{(B_\star^2 + B_\star)SAK_{M'}}\bar\iota_{M'} + BS^2A\bar\iota_{M'}^2\big).$$

By scaling $\delta \leftarrow \delta/6$ we have the following important bound

$$\widetilde{R}_{M'} \leq O\Bigg(\sqrt{(B_\star^2 + B_\star)SAK_{M'}}\log\left(\frac{\max\{B_\star, 1\}SAT_{M'}}{\delta}\right)$$
$$+ BS^2A\log^2\left(\frac{\max\{B_\star, 1\}SAT_{M'}}{\delta}\right)\Bigg). \tag{13}$$

The proof of Thm. 3 is concluded by taking $M' = M$, where $M$ denotes the number of intervals in which the first $K$ episodes elapse.

## E Missing Proofs

### E.1 Proofs of Lemmas 14, 16, 17, 18, 19, 20

**Restatement of Lemma 14.** For any non-negative vector $U \in \mathbb{R}^{S'}$ such that $U(g) = 0$, for any $(s, a) \in \mathcal{S} \times \mathcal{A}$, it holds that

$$\widetilde{P}_{s,a}U \leq \widehat{P}_{s,a}U \leq \widetilde{P}_{s,a}U + \frac{\|U\|_\infty}{n(s, a) + 1}, \qquad \big|\mathbb{V}(\widetilde{P}_{s,a}, U) - \mathbb{V}(\widehat{P}_{s,a}, U)\big| \leq \frac{2\|U\|_\infty^2 S'}{n(s, a) + 1}.$$

*Proof.* The proof uses the definition of $\widetilde{P}$ (Eq. 5) and simple algebraic manipulation. For any $s' \neq g$, we have $\widetilde{P}_{s,a,s'} \leq \widehat{P}_{s,a,s'}$ and $U(s') \geq 0$, as well as $U(g) = 0$, so $\widetilde{P}_{s,a}U \leq \widehat{P}_{s,a}U$, and

$$(\widehat{P}_{s,a} - \widetilde{P}_{s,a})U = \left(1 - \frac{n(s,a)}{n(s,a)+1}\right)\widehat{P}_{s,a}U \leq \frac{\|U\|_\infty}{n(s,a)+1}.$$

In addition, for any $s' \in \mathcal{S}'$,

$$|\widetilde{P}_{s,a,s'} - \widehat{P}_{s,a,s'}| \leq \left|\frac{n(s,a)}{n(s,a)+1} - 1\right|\widehat{P}_{s,a,s'} + \frac{\mathbb{I}[s'=g]}{n(s,a)+1} \leq \frac{2}{n(s,a)+1}.$$

Therefore we have that

$$\mathbb{V}(\widehat{P}_{s,a}, U) = \sum_{s'\in\mathcal{S}'} \widehat{P}_{s,a,s'}(U(s') - \widehat{P}_{s,a}U)^2 \leq \sum_{s'\in\mathcal{S}'} \widehat{P}_{s,a,s'}(U(s') - \widetilde{P}_{s,a}U)^2$$

$$\leq \sum_{s'\in\mathcal{S}'} \left(\widetilde{P}_{s,a,s'} + \frac{2}{n(s,a)+1}\right)(U(s') - \widetilde{P}_{s,a}U)^2 \leq \mathbb{V}(\widetilde{P}_{s,a}, U) + \frac{2\|U\|_\infty^2 S'}{n(s,a)+1},$$

where the first inequality is by the fact that $z^\star = \sum_i p_i x_i$ minimizes the quantity $\sum_i p_i(x_i - z)^2$. Conversely,

$$\mathbb{V}(\widetilde{P}_{s,a}, U) = \sum_{s'\in\mathcal{S}'} \widetilde{P}_{s,a,s'}(U(s') - \widetilde{P}_{s,a}U)^2 \leq \sum_{s'\in\mathcal{S}'} \widetilde{P}_{s,a,s'}(U(s') - \widehat{P}_{s,a}U)^2$$

$$\leq \sum_{s'\in\mathcal{S}'} \left(\widehat{P}_{s,a,s'} + \frac{2}{n(s,a)+1}\right)(U(s') - \widehat{P}_{s,a}U)^2 \leq \mathbb{V}(\widehat{P}_{s,a}, U) + \frac{2\|U\|_\infty^2 S'}{n(s,a)+1}.$$

$\square$

**Restatement of Lemma 16.** Let $\Upsilon := \{v \in \mathbb{R}^{S'} : v \geq 0,\ v(g) = 0,\ \|v\|_\infty \leq B\}$. Let $f : \Delta^{S'} \times \Upsilon \times \mathbb{R} \times \mathbb{R} \times \mathbb{R} \to \mathbb{R}$ with $f(p, v, n, B, \iota) := pv - \max\left\{c_1\sqrt{\frac{\mathbb{V}(p,v)\iota}{n}},\ c_2\frac{B\iota}{n}\right\}$, with $c_1 = 6$ and $c_2 = 36$ (here taking any pair of constants such that $c_1^2 \leq c_2$ works). Then $f$ satisfies, for all $p \in \Delta^{S'}$, $v \in \Upsilon$ and $n, \iota > 0$,

1. $f(p, v, n, B, \iota)$ is non-decreasing in $v(s)$, i.e.,

$$\forall(v, v') \in \Upsilon^2,\ v \leq v' \implies f(p, v, n, B, \iota) \leq f(p, v', n, B, \iota);$$

2. $f(p, v, n, B, \iota) \leq pv - \frac{c_1}{2}\sqrt{\frac{\mathbb{V}(p,v)\iota}{n}} - \frac{c_2}{2}\frac{B\iota}{n} \leq pv - 2\sqrt{\frac{\mathbb{V}(p,v)\iota}{n}} - 14\frac{B\iota}{n}$;

3. If $p(g) > 0$, then $f(p, v, n, B, \iota)$ is $\rho_p$-contractive in $v(s)$, with $\rho_p := 1 - p(g) < 1$, i.e.,

$$\forall(v, v') \in \Upsilon^2,\ |f(p, v, n, B, \iota) - f(p, v', n, B, \iota)| \leq \rho_p\|v - v'\|_\infty.$$

*Proof.* The second claim holds by $\max\{x, y\} \geq (x + y)/2, \forall x, y$, by the choices of $c_1, c_2$ and because both $\sqrt{\frac{\mathbb{V}(p,v)\iota}{n}}$ and $\frac{B\iota}{n}$ are non-negative. To verify the first and third claims, we fix all other variables but $v(s)$ and view $f$ as a function in $v(s)$. Because the derivative of $f$ in $v(s)$ does not exist only when $c_1\sqrt{\frac{\mathbb{V}(p,v)\iota}{n}} = c_2\frac{B\iota}{n}$, where the condition has at most two solutions, it suffices to prove that $\frac{\partial f}{\partial v(s)} \geq 0$ when $c_1\sqrt{\frac{\mathbb{V}(p,v)\iota}{n}} \neq c_2\frac{B\iota}{n}$. Direct computation gives

$$\frac{\partial f}{\partial v(s)} = p(s) - c_1\mathbb{I}\left[c_1\sqrt{\frac{\mathbb{V}(p,v)\iota}{n}} \geq c_2\frac{B\iota}{n}\right]\frac{p(s)(v(s) - pv)\iota}{\sqrt{n\mathbb{V}(p,v)\iota}}$$

$$\geq \min\left\{p(s),\ p(s) - \frac{c_1^2}{c_2 B}p(s)(v(s) - pv)\right\}$$

$$\overset{(i)}{\geq} \min\left\{p(s),\ p(s) - \frac{c_1^2}{c_2}p(s)\right\}$$

$$\geq p(s)\left(1 - \frac{c_1^2}{c_2}\right) = 0.$$

Here (i) is by $v(s) - pv \leq v(s) \leq B$. For the third claim, we perform a distinction of cases. If $c_1\sqrt{\frac{\mathbb{V}(p,v)\iota}{n}} = c_2\frac{B\iota}{n}$, where the condition has at most two solutions, then $f(v) = pv - c_2\frac{B\iota}{n}$, which corresponds to a $\rho_p$-contraction since

$$|f(v_1) - f(v_2)| = \left|\sum_{s\in\mathcal{S}} p(s)(v_1(s) - v_2(s))\right| \leq \sum_{s\in\mathcal{S}} p(s)\cdot\|v_1 - v_2\|_\infty = (1 - p(g))\|v_1 - v_2\|_\infty.$$

Otherwise $c_1\sqrt{\frac{\mathbb{V}(p,v)\iota}{n}} \neq c_2\frac{B\iota}{n}$, then the derivative of $f$ in $v(s)$ exists and it verifies

$$\left\|\frac{\partial f}{\partial v}\right\|_1 = \sum_{s\in\mathcal{S}}\left|\frac{\partial f}{\partial v(s)}\right| = \sum_{s\in\mathcal{S}}\frac{\partial f}{\partial v(s)}$$

$$= \sum_{s\in\mathcal{S}}\left[p(s) - c_1\mathbb{I}\left[c_1\sqrt{\frac{\mathbb{V}(p,v)\iota}{n}} \geq c_2\frac{B\iota}{n}\right]\frac{p(s)(v(s) - pv)\iota}{\sqrt{n\mathbb{V}(p,v)\iota}}\right]$$

$$= 1 - p(g) - c_1\mathbb{I}\left[c_1\sqrt{\frac{\mathbb{V}(p,v)\iota}{n}} \geq c_2\frac{B\iota}{n}\right]\sqrt{\frac{\iota}{n\mathbb{V}(p,v)}}[pv - (1 - p(g))\cdot pv]\}$$

$$\leq 1 - p(g).$$

In this case, by the mean value theorem we obtain that $f$ is $\rho_p$-contractive. □

**Restatement of Lemma 17.** Conditioned on the event $\mathcal{E}$, for any output $Q$ of the VISGO procedure (line 22 of Alg. 1) and for any state-action pair $(s,a) \in \mathcal{S} \times \mathcal{A}$, it holds that

$$Q(s,a) \leq Q^\star(s,a).$$

*Proof.* We prove by induction that for any inner iteration $i$ of VISGO, $Q^{(i)}(s,a) \leq Q^\star(s,a)$. By definition we have $Q^{(0)} = 0 \leq Q^\star$. Assume that the property holds for iteration $i$, then

$$Q^{(i+1)}(s,a) = \max\left\{\widehat{c}(s,a) + \widetilde{P}_{s,a}V^{(i)} - b^{(i+1)}(s,a), 0\right\},$$

where

$$\widehat{c}(s,a) + \widetilde{P}_{s,a}V^{(i)} - b^{(i+1)}(s,a)$$

$$= \widehat{c}(s,a) + \widetilde{P}_{s,a}V^{(i)} - \max\left\{c_1\sqrt{\frac{\mathbb{V}(\widetilde{P}_{s,a},V^{(i)})\iota_{s,a}}{n^+(s,a)}}, c_2\frac{B\iota_{s,a}}{n^+(s,a)}\right\} - c_3\sqrt{\frac{\widehat{c}(s,a)\iota_{s,a}}{n^+(s,a)}} - c_4\frac{B\sqrt{S'\iota_{s,a}}}{n^+(s,a)}$$

$$\overset{(i)}{\leq} c(s,a) + \widetilde{P}_{s,a}V^{(i)} - \max\left\{c_1\sqrt{\frac{\mathbb{V}(\widetilde{P}_{s,a},V^{(i)})\iota_{s,a}}{n^+(s,a)}}, c_2\frac{B\iota_{s,a}}{n^+(s,a)}\right\} + \frac{28\iota_{s,a}}{3n^+(s,a)} - c_4\frac{B\sqrt{S'\iota_{s,a}}}{n^+(s,a)}$$

$$= c(s,a) + f(\widetilde{P}_{s,a}, V^{(i)}, n^+(s,a), B, \iota_{s,a}) + \frac{28\iota_{s,a}}{3n^+(s,a)} - c_4\frac{B\sqrt{S'\iota_{s,a}}}{n^+(s,a)}$$

$$\overset{(ii)}{\leq} c(s,a) + f(\widetilde{P}_{s,a}, V^\star, n^+(s,a), B, \iota_{s,a}) + \frac{28\iota_{s,a}}{3n^+(s,a)} - c_4\frac{B\sqrt{S'\iota_{s,a}}}{n^+(s,a)}$$

$$\overset{(iii)}{\leq} c(s,a) + \widetilde{P}_{s,a}V^\star - 2\sqrt{\frac{\mathbb{V}(\widetilde{P}_{s,a},V^\star)\iota_{s,a}}{n^+(s,a)}} - \frac{14B\iota_{s,a}}{3n^+(s,a)} - c_4\frac{B\sqrt{S'\iota_{s,a}}}{n^+(s,a)}$$

$$\overset{(iv)}{\leq} c(s,a) + \widehat{P}_{s,a}V^\star - 2\sqrt{\frac{\mathbb{V}(\widehat{P}_{s,a},V^\star)\iota_{s,a}}{n^+(s,a)}} - \frac{14B\iota_{s,a}}{3n^+(s,a)} - c_4\frac{B\sqrt{S'\iota_{s,a}}}{n^+(s,a)}$$

$$\overset{(v)}{\leq} c(s,a) + P_{s,a}V^\star + 2\sqrt{\frac{\mathbb{V}(\widehat{P}_{s,a},V^\star)\iota_{s,a}}{n^+(s,a)}} - 2\sqrt{\frac{\mathbb{V}(\widetilde{P}_{s,a},V^\star)\iota_{s,a}}{n^+(s,a)}} - (B - B_\star)\frac{14\iota_{s,a}}{3n^+(s,a)} - c_4\frac{B\sqrt{S'\iota_{s,a}}}{n^+(s,a)}$$

$$\overset{(vi)}{\leq} c(s,a) + P_{s,a}V^\star + 2\sqrt{\frac{|\mathbb{V}(\widehat{P}_{s,a},V^\star) - \mathbb{V}(\widetilde{P}_{s,a},V^\star)|\iota_{s,a}}{n^+(s,a)}} - (B - B_\star)\frac{14\iota_{s,a}}{3n^+(s,a)} - c_4\frac{B\sqrt{S'\iota_{s,a}}}{n^+(s,a)}$$

$$\overset{\text{(vii)}}{\leq} \underbrace{c(s,a) + P_{s,a}V^\star}_{=Q^\star(s,a)} - (B - B_\star)\left(\frac{14\iota_{s,a}}{3n^+(s,a)} + \frac{2\sqrt{2S'\iota_{s,a}}}{n^+(s,a)}\right)$$

$$\leq Q^\star(s,a),$$

where (i) is by definition of $\mathcal{E}_2$ and choice of $c_3$, (ii) uses the first property of Lem. 16 and the induction hypothesis that $V^{(i)} \leq V^\star$, (iii) uses the second property of Lem. 16 and assumption $B \geq \max\{B_\star, 1\}$, (iv) uses Lem. 14, (v) is by definition of $\mathcal{E}_1$, (vi) uses the inequality $\left|\sqrt{x} - \sqrt{y}\right| \leq \sqrt{|x-y|}, \forall x, y \geq 0$, and (vii) uses the second inequality of Lem. 14 and the choice of $c_4$. Ultimately,

$$Q^{(i+1)}(s,a) \leq \max\left\{Q^\star(s,a), 0\right\} \leq Q^\star(s,a).$$

$\square$

**Restatement of Lemma 18.** The sequence $(V^{(i)})_{i \geq 0}$ is non-decreasing. Combining this with the fact that it is upper bounded by $V^\star$ from Lem. 17, the sequence must converge.

*Proof.* We recognize that $V^{(i+1)}(s) \leftarrow \min_a Q^{(i+1)}(s,a)$, with

$$Q^{(i+1)}(s,a) \leftarrow \max\left\{\widehat{c}(s,a) + \underbrace{f\left(\widetilde{P}_{s,a}, V^{(i)}, n^+(s,a), B, \iota_{s,a}\right)}_{:=g_{s,a}(V^{(i)})} - c_3\sqrt{\frac{\widehat{c}(s,a)\iota_{s,a}}{n^+(s,a)}} - c_4\frac{B\sqrt{S'\iota_{s,a}}}{n^+(s,a)}, \ 0\right\},$$

where we introduce the function $g_{s,a}(V) := f\left(\widetilde{P}_{s,a}, V, n^+(s,a), B, \iota_{s,a}\right)$ for notational ease as all other parameters (apart from $V$) will remain the same throughout the analysis.

We prove by induction on the iterations indexed by $i$ that $Q^{(i)} \leq Q^{(i+1)}$. First, $Q^{(0)} = 0 \leq Q^{(1)}$. Now assume that $Q^{(i-1)} \leq Q^{(i)}$. Then

$$Q^{(i+1)}(s,a) = \max\left\{\widehat{c}(s,a) + g_{s,a}(V^{(i)}) - c_3\sqrt{\frac{\widehat{c}(s,a)\iota_{s,a}}{n^+(s,a)}} - c_4\frac{B\sqrt{S'\iota_{s,a}}}{n^+(s,a)}, \ 0\right\}$$

$$\geq \max\left\{\widehat{c}(s,a) + g_{s,a}(V^{(i-1)}) - c_3\sqrt{\frac{\widehat{c}(s,a)\iota_{s,a}}{n^+(s,a)}} - c_4\frac{B\sqrt{S'\iota_{s,a}}}{n^+(s,a)}, \ 0\right\}$$

$$= Q^{(i)}(s,a),$$

where the inequality uses the induction hypothesis $V^{(i)} \geq V^{(i-1)}$ and the fact that $g_{s,a}$ is non-decreasing from the first claim of Lem. 16. $\square$

**Restatement of Lemma 19.** Denote by $\nu > 0$ the probability of reaching the goal from any state-action pair in $\widetilde{P}$, i.e., $\nu := \min_{s,a} \widetilde{P}_{s,a,g}$. Then $\widetilde{\mathcal{L}}$ is a $\rho$-contractive operator with modulus $\rho := 1 - \nu < 1$.

*Proof.* Take any two vectors $U_1, U_2$, then for any state $s \in \mathcal{S}$,

$$|\widetilde{\mathcal{L}}U_1(s) - \widetilde{\mathcal{L}}U_2(s)| = \left|\min_a \widetilde{\mathcal{L}}U_1(s,a) - \min_a \widetilde{\mathcal{L}}U_2(s,a)\right|$$

$$\leq \left|\max_a\left\{\widetilde{\mathcal{L}}U_1(s,a) - \widetilde{\mathcal{L}}U_2(s,a)\right\}\right|,$$

and we have that for any action $a \in \mathcal{A}$,

$$|\widetilde{\mathcal{L}}U_1(s,a) - \widetilde{\mathcal{L}}U_2(s,a)| \leq \left|\max\left\{\widehat{c}(s,a) + g_{s,a}(U_1), \ 0\right\} - \max\left\{\widehat{c}(s,a) + g_{s,a}(U_2), \ 0\right\}\right|$$

$$\leq \left|g_{s,a}(U_1) - g_{s,a}(U_2)\right|$$

$$\overset{\text{(i)}}{\leq} \rho_{s,a}\|U_1 - U_2\|_\infty.$$

The third claim of Lem. 16 is employed to justify inequality (i): $g_{s,a}$ is $\rho_{s,a}$-contractive (where $g_{s,a}$ is defined in the proof of Lem. 18) with (recall Eq. 12)

$$\rho_{s,a} := 1 - \widetilde{P}_{s,a,g} = 1 - \nu_{s,a}.$$

Taking the maximum over $(s,a)$ pairs, $\widetilde{\mathcal{L}}$ is thus $\rho$-contractive with modulus $\rho := 1 - \nu < 1$. $\square$

**Restatement of Lemma 20.** Conditioned on the event $\mathcal{E}$, for any interval $m$ and state-action pair $(s,a) \in \mathcal{S} \times \mathcal{A}$,

$$|c(s,a) + P_{s,a}V^m - Q^m(s,a)| \leq \min\left\{\beta^m(s,a), B_\star + 1\right\},$$

where we define

$$\beta^m(s,a) := 4b^m(s,a) + \sqrt{\frac{2\mathbb{V}(P_{s,a}, V^\star)\iota_{s,a}}{n^m(s,a)}} + \sqrt{\frac{2S'\mathbb{V}(P_{s,a}, V^\star - V^m)\iota_{s,a}}{n^m(s,a)}}$$

$$+ \frac{3B_\star S'\iota_{s,a}}{n^m(s,a)} + \left(1 + c_1\sqrt{\iota_{s,a}/2}\right)\epsilon_{\text{VI}}^m.$$

*Proof.* First we see that $c(s,a) + P_{s,a}V^m - Q^m(s,a) \leq c(s,a) + P_{s,a}V^\star = Q^\star(s,a) \leq B_\star + 1$ and that $Q^m(s,a) - c(s,a) - P_{s,a}V^m \leq Q^\star(s,a) \leq B_\star + 1$, from Lem. 17 and the Bellman optimality equation (Lem. 2). Now we prove that $|c(s,a) + P_{s,a}V^m - Q^m(s,a)| \leq \beta^m(s,a)$.

**Bounding** $c(s,a) + P_{s,a}V^m - Q^m(s,a)$**.** From the VISGO loop of Alg. 1, the vectors $Q^m$ and $V^m$ can be associated to a finite iteration $l$ of a sequence of vectors $(Q^{(i)})_{i \geq 0}$ and $(V^{(i)})_{i \geq 0}$ such that

(i) $Q^m(s,a) := Q^{(l)}(s,a)$,

(ii) $V^m(s) := V^{(l)}(s)$,

(iii) $\|V^{(l)} - V^{(l-1)}\|_\infty \leq \epsilon_{\text{VI}}^m$,

(iv) $b^m(s,a) := b^{(l+1)}(s,a) = \max\left\{c_1\sqrt{\frac{\mathbb{V}(\widetilde{P}_{s,a}, V^{(l)})\iota_{s,a}}{n^m(s,a)}},\ c_2\frac{B\iota_{s,a}}{n^m(s,a)}\right\} + c_3\sqrt{\frac{\widehat{c}^m(s,a)\iota_{s,a}}{n^m(s,a)}} + c_4\frac{B\sqrt{S'\iota_{s,a}}}{n^m(s,a)}$.

First, we examine the gap between the exploration bonuses at the final VISGO iterations $l$ and $l+1$ as follows

$$b^{(l)}(s,a) \overset{(i)}{\leq} c_1\sqrt{\frac{\mathbb{V}(\widetilde{P}_{s,a}, V^{(l-1)})\iota_{s,a}}{n^+(s,a)}} + c_2\frac{B\iota_{s,a}}{n^+(s,a)} + c_3\sqrt{\frac{\widehat{c}(s,a)\iota_{s,a}}{n^+(s,a)}} + c_4\frac{B\sqrt{S'\iota_{s,a}}}{n^+(s,a)}$$

$$\overset{(ii)}{\leq} c_1\sqrt{2\frac{\mathbb{V}(\widetilde{P}_{s,a}, V^{(l)})\iota_{s,a}}{n^+(s,a)}} + c_1\sqrt{2\frac{\mathbb{V}(\widetilde{P}_{s,a}, V^{(l-1)} - V^{(l)})\iota_{s,a}}{n^+(s,a)}} + c_2\frac{B\iota_{s,a}}{n^+(s,a)}$$

$$+ c_3\sqrt{\frac{\widehat{c}(s,a)\iota_{s,a}}{n^+(s,a)}} + c_4\frac{B\sqrt{S'\iota_{s,a}}}{n^+(s,a)}$$

$$\overset{(iii)}{\leq} 2\sqrt{2}b^{(l+1)}(s,a) + c_1\sqrt{\frac{(\epsilon_{\text{VI}}^m)^2\iota_{s,a}}{2n^+(s,a)}}$$

$$\leq 2\sqrt{2}b^{(l+1)}(s,a) + \epsilon_{\text{VI}}^m c_1\sqrt{\iota_{s,a}/2},$$

where (i) uses $\max\{x,\ y\} \leq x + y$; (ii) uses $\mathbb{V}(P, X+Y) \leq 2(\mathbb{V}(P,X) + \mathbb{V}(P,Y))$ and $\sqrt{x+y} \leq \sqrt{x} + \sqrt{y}$; (iii) uses $x + y \leq 2\max\{x,\ y\}$ and Popoviciu's inequality (Lem. 28) applied to $V^{(l-1)} - V^{(l)} \in [-\epsilon_{\text{VI}}^m, 0]$. Moreover, we have that $Q^{(l)}(s,a) \geq \widehat{c}(s,a) + \widetilde{P}_{s,a}V^{(l-1)} - b^{(l)}(s,a)$ from Eq. 3. Combining everything yields

$$-Q^m(s,a) \leq -\widehat{c}(s,a) - \widetilde{P}_{s,a}(V^m - \epsilon_{\text{VI}}) + \epsilon_{\text{VI}}c_1\sqrt{\iota_{s,a}/2} + 2\sqrt{2}b^m(s,a)$$

$$\leq -\widehat{c}(s,a) - \widetilde{P}_{s,a}V^m + 2\sqrt{2}b^m(s,a) + \left(1 + c_1\sqrt{\iota_{s,a}/2}\right)\epsilon_{\text{VI}}^m.$$

Therefore, we have

$$c(s,a) + P_{s,a}V^m - Q^m(s,a)$$

$$\leq c(s,a) + P_{s,a}V^m - \widehat{c}^m(s,a) - \widetilde{P}_{s,a}V^m + 2\sqrt{2}b^m(s,a) + \left(1 + c_1\sqrt{\iota_{s,a}/2}\right)\epsilon_{\text{VI}}^m$$

$$\overset{(i)}{\leq} P_{s,a}V^m - \widehat{P}_{s,a}V^m + \frac{B_\star}{n^m(s,a) + 1} + 4b^m(s,a) + \left(1 + c_1\sqrt{\iota_{s,a}/2}\right)\epsilon_{\text{VI}}^m$$

$$\leq \underbrace{(P_{s,a} - \widehat{P}_{s,a})V^\star}_{:=Y_1} + \underbrace{(P_{s,a} - \widehat{P}_{s,a})(V^m - V^\star)}_{:=Y_2} + \frac{B_\star}{n^m(s,a)} + 4b^m(s,a) + \left(1 + c_1\sqrt{\iota_{s,a}/2}\right)\epsilon_{\mathrm{VI}}^m,$$

where (i) comes from Lem. 14, the event $\mathcal{E}_2$, Lem. 17 and (loosely) bounding $|c(s,a) - \widehat{c}(s,a)| \leq b^m(s,a)$. It holds under the event $\mathcal{E}_1$ that

$$|Y_1| \leq \sqrt{\frac{2\mathbb{V}(P_{s,a}, V^\star)\iota_{s,a}}{n^m(s,a)}} + \frac{B_\star \iota_{s,a}}{n^m(s,a)}.$$

Moreover, we have

$$
\begin{aligned}
|Y_2| &\overset{\text{(i)}}{=} |\sum_{s'}(\widehat{P}_{s,a,s'} - P_{s,a,s'})(V^m(s') - V^\star(s') - P_{s,a}(V^m - V^\star))| \\
&\leq \sum_{s'}|P_{s,a,s'} - \widehat{P}_{s,a,s'}||V^m(s') - V^\star(s') - P_{s,a}(V^m - V^\star)| \\
&\overset{\text{(ii)}}{\leq} \sum_{s'}\sqrt{\frac{2P_{s,a,s'}\iota_{s,a}}{n^m(s,a)}}|V^m(s') - V^\star(s') - P_{s,a}(V^m - V^\star)| + \frac{B_\star S'\iota_{s,a}}{n^m(s,a)} \\
&\overset{\text{(iii)}}{\leq} \sqrt{\frac{2S'\mathbb{V}(P_{s,a}, V^m - V^\star)\iota_{s,a}}{n^m(s,a)}} + \frac{B_\star S'\iota_{s,a}}{n^m(s,a)},
\end{aligned}
$$

where the shift performed in (i) is by $\sum_{s'} P_{s,a,s'} = \sum_{s'} \widehat{P}_{s,a,s'} = 1$; (ii) holds under the event $\mathcal{E}_3$ and Lem. 17 ($V^m(s) \in [0, B_\star]$); (iii) is by Cauchy-Schwarz inequality.

**Bounding $Q^m(s,a) - c(s,a) - P_{s,a}V^m$.** If $Q^m(s,a) = Q^{(l)}(s,a) = 0$, then $Q^m(s,a) - \widehat{c}(s,a) - P_{s,a}V^m \leq 0 \leq \min\{\beta^m(s,a), B_\star\}$. Otherwise, we have $Q^m(s,a) = Q^{(l)}(s,a) = \widehat{c}(s,a) + \widetilde{P}_{s,a}V^{(l-1)} - b^{(l)}(s,a)$. Using that $V^m \geq V^{(l-1)}$ (Lem. 18) and $\widehat{P}_{s,a}V^m \geq \widetilde{P}_{s,a}V^m$ (Lem. 14), we get

$$
\begin{aligned}
Q^m(s,a) - c(s,a) - P_{s,a}V^m &\leq Q^m(s,a) - \widehat{c}(s,a) - P_{s,a}V^m + b^m(s,a) \\
&= \widetilde{P}_{s,a}V^{(l-1)} - b^{(l)}(s,a) - P_{s,a}V^m + b^m(s,a) \\
&\leq \widehat{P}_{s,a}V^m - P_{s,a}V^m + b^m(s,a) \\
&= (\widehat{P}_{s,a} - P_{s,a})V^\star - (\widehat{P}_{s,a} - P_{s,a})(V^\star - V^m) + b^m(s,a) \\
&\leq |Y_1| + |Y_2| + b^m(s,a),
\end{aligned}
$$

which can be bounded as above. $\qquad\square$

### E.2 Additional lemmas

**Lemma 21.** *Let $\widetilde{Q}^m(s,a) := Q^\star(s,a) - Q^m(s,a)$ and $\widetilde{V}^m(s) := V^\star(s) - V^m(s)$. Then conditioned on the event $\mathcal{E}$, we have that for all $(s,a,m,h)$,*

$$\widetilde{V}(s_h^m) - P_{s_h^m, a_h^m}\widetilde{V}(s_{h+1}^m) \leq \beta^m(s_h^m, a_h^m).$$

*Proof.* We write that

$$
\begin{aligned}
\widetilde{V}^m(s_h^m) - P_{s_h^m, a_h^m}\widetilde{V}^m(s_{h+1}^m) &= V^\star(s_h^m) - P_{s_h^m, a_h^m}V^\star + P_{s_h^m, a_h^m}V^m - V^m(s_h^m) \\
&\leq Q^\star(s_h^m, a_h^m) - P_{s_h^m, a_h^m}V^\star + P_{s_h^m, a_h^m}V^m - V^m(s_h^m) \\
&\overset{\text{(i)}}{=} c(s_h^m, a_h^m) + P_{s_h^m, a_h^m}V^m - Q^m(s_h^m, a_h^m) \\
&\overset{\text{(ii)}}{\leq} \beta^m(s_h^m, a_h^m),
\end{aligned}
$$

where (i) uses the Bellman optimality equation (Lem. 2) and the fact that $V^m(s_h^m) = Q^m(s_h^m, a_h^m)$, and (ii) comes from Lem. 20. $\qquad\square$

**Lemma 22.** *For any $M' \leq M$, it holds that*

$$\sum_{m=1}^{M'} \left( \sum_{h=1}^{H^m} V^m(s_h^m) - V^m(s_{h+1}^m) \right) - \sum_{m \in \mathcal{M}_0(M')} V^m(s_0) \leq 2SA \log_2(T_{M'}) \max_{1 \leq m \leq M'} \|V^m\|_\infty.$$

*Proof.* We recall that we denote by $\mathcal{M}_0(M')$ the set of intervals among the first $M'$ intervals that constitute the first intervals in each episode. From the analytical construction of intervals, an interval $m < M'$ can end due to one of the following three conditions:

(i) If interval $m$ ends in the goal state, then
$$V^{m+1}(s_1^{m+1}) - V^m(s_{H^m+1}^m) = V^{m+1}(s_0) - V^m(g) = V^{m+1}(s_0).$$
This happens for all the intervals $m + 1 \in \mathcal{M}_0(M')$.

(ii) If interval $m$ ends when the count to a state-action pair is doubled, then we replan with a VISGO procedure. Thus we get
$$V^{m+1}(s_1^{m+1}) - V^m(s_{H^m+1}^m) \leq V^{m+1}(s_1^{m+1}) \leq \max_{1 \leq m \leq M'} \|V^m\|_\infty.$$
This happens at most $2SA \log_2(T_{M'})$ times.

Combining the three conditions above implies that

$$\sum_{m=1}^{M'} \left( \sum_{h=1}^{H^m} V^m(s_h^m) - V^m(s_{h+1}^m) \right)$$

$$= \sum_{m=1}^{M'} V^m(s_1^m) - V^m(s_{H^m+1}^m)$$

$$= \sum_{m=1}^{M'-1} \left( V^{m+1}(s_1^{m+1}) - V^m(s_{H^m+1}^m) \right) + \underbrace{\sum_{m=1}^{M'-1} \left( V^m(s_1^m) - V^{m+1}(s_1^{m+1}) \right)}_{=V^1(s_1^1)-V^{M'}(s_1^{M'})} + \underbrace{V^{M'}(s_1^{M'}) - V^{M'}(s_{H^{M'}+1}^{M'})}_{\leq 0}$$

$$\leq \sum_{m=1}^{M'-1} \left( V^{m+1}(s_1^{m+1}) - V^m(s_{H^m+1}^m) \right) + V^1(s_0)$$

$$\leq \sum_{m=1}^{M'-1} V^{m+1}(s_0) \mathbb{I}[m+1 \in \mathcal{M}_0(M')] + 2SA \log_2(T_{M'}) \max_{1 \leq m \leq M'} \|V^m\|_\infty + V^1(s_0)$$

$$= \sum_{m \in \mathcal{M}_0(M')} V^m(s_0) + 2SA \log_2(T_{M'}) \max_{1 \leq m \leq M'} \|V^m\|_\infty.$$

$\square$

### E.3 Full proof of the bound on $X_2(M')$

① **First, bound $\beta^m$.**

Recall that we assume that the event $\mathcal{E}$ holds. From Lem. 20, we have for any $m, s, a$,

$$\beta^m(s,a) = O\left( \sqrt{\frac{\mathbb{V}(\widetilde{P}_{s,a}, V^m)\iota_{s,a}}{n^m(s,a)}} + \sqrt{\frac{\mathbb{V}(P_{s,a}, V^\star)\iota_{s,a}}{n^m(s,a)}} + \sqrt{\frac{S\mathbb{V}(P_{s,a}, V^\star - V^m)\iota_{s,a}}{n^m(s,a)}} \right.$$

$$\left. + \sqrt{\frac{\widehat{c}^m(s,a)\iota_{s,a}}{n^m(s,a)}} + \frac{B_\star S\iota_{s,a}}{n^m(s,a)} + \frac{B\sqrt{S}\iota_{s,a}}{n^m(s,a)} + \left( 1 + c_1\sqrt{\iota_{s,a}/2} \right)\epsilon_{\text{VI}}^m \right).$$

Here we interchange $S'$ and $S$ since we use the $O()$ notation. From Lem. 14 and Lem. 17, for any $m, s, a$,

$$\mathbb{V}(\widetilde{P}_{s,a}, V^m) \leq \mathbb{V}(\widehat{P}_{s,a}, V^m) + \frac{2B_\star^2 S'}{n^m(s,a)+1} < \mathbb{V}(\widehat{P}_{s,a}, V^m) + \frac{2B_\star^2 S'}{n^m(s,a)}.$$

Under the event $\mathcal{E}_3$, it holds that

$$\widehat{P}_{s,a,s'} \leq P_{s,a,s'} + \sqrt{\frac{2P_{s,a,s'}\iota_{s,a}}{n^m(s,a)}} + \frac{\iota_{s,a}}{n^m(s,a)} \leq \frac{3}{2}P_{s,a,s'} + \frac{2\iota_{s,a}}{n^m(s,a)}.$$

Thus, it holds that for any $m, s, a$,

$$
\begin{aligned}
\mathbb{V}(\widehat{P}_{s,a}, V^m) &= \sum_{s'} \widehat{P}_{s,a,s'}\left(V^m(s') - \widehat{P}_{s,a}V^m\right)^2 \\
&\overset{(i)}{\leq} \sum_{s'} \widehat{P}_{s,a,s'}(V^m(s') - P_{s,a}V^m)^2 \\
&\leq \sum_{s'} \left(\frac{3}{2}P_{s,a,s'} + \frac{2\iota_{s,a}}{n^m(s,a)}\right)(V^m(s') - P_{s,a}V^m)^2 \\
&\leq \frac{3}{2}\mathbb{V}(P_{s,a}, V^m) + \frac{2B_\star^2 S'\iota_{s,a}}{n^m(s,a)}.
\end{aligned}
$$

(i) is by the fact that $z^\star = \sum_i p_i x_i$ minimizes the quantity $\sum_i p_i(x_i - z)^2$. As a result,

$$\mathbb{V}(\widetilde{P}_{s,a}, V^m) < \frac{3}{2}\mathbb{V}(P_{s,a}, V^m) + \frac{2B_\star^2 S'}{n^m(s,a)} + \frac{2B_\star^2 S'\iota_{s,a}}{n^m(s,a)}.$$

Utilizing $\mathbb{V}(P, X+Y) \leq 2(\mathbb{V}(P,X)+\mathbb{V}(P,Y))$ with $X = V^\star - V^m$ and $Y = V^m$ and $\sqrt{x+y} \leq \sqrt{x} + \sqrt{y}$, finally we have

$$
\begin{aligned}
\beta^m(s,a) \leq O\Bigg(&\sqrt{\frac{\mathbb{V}(P_{s,a}, V^m)\iota_{s,a}}{n^m(s,a)}} + \sqrt{\frac{S\mathbb{V}(P_{s,a}, V^\star - V^m)\iota_{s,a}}{n^m(s,a)}} \\
&+ \sqrt{\frac{\widehat{c}(s,a)\iota_{s,a}}{n^m(s,a)}} + \frac{B_\star S\iota_{s,a}}{n^m(s,a)} + \frac{B\sqrt{S}\iota_{s,a}}{n^m(s,a)} + \left(1 + c_1\sqrt{\iota_{s,a}/2}\right)\epsilon_{\mathrm{VI}}^m\Bigg).
\end{aligned}
$$

**② Second, bound a special type of summation.**

**Lemma 23.** *Let $w = \{w_h^m \geq 0 \ : \ 1 \leq m \leq M, \ 1 \leq h \leq H^m\}$ be a group of weights, then for any $M' \leq M$,*

$$\sum_{m=1}^{M'} \sum_{h=1}^{H^m} \sqrt{\frac{w_h^m}{n^m(s_h^m, a_h^m)}} \leq O\left(\sqrt{SA\log_2(T_{M'}) \sum_{m=1}^{M'} \sum_{h=1}^{H^m} w_h^m}\right).$$

*Proof.* For $m \leq M'$, $n^m(s,a) \in \{2^i \ : \ i \in \mathbb{N}, i \leq \log_2(T_{M'})\}$. We can count the occurrences of a fixed value of $n^m(s,a)$ by the doubling property of $\mathtt{VISGO}$: $\forall i, s, a$

$$\sum_{m=1}^{M'} \sum_{h=1}^{H^m} \mathbb{I}[(s_h^m, a_h^m) = (s,a), n^m(s,a) = 2^i] \leq 2^i.$$

Thus

$$
\begin{aligned}
\sum_{m=1}^{M'} \sum_{h=1}^{H^m} \frac{1}{n^m(s_h^m, a_h^m)} &= \sum_{s,a} \sum_{0 \leq i \leq \log_2(T_{M'})} \sum_{m=1}^{M'} \sum_{h=1}^{H^m} \mathbb{I}[(s_h^m, a_h^m) = (s,a), n^m(s,a) = 2^i]\frac{1}{2^i} \\
&= \sum_{s,a} \sum_{0 \leq i \leq \log_2(T_{M'})} 1 \\
&\leq SA(\log_2(T_{M'}) + 1) \\
&\leq O(SA\log_2(T_{M'})).
\end{aligned}
\tag{14}
$$

By Cauchy-Schwarz inequality,

$$\sum_{m=1}^{M'}\sum_{h=1}^{H^m}\sqrt{\frac{w_h^m}{n^m(s_h^m,a_h^m)}} \le \sqrt{\left(\sum_{m=1}^{M'}\sum_{h=1}^{H^m}w_h^m\right)\left(\sum_{m=1}^{M'}\sum_{h=1}^{H^m}\frac{1}{n^m(s_h^m,a_h^m)}\right)}$$

$$\le O\left(\sqrt{SA\log_2(T_{M'})\sum_{m=1}^{M'}\sum_{h=1}^{H^m}w_h^m}\right).$$

$\square$

By setting successively $w_h^m = \mathbb{V}(P_{s_h^m,a_h^m},V^m)$, $\mathbb{V}(P_{s_h^m,a_h^m},V^\star-V^m)$ and $\widehat{c}(s_h^m,a_h^m)$, and relaxing $\iota_{s_h^m,a_h^m}$ to its upper-bound $\iota_{M'} = \ln\left(\frac{12SAS'T_{M'}^2}{\delta}\right)$ we have

$$X_2(M') \le O\left(\sqrt{SA\log_2(T_{M'})\iota_{M'}\underbrace{\sum_{m=1}^{M'}\sum_{h=1}^{H^m}\mathbb{V}(P_{s_h^m,a_h^m},V^m)}_{:=X_4(M')}}\right.$$

$$+\sqrt{S^2A\log_2(T_{M'})\iota_{M'}\underbrace{\sum_{m=1}^{M'}\sum_{h=1}^{H^m}\mathbb{V}(P_{s_h^m,a_h^m},V^\star-V^m)}_{:=X_5(M')}}$$

$$+\sqrt{SA\log_2(T_{M'})\iota_{M'}\sum_{m=1}^{M'}\sum_{h=1}^{H^m}\widehat{c}(s_h^m,a_h^m)+B_\star S^2A\log_2(T_{M'})}$$

$$\left.+BS^{3/2}A\log_2(T_{M'})\iota_{M'}+\sum_{m=1}^{M'}\sum_{h=1}^{H^m}(1+c_1\sqrt{\iota_{M'}/2})\epsilon_{\mathrm{VI}}^m\right).$$

③ **Third, bound each summation separately.**

***Regret contribution of the estimated costs.*** From line 15 in EB-SSP, we have that $\widehat{c}(s,a) \le \frac{2\theta(s,a)}{N(s,a)}$. Let $\theta^m(s,a)$ denote the value of $\theta(s,a)$ for calculating $\widehat{c}^m$. By definition,

$$\theta^m(s_h^m,a_h^m) = \sum_{m'=1}^{M'}\sum_{h'=1}^{H^{m'}}\mathbb{I}[(s_h^m,a_h^m)=(s_{h'}^{m'},a_{h'}^{m'}),\ n^m(s_h^m,a_h^m)=2n^{m'}(s_{h'}^{m'},a_{h'}^{m'})]c_{h'}^{m'}$$

$$-\mathbb{I}\big[\text{first occurrence of }(m',h')\text{ such that }(s_h^m,a_h^m)=(s_{h'}^{m'},a_{h'}^{m'}),\ n^m(s_h^m,a_h^m)=2n^{m'}(s_{h'}^{m'},a_{h'}^{m'})\big]c_{h'}^{m'}$$

$$+\mathbb{I}\big[\text{first occurrence of }(m',h')\text{ such that }(s_h^m,a_h^m)=(s_{h'}^{m'},a_{h'}^{m'}),\ n^m(s_h^m,a_h^m)=n^{m'}(s_{h'}^{m'},a_{h'}^{m'})\big]c_{h'}^{m'}$$

$$\le \sum_{m'=1}^{M'}\sum_{h'=1}^{H^{m'}}\mathbb{I}[(s_h^m,a_h^m)=(s_{h'}^{m'},a_{h'}^{m'}),\ n^m(s_h^m,a_h^m)=2n^{m'}(s_{h'}^{m'},a_{h'}^{m'})]c_{h'}^{m'}+1.$$

For any $M' \le M$ we have

$$\sum_{m=1}^{M'}\sum_{h=1}^{H^m}\widehat{c}^m(s_h^m,a_h^m)$$

$$\le \sum_{m=1}^{M'}\sum_{h=1}^{H^m}\frac{2\theta^m(s_h^m,a_h^m)}{n^m(s_h^m,a_h^m)}$$

$$= \sum_{m=1}^{M'}\sum_{h=1}^{H^m}\sum_{m'=1}^{M'}\sum_{h'=1}^{H^{m'}}\mathbb{I}[(s_h^m,a_h^m)=(s_{h'}^{m'},a_{h'}^{m'}),\ n^m(s_h^m,a_h^m)=2n^{m'}(s_{h'}^{m'},a_{h'}^{m'})]\frac{2c_{h'}^{m'}}{n^m(s_h^m,a_h^m)}$$

$$+ \sum_{m=1}^{M'} \sum_{h=1}^{H^m} \frac{2}{n^m(s_h^m, a_h^m)}$$

$$\overset{(i)}{\leq} \sum_{m'=1}^{M'} \sum_{h'=1}^{H^{m'}} \frac{c_{h'}^{m'}}{n^{m'}(s_{h'}^{m'}, a_{h'}^{m'})} \cdot \sum_{m=1}^{M'} \sum_{h=1}^{H^m} \mathbb{I}[(s_h^m, a_h^m) = (s_{h'}^{m'}, a_{h'}^{m'}), \ n^m(s_h^m, a_h^m) = 2n^{m'}(s_{h'}^{m'}, a_{h'}^{m'})]$$
$$+ 2SA(\log_2(T_{M'}) + 1)$$

$$\leq 2SA(\log_2(T_{M'}) + 1) + \sum_{m'=1}^{M'} \sum_{h'=1}^{H^{m'}} \frac{c_{h'}^{m'}}{n^{m'}(s_{h'}^{m'}, a_{h'}^{m'})} \cdot 2n^{m'}(s_{h'}^{m'}, a_{h'}^{m'})$$

$$= 2SA(\log_2(T_{M'}) + 1) + 2 \sum_{m'=1}^{M'} \sum_{h'=1}^{H^{m'}} c_{h'}^{m'}$$

$$= 2SA(\log_2(T_{M'}) + 1) + 2C_{M'},$$

where (i) comes from Eq. 14.

***Regret contribution of the* VISGO *precision errors.*** For any $M' \leq M$, denote by $J_{M'}$ the (unknown) total number of triggers in the first $M'$ intervals. For $1 \leq j \leq J_{M'}$, denote by $L_j$ the number of time steps elapsed between the $(j-1)$-th and the $j$-th trigger. The doubling condition implies that $L_j \leq 2^j SA$ and that there are at most $J_{M'} = O(SA \log_2(T_{M'}/(SA)))$ triggers. Using that Alg. 1 selects as error $\epsilon_{\text{VI}}^j = 2^{-j}/(SA)$, we have that

$$\sum_{m=1}^{M'} \sum_{h=1}^{H^m} (1 + c_1 \sqrt{\iota_{M'}/2}) \epsilon_{\text{VI}}^m \leq (1 + c_1 \sqrt{\iota_{M'}/2}) \sum_{j=1}^{J_{M'}} L_j \epsilon_{\text{VI}}^j$$
$$\leq (1 + c_1 \sqrt{\iota_{M'}/2}) J_{M'}$$
$$= O\left(SA \log_2(T_{M'}) \sqrt{\iota_{M'}}\right).$$

**Lemma 24.** *Conditioned on Lem. 20, for a fixed $M' \leq M$ with probability $1 - 2\delta$,*

$$X_4(M') \leq O\left(B_\star(C_{M'} + X_2(M')) + (B_\star^2 SA + B_\star)(\log_2(T_{M'}) + \ln(2/\delta))\right).$$

*Proof.* We introduce the normalized value function $\overline{V}^m := V^m/B_\star \in [0, 1]$. Define

$$F(d) := \sum_{m=1}^{M'} \sum_{h=1}^{H^m} (P_{s_h^m, a_h^m}(\overline{V}^m)^{2^d} - (\overline{V}^m(s_{h+1}^m))^{2^d}), \ G(d) := \sum_{m=1}^{M'} \sum_{h=1}^{H^m} \mathbb{V}(P_{s_h^m, a_h^m}, (\overline{V}^m)^{2^d}).$$

Then $X_4(M') = B_\star^2 G(0)$. Direct computation gives that

$$G(d) = \sum_{m=1}^{M'} \sum_{h=1}^{H^m} \left( P_{s_h^m, a_h^m}(\overline{V}^m)^{2^{d+1}} - (P_{s_h^m, a_h^m}(\overline{V}^m)^{2^d})^2 \right)$$

$$\overset{(i)}{\leq} \sum_{m=1}^{M'} \sum_{h=1}^{H^m} \left( P_{s_h^m, a_h^m}(\overline{V}^m)^{2^{d+1}} - (\overline{V}^m(s_{h+1}^m))^{2^{d+1}} \right) + \underbrace{\sum_{m=1}^{M'} (\overline{V}^m(s_{H^m+1}^m))^{2^{d+1}}}_{\leq M_1'}$$

$$+ \sum_{m=1}^{M'} \sum_{h=1}^{H^m} \left( (\overline{V}^m(s_h^m))^{2^{d+1}} - (P_{s_h^m, a_h^m}\overline{V}^m)^{2^{d+1}} \right) - \underbrace{\sum_{m=1}^{M'} (\overline{V}^m(s_1^m))^{2^{d+1}}}_{\leq 0}$$

$$\overset{(ii)}{\leq} F(d+1) + M_1' + 2^{d+1} \sum_{m=1}^{M'} \sum_{h=1}^{H^m} \max\{\overline{V}^m(s_h^m) - P_{s_h^m, a_h^m}\overline{V}^m, \ 0\}$$

$$= F(d+1) + M_1' + \frac{2^{d+1}}{B_\star} \sum_{m=1}^{M'} \sum_{h=1}^{H^m} \max\{Q^m(s_h^m, a_h^m) - P_{s_h^m, a_h^m} V^m, 0\}$$

$$\overset{\text{(iii)}}{\leq} F(d+1) + M_1' + \frac{2^{d+1}}{B_\star} \sum_{m=1}^{M'} \sum_{h=1}^{H^m} (c(s_h^m, a_h^m) + \beta^m(s_h^m, a_h^m))$$

$$= F(d+1) + M_1' + \frac{2^{d+1}}{B_\star} \sum_{m=1}^{M'} \sum_{h=1}^{H^m} (c_h^m + \beta^m(s_h^m, a_h^m) + (c(s_h^m, a_h^m) - c_h^m))$$

$$\leq F(d+1) + M_1' + \frac{2^{d+1}}{B_\star} (C_{M'} + X_2(M') + |X_3(M')|),$$

where $M_1'$ denotes the number of intervals satisfying $\overline{V}^m(s_{H^m+1}^m) \neq 0$; (i) is by convexity of $f(x) = x^{2^d}$; (ii) is by Lem. 34; (iii) is by Lem. 20.

For a fixed $d$, $F(d)$ is a martingale. By taking $c = 1$ in Lem. 30, we have

$$\mathbb{P}\Big[F(d) > 2\sqrt{2G(d)(\log_2(T_{M'}) + \ln(2/\delta))} + 5(\log_2(T_{M'}) + \ln(2/\delta))\Big] \leq \delta.$$

Taking $\delta' = \delta/(\log_2(T_{M'}) + 1)$, using $x \geq \ln(x) + 1$ and finally swapping $\delta$ and $\delta'$, we have that

$$\mathbb{P}\Big[F(d) > 2\sqrt{2G(d)(2\log_2(T_{M'}) + \ln(2/\delta))} + 5(2\log_2(T_{M'}) + \ln(2/\delta))\Big] \leq \frac{\delta}{\log_2(T_{M'}) + 1}.$$

Taking a union bound over $d = 1, 2, \ldots, \log_2(T_{M'})$, we have that with probability $1 - \delta$,

$$F(d) \overset{\text{(i)}}{\leq} 2\sqrt{2(2\log_2(T_{M'}) + \ln(2/\delta))} \cdot \sqrt{F(d+1) + 2^{d+1} \cdot \frac{C_{M'} + X_2(M') + |X_3(M')|}{B_\star}}$$

$$+ 5(2\log_2(T_M) + \ln(2/\delta)) + 2\sqrt{2(2\log_2(T_{M'}) + \ln(2/\delta))M_1'}.$$

From Lem. 32, taking $\lambda_1 = T_{M'}$, $\lambda_2 = 2\sqrt{2(2\log_2(T_{M'}) + \ln(2/\delta))}$, $\lambda_3 = (C_{M'} + X_2(M') + |X_3(M')|)/B_\star$, $\lambda_4 = 5(2\log_2(T_M) + \ln(2/\delta)) + 2\sqrt{2(2\log_2(T_{M'}) + \ln(2/\delta))M_1'}$, we have that

$$F(1) \leq O\Big(\log_2(T_{M'}) + \ln(2/\delta) + \frac{C_{M'} + X_2(M') + |X_3(M')|}{B_\star} + M_1'\Big).$$

Hence

$$X_4(M') \leq O\big(B_\star(C_{M'} + X_2(M') + |X_3(M')|) + B_\star^2(\log_2(T_{M'}) + \ln(2/\delta) + M_1')\big).$$

By definition, $M_1' \leq O(SA \log_2(T_{M'}))$ since only those intervals ending by triggering the doubling condition are taken into account. From the bound of $|X_3(M')|$, the following holds with probability $1 - 2\delta$:

$$X_4(M') \leq O\big(B_\star(C_{M'} + X_2(M')) + (B_\star^2 SA + B_\star)(\log_2(T_{M'}) + \ln(2/\delta))\big).$$

Throughout the proof, the inequality $O(\sqrt{xy}) \leq O(x + y)$ is utilized to simplify the bound. $\qquad\square$

**Lemma 25.** *Conditioned on Lem. 20, for a fixed $M' \leq M$ with probability $1 - \delta$,*
$$X_5(M') \leq O\big(B_\star^2 SA(\log_2(T_{M'}) + \ln(2/\delta)) + B_\star X_2(M')\big).$$

*Proof.* We introduce the normalized quantity $\overline{\widetilde{V}}^m := \widetilde{V}^m/B_\star \in [-1, 1]$ (recall the definition in Lem. 21). Define

$$\widetilde{F}(d) := \sum_{m=1}^{M'} \sum_{h=1}^{H^m} (P_{s_h^m, a_h^m} (\overline{\widetilde{V}}^m)^{2^d} - (\overline{\widetilde{V}}^m(s_{h+1}^m))^{2^d}), \quad \widetilde{G}(d) := \sum_{m=1}^{M'} \sum_{h=1}^{H^m} \mathbb{V}(P_{s_h^m, a_h^m}, (\overline{\widetilde{V}}^m)^{2^d}).$$

Then $X_5(M') = \widetilde{G}(0)B_\star^2$. Direct computation gives that

$$\widetilde{G}(d) = \sum_{m=1}^{M'} \sum_{h=1}^{H^m} \Big(P_{s_h^m, a_h^m} (\overline{\widetilde{V}}^m)^{2^{d+1}} - (P_{s_h^m, a_h^m} (\overline{\widetilde{V}}^m)^{2^d})^2\Big)$$

$$\leq \sum_{m=1}^{M'} \sum_{h=1}^{H^m} \left( P_{s_h^m, a_h^m} (\overline{\widetilde{V}}^m)^{2^{d+1}} - (\overline{\widetilde{V}}^m (s_{h+1}^m))^{2^{d+1}} \right) + \underbrace{\sum_{m=1}^{M'} (\overline{\widetilde{V}}^m (s_{H^m+1}^m))^{2^{d+1}}}_{\leq \widetilde{M}_1'}$$

$$+ \sum_{m=1}^{M'} \sum_{h=1}^{H^m} \left( (\overline{\widetilde{V}}^m (s_h^m))^{2^{d+1}} - (P_{s_h^m, a_h^m} \overline{\widetilde{V}}^m)^{2^{d+1}} \right) - \underbrace{\sum_{m=1}^{M'} (\overline{\widetilde{V}}^m (s_1^m))^{2^{d+1}}}_{\leq 0}$$

$$\leq \widetilde{F}(d+1) + \widetilde{M}_1' + 2^{d+1} \sum_{m=1}^{M'} \sum_{h=1}^{H^m} \max\{\overline{\widetilde{V}}^m (s_h^m) - P_{s_h^m, a_h^m} \overline{\widetilde{V}}^m, \, 0\}$$

$$= \widetilde{F}(d+1) + \widetilde{M}_1' + \frac{2^{d+1}}{B_\star} \sum_{m=1}^{M'} \sum_{h=1}^{H^m} \max\{\widetilde{V}^m (s_h^m) - P_{s_h^m, a_h^m} \widetilde{V}^m, \, 0\}$$

$$\overset{(i)}{\leq} \widetilde{F}(d+1) + \widetilde{M}_1' + \frac{2^{d+1}}{B_\star} \sum_{m=1}^{M'} \sum_{h=1}^{H^m} \beta^m (s_h^m, a_h^m)$$

$$= \widetilde{F}(d+1) + \widetilde{M}_1' + \frac{2^{d+1}}{B_\star} X_2(M'),$$

where $\widetilde{M}_1'$ denotes the number of intervals satisfying $\overline{\widetilde{V}}^m (s_{H^m+1}^m) \neq 0$; (i) come from Lem. 21.

For a fixed $d$, $\widetilde{F}(d)$ is a martingale. By taking $c = 1$ in Lem. 30, we have

$$\mathbb{P}\left[ \widetilde{F}(d) > 2\sqrt{2\widetilde{G}(d)(\log_2(T_{M'}) + \ln(2/\delta))} + 5(\log_2(T_{M'}) + \ln(2/\delta)) \right] \leq \delta.$$

Taking $\delta' = \delta/(\log_2(T_{M'}) + 1)$, using $x \geq \ln(x) + 1$ and finally swapping $\delta$ and $\delta'$, we have that

$$\mathbb{P}\left[ \widetilde{F}(d) > 2\sqrt{2\widetilde{G}(d)(2\log_2(T_{M'}) + \ln(2/\delta))} + 5(2\log_2(T_{M'}) + \ln(2/\delta)) \right] \leq \frac{\delta}{\log_2(T_{M'}) + 1}.$$

Taking a union bound over $d = 1, 2, \ldots, \log_2(T_{M'})$, we have that with probability $1 - \delta$,

$$\widetilde{F}(d) \leq 2\sqrt{2(2\log_2(T_{M'}) + \ln(2/\delta))} \cdot \sqrt{\widetilde{F}(d+1) + 2^{d+1}\frac{X_2(M')}{B_\star}}$$

$$+ 5(2\log_2(T_{M'}) + \ln(2/\delta)) + 2\sqrt{2(2\log_2(T_{M'}) + \ln(2/\delta))\widetilde{M}_1'}.$$

From Lem. 32, taking $\lambda_1 = T_{M'}$, $\lambda_2 = 2\sqrt{2(2\log_2(T_{M'}) + \ln(2/\delta))}$, $\lambda_3 = X_2(M')/B_\star$, $\lambda_4 = 5(2\log_2(T_{M'}) + \ln(2/\delta)) + 2\sqrt{2(2\log_2(T_{M'}) + \ln(2/\delta))\widetilde{M}_1'}$, we have that

$$\widetilde{F}(1) \leq O\left( \log_2(T_{M'}) + \ln(2/\delta) + \frac{X_2(M')}{B_\star} + \widetilde{M}_1' \right).$$

Since $V^\star(g) - V^m(g) = 0 - 0 = 0$, similar as bounding $M_1'$, we have $\widetilde{M}_1' \leq O(SA\log_2(T_{M'}))$. Hence with probability $1 - \delta$, we have

$$X_5(M') \leq O\left( B_\star^2 SA(\log_2(T_{M'}) + \ln(2/\delta)) + B_\star X_2(M') \right).$$

Throughout the proof, the inequality $O(\sqrt{xy}) \leq O(x + y)$ is utilized to simplify the bound. $\qquad \square$

### ④ Finally, bind them together.

Let $\bar{\iota}_{M'} := \ln\left( \frac{12SAS'T_{M'}^2}{\delta} \right) + \log_2((\max\{B_\star, 1\})^2 T_{M'}) + \ln\left( \frac{2}{\delta} \right)$ be the upper bound of all previous log terms.

$$X_2(M') \leq O\left( \sqrt{SAX_4(M')}\bar{\iota}_{M'} + \sqrt{S^2 AX_5(M')}\bar{\iota}_{M'} \right.$$

$$+ SA\bar{\iota}_{M'}^{3/2} + \sqrt{SAC_{M'}}\bar{\iota}_{M'} + B_\star S^2 A\bar{\iota}_{M'}^2 + BS^{3/2} A\bar{\iota}_{M'}^2\Big),$$

$$X_4(M') \le O\big(B_\star(C_{M'} + X_2(M')) + (B_\star^2 SA + B_\star)\bar{\iota}_{M'}\big),$$

$$X_5(M') \le O\big(B_\star^2 SA\bar{\iota}_{M'} + B_\star X_2(M')\big).$$

This implies that

$$X_2(M') \overset{(i)}{\le} O\Big(\sqrt{B_\star S^2 A\bar{\iota}_{M'}} \cdot \sqrt{X_2(M')} + (\sqrt{B_\star} + 1)\sqrt{SAC_{M'}}\bar{\iota}_{M'} + BS^2 A\bar{\iota}_{M'}^2\Big)$$

$$\le O\Big(\max\Big\{\sqrt{B_\star S^2 A\bar{\iota}_{M'}} \cdot \sqrt{X_2(M')},\ (\sqrt{B_\star} + 1)\sqrt{SAC_{M'}}\bar{\iota}_{M'} + BS^2 A\bar{\iota}_{M'}^2\Big\}\Big),$$

where (i) uses the assumption $B \ge \max\{B_\star, 1\}$ to simplify the bound. Considering terms in $\max\{\}$ separately, we obtain two bounds:

$$X_2(M') \le O(B_\star S^2 A\bar{\iota}_{M'}^2),$$

$$X_2(M') \le O((\sqrt{B_\star} + 1)\sqrt{SAC_{M'}}\bar{\iota}_{M'} + BS^2 A\bar{\iota}_{M'}^2).$$

By taking the maximum of these bounds, we have

$$X_2(M') \le O((\sqrt{B_\star} + 1)\sqrt{SAC_{M'}}\bar{\iota}_{M'} + BS^2 A\bar{\iota}_{M'}^2).$$

## F  Technical Lemmas

**Lemma 26** (Bennett's Inequality, anytime version). *Let $Z, Z_1, \ldots, Z_n$ be i.i.d. random variables with values in $[0, b]$ and let $\delta > 0$. Define $\mathbb{V}[Z] = \mathbb{E}[(Z - \mathbb{E}[Z])^2]$. Then we have*

$$\mathbb{P}\left[\forall n \ge 1,\ \left|\mathbb{E}[Z] - \frac{1}{n}\sum_{i=1}^n Z_i\right| > \sqrt{\frac{2\mathbb{V}[Z]\ln(4n^2/\delta)}{n}} + \frac{b\ln(4n^2/\delta)}{n}\right] \le \delta.$$

*Proof.* From Bennett's inequality, if the variables have values in $[0, 1]$, then for a specific $n \ge 1$,

$$\mathbb{P}\left[\left|\mathbb{E}[Z] - \frac{1}{n}\sum_{i=1}^n Z_i\right| > \sqrt{\frac{2\mathbb{V}[Z]\ln(2/\delta)}{n}} + \frac{\ln(2/\delta)}{n}\right] \le \delta.$$

We then choose $\delta \leftarrow \frac{\delta}{2n^2}$ and take a union bound over all possible values of $n \ge 1$, and the result follows given that $\sum_{n \ge 1} \frac{\delta}{2n^2} < \delta$. To account for the case $b \ne 1$ we apply the result to $(Z_n/b)$. □

**Lemma 27** (Theorem 4 in Maurer and Pontil [2009], anytime version). *Let $Z, Z_1, \ldots, Z_n$ $(n \ge 2)$ be i.i.d. random variables with values in $[0, b]$ and let $\delta > 0$. Define $\bar{Z} = \frac{1}{n}Z_i$ and $\hat{V}_n = \frac{1}{n}\sum_{i=1}^n (Z_i - \bar{Z})^2$. Then we have*

$$\mathbb{P}\left[\forall n \ge 1,\ \left|\mathbb{E}[Z] - \frac{1}{n}\sum_{i=1}^n Z_i\right| > \sqrt{\frac{2\hat{V}_n\ln(4n^2/\delta)}{n-1}} + \frac{7b\ln(4n^2/\delta)}{3(n-1)}\right] \le \delta.$$

**Lemma 28** (Popoviciu's Inequality). *Let $X$ be a random variable whose value is in a fixed interval $[a, b]$, then $\mathbb{V}[X] \le \frac{1}{4}(b - a)^2$.*

**Lemma 29** (Lemma 11 in Zhang et al. [2021c]). *Let $(M_n)_{n \ge 0}$ be a martingale such that $M_0 = 0$ and $|M_n - M_{n-1}| \le c$ for some $c > 0$ and any $n \ge 1$. Let $\mathrm{Var}_n = \sum_{k=1}^n \mathbb{E}[(M_k - M_{k-1})^2|\mathcal{F}_{k-1}]$ for $n \ge 0$, where $\mathcal{F}_k = \sigma(M_1, \ldots, M_k)$. Then for any positive integer $n$ and any $\epsilon, \delta > 0$, we have that*

$$\mathbb{P}\Big[|M_n| \ge 2\sqrt{2\mathrm{Var}_n \ln(1/\delta)} + 2\sqrt{\epsilon \ln(1/\delta)} + 2c\ln(1/\delta)\Big] \le 2\left(\log_2\left(\frac{nc^2}{\epsilon}\right) + 1\right)\delta.$$

**Lemma 30.** *Let $(M_n)_{n \ge 0}$ be a martingale such that $M_0 = 0$ and $|M_n - M_{n-1}| \le c$ for some $c > 0$ and any $n \ge 1$. Let $\mathrm{Var}_n = \sum_{k=1}^n \mathbb{E}[(M_k - M_{k-1})^2|\mathcal{F}_{k-1}]$ for $n \ge 0$, where $\mathcal{F}_k = \sigma(M_1, \ldots, M_k)$. Then for any positive integer $n$ and $\delta \in (0, 2(nc^2)^{1/\ln 2}]$, we have that*

$$\mathbb{P}\Big[|M_n| \ge 2\sqrt{2\mathrm{Var}_n(\log_2(nc^2) + \ln(2/\delta))} + 2\sqrt{\log_2(nc^2) + \ln(2/\delta)} + 2c(\log_2(nc^2) + \ln(2/\delta))\Big] \le \delta.$$

*Proof.* Take $\epsilon = 1$ and $\delta' = 2(\log_2(nc^2) + 1)\delta$ in Lem. 29. By $x \geq \ln(x) + 1$, we have

$$\ln(1/\delta) = \ln(2(\log_2(nc^2) + 1)/\delta') = \ln(\log_2(nc^2) + 1) + \ln(2/\delta') \leq \log_2(nc^2) + \ln(2/\delta').$$

Hence,

$$\mathbb{P}\Big[|M_n| \geq 2\sqrt{2\mathrm{Var}_n(\log_2(nc^2) + \ln(2/\delta'))} + 2\sqrt{\log_2(nc^2) + \ln(2/\delta')} + 2c(\log_2(nc^2) + \ln(2/\delta'))\Big]$$

$$\leq \mathbb{P}\Big[|M_n| \geq 2\sqrt{2\mathrm{Var}_n \ln(1/\delta)} + 2\sqrt{\ln(1/\delta)} + 2c\ln(1/\delta)\Big]$$

$$\leq \delta'.$$

By swapping $\delta$ and $\delta'$ we complete the proof. $\qquad\square$

**Lemma 31** (Lemma 11 in Zhang et al. [2021a]). *Let $\lambda_1, \lambda_2, \lambda_4 \geq 0$, $\lambda_3 \geq 1$ and $i' = \log_2 \lambda_1$. Let $a_1, a_2, \ldots, a_{i'}$ be non-negative reals such that $a_i \leq \lambda_1$ and $a_i \leq \lambda_2\sqrt{a_{i+1} + 2^{i+1}\lambda_3} + \lambda_4$ for any $1 \leq i \leq i'$. Then we have that $a_1 \leq \max\{(\lambda_2 + \sqrt{\lambda_2^2 + \lambda_4})^2, \lambda_2\sqrt{8\lambda_3} + \lambda_4\}$.*

**Lemma 32.** *Let $\lambda_1, \lambda_2, \lambda_4 \geq 0$, $\lambda_3 \geq 1$ and $i' = \log_2 \lambda_1$. Let $a_1, a_2, \ldots, a_{i'}$ be non-negative reals such that $a_i \leq \lambda_1$ and $a_i \leq \lambda_2\sqrt{a_{i+1} + 2^{i+1}\lambda_3} + \lambda_4$ for any $1 \leq i \leq i'$. Then we have that $a_1 \leq O(\lambda_2^2 + \lambda_3 + \lambda_4)$.*

*Proof.* Since $\max\{a, b\} \leq a + b$ and $2ab \leq a^2 + b^2$ for any choice of non-negative $a$ and $b$, we can transform the result of Lem. 31 into

$$a_1 \leq \max\left\{\left(\lambda_2 + \sqrt{\lambda_2^2 + \lambda_4}\right)^2, \lambda_2\sqrt{8\lambda_3} + \lambda_4\right\}$$

$$\leq O\left(\left(\lambda_2 + \sqrt{\lambda_2^2 + \lambda_4}\right)^2 + \lambda_2\sqrt{8\lambda_3} + \lambda_4\right)$$

$$\leq O(\lambda_2^2 + \lambda_2^2 + \lambda_4 + \lambda_2^2 + \lambda_3 + \lambda_4)$$

$$\leq O(\lambda_2^2 + \lambda_3 + \lambda_4).$$

$\qquad\square$

**Lemma 33.** *For random variable $Z \in [0, 1]$, $\mathbb{V}[Z] \leq \mathbb{E}[Z]$.*

*Proof.* $\mathbb{V}[Z] = \mathbb{E}[Z^2] - (\mathbb{E}[Z])^2 \leq \mathbb{E}[Z^2] \leq \mathbb{E}[Z]$. $\qquad\square$

**Lemma 34.** *For any $a, b \in [0, 1]$ and $k \in \mathbb{N}$, $a^k - b^k \leq k\max\{a - b, 0\}$.*

*Proof.* $a^k - b^k = (a - b)\sum_{i=0}^{k-1} a^i b^{k-1-i} \leq \max\{a - b, 0\} \cdot \sum_{i=0}^{k-1} 1 = k\max\{a - b, 0\}$. $\qquad\square$

**Lemma 35.** *For $a, b, x \geq 0$, $x \leq a\sqrt{x} + b$ implies $x \leq (a + \sqrt{b})^2$.*

*Proof.* $x \leq a\sqrt{x} + b \Rightarrow x \leq \left(\frac{a + \sqrt{a^2 + b}}{2}\right)^2 \leq (a + \sqrt{b})^2$. $\qquad\square$

## G   Computational Complexity of `EB-SSP`

Here we complement Remarks 1 and 3 on the computational complexity of `EB-SSP` (Alg. 1).

The computational complexity of a `VISGO` procedure can be bounded as $O(\frac{S^2 A}{1-\rho}\log(B_\star/\epsilon_{\mathrm{VI}}))$ (assuming for simplicity that $B_\star \geq 1$, otherwise replace $\max\{B_\star, 1\} \leftarrow B_\star$). By the fact that total number of `VISGO` procedure is bounded by $O(SA\log T)$, we derive $\log(B_\star/\epsilon_{\mathrm{VI}}) = O(SA\log(B_\star T))$ by choice of $\epsilon_{\mathrm{VI}}$. As a result, the total computational complexity for `EB-SSP` is $O(TS^2 A \cdot SA\log(B_\star T) \cdot SA\log T)$, which is polynomially bounded and in particular near-linear in $T$. Also note that $T$ is bounded polynomially w.r.t. $K$ as shown in the various cases of Sect. 4.1. Indeed, in the case of positive costs lower bounded by $c_{\min} > 0$, Cor. 5 entails that $T \leq c_{\min}^{-1} KV^\star(s_0) + c_{\min}^{-1}\widetilde{O}(B_\star\sqrt{SAK} + B_\star S^2 A)$. In the general cost case, the cost perturbation

trick is applied and the minimum cost becomes $K^{-n}$ for Cor. 6 or $(\overline{T}_\star K)^{-1}$ for Cor. 8, i.e., $c_{\min}^{-1}$ depends polynomially on $K$.

We note that the analysis of the computational complexity of `EB-SSP` may likely be refined. Indeed, we see that i) on the one hand, if $n(s,a)$ is small, then the optimistic skewing of $\widetilde{P}_{s,a}$ is not too small so the probability of reaching the goal from $(s,a)$ is not too small (so the associated contraction modulus is bounded away from 1) and ii) on the other hand, if $n(s,a) \to +\infty$, then $\widetilde{P}_{s,a} \to \widehat{P}_{s,a} \to P_{s,a}$, so to the limit we should recover the convergence properties of VI of the optimal Bellman operator under the true model, which by assumption admits a proper policy in $P$. Thus we see that studying further the "intermediate regime" may bring into the picture the computational complexity of running VI in the true model, yet this is not our main focus here, as our complexity analysis is sufficient to ensure the computational efficiency of `EB-SSP`.

# H  Unknown $B_\star$: **Parameter-Free** `EB-SSP`

In this section, we relax the assumption that (an upper bound of) $B_\star$ is known to `EB-SSP`. In Alg. 2 we propose a parameter-free `EB-SSP` that bypasses the requirement $B \geq B_\star$ (line 2 of Alg. 1) to tune the exploration bonus. As in Sect. 4 we consider for ease of exposition that $B_\star \geq 1$. We structure the section as follows: App. H.1 presents our algorithm and provides intuition, App. H.2 spells out its regret guarantee, and App. H.3 gives its proof.

## H.1  Algorithm and Intuition

Parameter-free `EB-SSP` (Alg. 2) initializes an estimate $\widetilde{B} = 1$ and decomposes the time steps into *phases*, indexed by $\phi$. The execution of a phase is reported in the subroutine `PHASE` (Alg. 3). Given any estimate $\widetilde{B}$, a subroutine `PHASE` has the same structure as Alg. 1, up to two key differences:

- **Halting due to exceeding cumulative cost.** `PHASE` tracks the cumulative cost within the current phase, and terminates whenever it exceeds a threshold $C_{\text{bound}}$ (Eq. 17) that depends on $\widetilde{B}$, $S$, $A$, $\delta$ and the current episode and time indexes $k$ and $t$, which are all computable quantities to the agent.
- **Halting due to exceeding `VISGO` range.** During each `VISGO` procedure, `PHASE` tracks the range of the value function $V^{(i)}$ at each `VISGO` iteration $i$, and terminates if $\|V^{(i)}\|_\infty > \widetilde{B}$.

The estimate $\widetilde{B}$ can be incremented in two different ways and speeds:

- **Doubling increment of $\widetilde{B}$.** On the one hand, whenever a phase ends (i.e., one of the two halting conditions above is met), $\widetilde{B}$ is doubled ($\widetilde{B} \leftarrow 2\widetilde{B}$).
- **Episode-driven increment of $\widetilde{B}$.** On the other hand, at the beginning of each new episode $k$, the estimate is automatically increased to $\widetilde{B} \leftarrow \max\{\widetilde{B}, \sqrt{k}/(S^{3/2}A^{1/2})\}$.

We now explain the rationale behind our scheme:

- *Reason for episode-driven increment of $\widetilde{B}$.* The fact that $\widetilde{B}$ grows as a function of $k$ implies that at some (unknown) point it will hold that $\widetilde{B} \geq B_\star$ for large enough $k$. This will enable us to recover the analysis and the regret bound of Thm. 3.
- *Reason for doubling increment of $\widetilde{B}$.* The doubling increment comes into play whenever a phase terminates due to an exceeding cumulative cost or `VISGO` range. At this point, the agent becomes aware that $\widetilde{B}$ is too small and thus it doubles it. It is crucial to allow intra-episode increments of $\widetilde{B}$ to avoid getting *stuck* in an episode with an underestimate $\widetilde{B} < B_\star$.
- *Reason for cumulative cost halting.* The cost threshold $C_{\text{bound}}$ is designed so that (w.h.p.) it can be exceeded at most once in the case of $\widetilde{B} \geq B_\star$, and so that it can serve as a tight enough bound on the regret in the case of $\widetilde{B} < B_\star$.
- *Reason for `VISGO` range halting.* The threshold $\widetilde{B}$ on the range of the `VISGO` value functions is chosen so that (w.h.p.) it is never exceeded in the case of $\widetilde{B} \geq B_\star$, and so that it can serve as a guarantee of finite-time near-convergence of a `VISGO` procedure (i.e., the contraction property) in the case of $\widetilde{B} < B_\star$.

## H.2 Regret Guarantee of Parameter-Free EB-SSP

Parameter-free EB-SSP satisfies the following guarantee (which extends Thm. 3 to unknown $B_\star$).

**Restatement of Theorem 9.** Assume the conditions of Lem. 2 hold. Then with probability at least $1 - \delta$ the regret of parameter-free EB-SSP (Alg. 2, App. H) can be bounded by

$$R_K = O\left( R_K^\star \log\left(\frac{B_\star SAT}{\delta}\right) + B_\star^3 S^3 A \log^3\left(\frac{B_\star SAT}{\delta}\right) \right),$$

where $T$ is the cumulative time within the $K$ episodes and $R_K^\star$ bounds the regret after $K$ episodes of EB-SSP in the case of known $B_\star$ (i.e., the bound of Thm. 3 with $B = B_\star$).

As a result, parameter-free EB-SSP is able to circumvent the knowledge of $B_\star$ at the cost of only logarithmic and lower-order terms.

## H.3 Proof of Theorem 9

We begin by defining notations and concepts exclusively used in this section:

- $C_t$ denotes the cumulative cost up to time step $t$ (included) that is accumulated in the execution of the subroutine PHASE in which time step $t$ belongs. Importantly, note that the cumulative cost $C_t$ is initialized to $0$ at the beginning of each PHASE (line 5 of Alg. 3). Also note that re-planning (i.e., a VISGO procedure) occurs whenever the estimate $\widetilde{B}$ is changed.
- Denote by $t_m$ the time step at the end of the current interval $m$, and by $k_m$ the episode in which the time step $t_m$ belongs. $\widetilde{B}_m$ denotes the value of $\widetilde{B}$ at time step $t_m$. $C_m$ denotes $C_{t_m}$, i.e., the cumulative cost up to interval $m$ (included) in the execution of the PHASE in which interval $m$ belongs.

Unlike EB-SSP of Alg. 1, the parameter-free version has an increasing $\widetilde{B}$ throughout the process. To utilize the regret bounds (Thm. 3 and Eq. 13) in the case of $\widetilde{B} \geq B_\star$, slight modifications are needed to be applied to the algorithm and some lemmas.

***Modification to EB-SSP.*** Previously, EB-SSP accepted a single value $B \geq \max\{B_\star, 1\}$ to compute the bonuses in Eq. 2. To satisfy the same regret bound when $\widetilde{B}$ changes, we require EB-SSP to accept a series of $B_k$ for $k \in \mathbb{N}^+$, such that $\max\{B_\star, 1\} \leq B_k \leq B$ for any $k$. In any episode $k$, the analysis simply substitutes $B_k$ for $B$ in Eq. 2.

***Modifications to the proofs of Lem. 17, 18 and 20.*** In the original version of the proofs, we proved the lemmas for any update of value functions, without mentioning any time relevant variables. Now since $B$ relies on episode $k$, the modified proofs need to incorporate the changes. Suppose that we are examining $Q(s, a)$, $V(s)$, $b(s, a)$ and $\beta(s, a)$ for any state-action pair $(s, a) \in \mathcal{S} \times \mathcal{A}$ in episode $k$. Lem. 17 and Lem. 18 utilize the property stated in Lem. 16, and the $B$ in Lem. 16 is a parameter that is able to vary each time step we utilize Lem. 16. Thus, in the proofs of Lem. 17, 18 and 20, all the $B$'s are substituted with $B_k$'s to ensure that these lemmas are compatible with our modified setting.

***Modification to the proof of bounding $\beta^m$ in App. E.3.*** Suppose that interval $m$ is in episode $k$ and recall that $B_k \leq B$, then

$$b^m(s, a) = \max\left\{ c_1 \sqrt{\frac{\mathbb{V}(\widetilde{P}_{s,a}, V^{(l)})\iota_{s,a}}{n^m(s, a)}}, \ c_2 \frac{B_k \iota_{s,a}}{n^m(s, a)} \right\} + c_3 \sqrt{\frac{\widehat{c}^m(s, a)\iota_{s,a}}{n^m(s, a)}} + c_4 \frac{B_k \sqrt{S' \iota_{s,a}}}{n^m(s, a)}$$

$$\leq O\left( \sqrt{\frac{\mathbb{V}(\widetilde{P}_{s,a}, V^{(l)})\iota_{s,a}}{n^m(s, a)}} + \frac{B\iota_{s,a}}{n^m(s, a)} + \sqrt{\frac{\widehat{c}^m(s, a)\iota_{s,a}}{n^m(s, a)}} + \frac{B\sqrt{S\iota_{s,a}}}{n^m(s, a)} \right).$$

Combining the above bound of $b^m(s, a)$ with Lem. 20, we get that the bound of $\beta^m$ in App. E.3 is unchanged.

Equipped with the slight modifications mentioned above, we now derive two key properties on which the analysis of parameter-free EB-SSP relies:

***Property 1: Optimism avoids the first halting condition.*** Let us study any phase starting with estimate $\widetilde{B} \geq B_\star$. From Eq. 13 (which is the interval-generalization of Thm. 3), for a fixed initial

state $s_0$ and a fixed interval $m$, the cumulative cost can be bounded with probability $1 - \delta$ by

$$k_m V^\star(s_0) + x \left( B_\star \sqrt{SAk_m} \log_2 \left( \frac{B_\star t_m SA}{\delta} \right) + \widetilde{B}_m S^2 A \log_2^2 \left( \frac{B_\star t_m SA}{\delta} \right) \right), \qquad (15)$$

where $x > 0$ is a large enough absolute constant (which can be retraced in the analysis leading to Eq. 13). By scaling $\delta \leftarrow \delta/(2St_m^2)$ for each $m \leq M$, we have the following cumulative cost bound that holds for any initial state in $\mathcal{S}$ and any interval $m \leq M$, with probability $1 - \delta$,

$$C_m \leq k_m V^\star(s_0) + x \left( B_\star \sqrt{SAk_m} \log_2 \left( \frac{B_\star t_m SA \cdot 2St_m^2}{\delta} \right) + \widetilde{B}_m S^2 A \log_2^2 \left( \frac{B_\star t_m SA \cdot 2St_m^2}{\delta} \right) \right)$$

$$\leq k_m B_\star + 3x \left( B_\star \sqrt{SAk_m} \log_2 \left( \frac{B_\star t_m SA}{\delta} \right) + \widetilde{B}_m S^2 A \log_2^2 \left( \frac{B_\star t_m SA}{\delta} \right) \right).$$

Since we are in the case of $\widetilde{B}_m \geq B_\star$, we have

$$C_m \leq k_m \widetilde{B}_m + 3x \left( \widetilde{B}_m \sqrt{SAk_m} \log_2 \left( \frac{\widetilde{B}_m t_m SA}{\delta} \right) + \widetilde{B}_m S^2 A \log_2^2 \left( \frac{\widetilde{B}_m t_m SA}{\delta} \right) \right). \qquad (16)$$

Since costs are non-negative, for any $t \leq t_m$, we have $C_t \leq C_m$ hence $C_t$ must also satisfy the bound of Eq. 16. There remains to predict the values of $k_m$, $t_m$, $\widetilde{B}_m$, given the current $k_{\text{cur}}$, $t_{\text{cur}}$, $\widetilde{B}_{\text{cur}}$. The upper bounds for $k_m$ and $\widetilde{B}_m$ are $k_{\text{cur}}$ and $\widetilde{B}_{\text{cur}}$ respectively, since they can only be incremented when reaching the goal $g$, which is a condition for ending the current interval. The upper bound for $t_m$ can be derived using the pigeonhole principle: since $t_{\text{cur}} = \sum_{(s,a) \in \mathcal{S} \times \mathcal{A}} n(s,a)$, we know that $2t_{\text{cur}} > \sum_{(s,a) \in \mathcal{S} \times \mathcal{A}} (2n(s,a) - 1)$. Thus by time step $2t_{\text{cur}}$ there must exist a trigger condition, which is a condition for ending the current interval. Hence, by replacing $k_m \leftarrow k_{\text{cur}}$, $\widetilde{B}_m \leftarrow \widetilde{B}_{\text{cur}}$ and $t_m \leftarrow 2t_{\text{cur}}$ in Eq. 16, we get, with probability at least $1 - \delta$, that the cumulative cost within a phase that starts with $\widetilde{B} \geq B_\star$ has the following anytime upper bound

$$C_{t_{\text{cur}}} \leq k_{\text{cur}} \widetilde{B}_{\text{cur}} + 3x \left( \widetilde{B}_{\text{cur}} \sqrt{SAk_{\text{cur}}} \log_2 \left( \frac{2\widetilde{B}_{\text{cur}} t_{\text{cur}} SA}{\delta} \right) + \widetilde{B}_{\text{cur}} S^2 A \log_2^2 \left( \frac{2\widetilde{B}_{\text{cur}} t_{\text{cur}} SA}{\delta} \right) \right).$$

Note that this bound corresponds exactly to the cumulative cost threshold $C_{\text{bound}}$ in Eq. 17. This means that with probability at least $1 - \delta$, the first halting condition cannot be met in a phase that starts with $\widetilde{B} \geq B_\star$.

***Property 2: Optimism avoids the second halting condition.*** Let us consider the case of $\widetilde{B} \geq B_\star$ whenever the algorithm re-plans (i.e., running `VISGO` procedure). The proof of Lem. 17 ensures that at any iteration, $\|V^{(i)}\|_\infty \leq B_\star \leq \widetilde{B}$, so the second halting condition is never met under the same high-probability event as above.

***Implications.*** The two properties above indicate that, if a phase starts with estimate $\widetilde{B} \geq B_\star$, with probability at least $1 - \delta$, this phase will never halt due to the two halting conditions (it can only terminate if it completes the final episode $K$), and Alg. 2 will thus never enter a new phase. Due to the doubling increment of $\widetilde{B}$ every time a phase ends, we can therefore bound the total number of phases as $\Phi \leq \lceil \log_2(B_\star) \rceil + 1$.

***Analysis.*** We now split the analysis of the regret contributions of the episodes in two *regimes*. To this end, let $\kappa_\star := \lceil B_\star^2 S^3 A \rceil$ denote a special episode (note that it is unknown to the learner since it depends on $B_\star$). We consider that the high-probability event mentioned above holds (which is the case with probability at least $1 - \delta$). Recall that at the beginning of each episode $k$, the algorithm sets $\widetilde{B} \leftarrow \max\{\widetilde{B}, \sqrt{k}/(S^{3/2} A^{1/2})\}$.

**① Regret contribution in the first regime (i.e., episodes $k < \kappa_\star$).**

We denote respectively by $R_{1 \to \kappa_\star}$ and $C_{1 \to \kappa_\star}$ the cumulative regret and the cumulative cost incurred by the algorithm before episode $\kappa_\star$ begins. For any phase $\phi$, we denote by

- $C_{1 \to \kappa_\star}^{(\phi)}$ the cumulative cost incurred during the time steps that are *both* in phase $\phi$ and in an episode $k < \kappa_\star$;

- $k^{(\phi)}$ the episode when phase $\phi$ ends;
- $t^{(\phi)}$ the time step when phase $\phi$ ends;
- $\widetilde{B}^{(\phi)}$ the value of $\widetilde{B}$ at the end of phase $\phi$.

Observe that

$$C_{1\to\kappa_\star} = \sum_{\phi=1}^{\Phi} C_{1\to\kappa_\star}^{(\phi)}.$$

Now, by definition of $\kappa^\star$, the episode-driven increment of $\widetilde{B}$ never exceeds $B_\star$, unless $\widetilde{B}$ is already larger or equal to $B_\star$ at the beginning of the phase. But Property 1 ensures that if $\widetilde{B} \geq B_\star$ in the beginning of a phase, then $\widetilde{B}$ will never be doubled afterwards. Hence, we are guaranteed that within the episodes $k < \kappa_\star$, the final value of the estimate $\widetilde{B}$ is at most $2B_\star$.

Since PHASE tracks the cumulative cost at each step using the threshold in Eq. 17 and since $c_t \leq 1$, by the fact that $C_{\text{bound}}$ is monotonously increasing with respect to $t$, we have that for any phase $\phi$,

$$C_{1\to\kappa_\star}^{(\phi)} \leq k^{(\phi)}\widetilde{B}^{(\phi)} + 3x\left(\widetilde{B}^{(\phi)}\sqrt{SAk^{(\phi)}}\log_2\left(\frac{2\widetilde{B}^{(\phi)}t^{(\phi)}SA}{\delta}\right) + \widetilde{B}^{(\phi)}S^2A\log_2^2\left(\frac{2\widetilde{B}^{(\phi)}t^{(\phi)}SA}{\delta}\right)\right) + 1$$

$$\leq \kappa_\star(2B_\star) + 3x\left((2B_\star)\sqrt{SA\kappa_\star}\log_2\left(\frac{2(2B_\star)TSA}{\delta}\right) + (2B_\star)S^2A\log_2^2\left(\frac{2(2B_\star)TSA}{\delta}\right)\right) + 1$$

$$\leq O\left(B_\star^3 S^3 A + B_\star^2 S^2 A\log\left(\frac{B_\star TSA}{\delta}\right) + B_\star S^2 A\log^2\left(\frac{B_\star TSA}{\delta}\right)\right).$$

In addition, we recall that $\Phi \leq \lceil\log_2(B_\star)\rceil + 1$. Hence, by plugging in the definition of $\kappa^\star$, we can bound the cost (and thus the regret) accumulated over the episodes $k < \kappa_\star$ as follows

$$R_{1\to\kappa_\star} \leq C_{1\to\kappa_\star} \leq \sum_{\phi=1}^{\lceil\log_2(B_\star)\rceil+1} O\left(B_\star^3 S^3 A + B_\star^2 S^2 A\log\left(\frac{B_\star TSA}{\delta}\right) + B_\star S^2 A\log^2\left(\frac{B_\star TSA}{\delta}\right)\right)$$

$$\leq O\left(B_\star^3 S^3 A\log(B_\star) + B_\star^2 S^2 A\log\left(\frac{B_\star TSA}{\delta}\right)\log(B_\star)\right.$$

$$\left. + B_\star S^2 A\log^2\left(\frac{B_\star TSA}{\delta}\right)\log(B_\star)\right)$$

$$\leq O\left(B_\star^3 S^3 A\bar{\iota} + B_\star^2 S^2 A\bar{\iota}^2 + B_\star S^2 A\bar{\iota}^3\right).$$

② **Regret contribution in the second regime (i.e., episodes $k \geq \kappa_\star$).**

We denote respectively by $R_{\kappa_\star\to K}$ and $C_{\kappa_\star\to K}$ the cumulative regret and the cumulative cost incurred during the episodes $k \geq \kappa^\star$. By definition of $\kappa^\star$, the episode-driven increment of $\widetilde{B}$ ensures that $\widetilde{B} \geq B_\star$. During this second regime there may be at most two phases: one that started at an episode $k < \kappa_\star$ (i.e., in the first regime) and that overlaps the two regimes, and one starting after that (note that properties 1 and 2 ensure that at this point neither halting condition can end this phase since it started with estimate $\widetilde{B} \geq B_\star$, thus it lasts until the end of the learning interaction). In addition, we can upper bound $\widetilde{B}$ as follows

$$\widetilde{B} \leq \max\left\{2B_\star, \frac{2\sqrt{K}}{S^{3/2}A^{1/2}}\right\}.$$

We now introduce a fourth condition of stopping an interval to the analysis performed in Sect. D.3: (4) an interval ends when a subroutine PHASE ends. This implies that the policy always stays the same within an interval when running Alg. 2. Condition (4) is met at most once in the second regime.

We now focus on only the second regime: we re-index intervals by $1, 2, \ldots, M'$ and let $T_m$ denote the time step counting from the beginning of $\kappa_\star$ to the end of interval $m$. To bound $R_{\kappa_\star\to K}$, we need to adapt the proofs in App. D.5 and App. E.3 to be compatible with our new interval decomposition. Concretely, there are two slight modifications in the analysis of the second regime:

- Statistics: For any statistic (i.e., $N(s, a, s')$, $\theta(s, a)$ and $\widehat{c}(s, a)$ for any $(s, a, s') \in \mathcal{S} \times \mathcal{A} \times \mathcal{S}'$), instead of learning from scratch, PHASE reuses all samples collected thus far. This difference does not affect the regret bound and the probability, since it can be viewed by taking a partial sum of terms in $\widetilde{R}_{M'}$.

- The regret decomposition: In the proof of Lem. 22, we need to incorporate condition (4) which is met at most once during the second regime. It falls into case (ii) in the proof of Lem. 22, which thus happens at most $2SA\log_2(T_{M'}) + 1$ times, and the regret decomposition should be

$$\widetilde{R}_{M'} \leq X_1(M') + X_2(M') + X_3(M') + 2B_\star SA\log_2(T_{M'}) + B_\star.$$

Hence by incorporating these slight modifications in the proof of Thm. 3, we get probability at least $1 - \delta$,

$$R_{\kappa_\star \to K} \leq O\left( B_\star \sqrt{SAK} \log\left(\frac{B_\star TSA}{\delta}\right) + S^2 A \widetilde{B}_{M'} \log^2\left(\frac{B_\star TSA}{\delta}\right) \right)$$

$$\leq O\left( B_\star \sqrt{SAK} \log\left(\frac{B_\star TSA}{\delta}\right) + S^2 A \frac{\sqrt{K}}{S^{3/2} A^{1/2}} \log^2\left(\frac{B_\star TSA}{\delta}\right) \right)$$

$$\leq O\left( B_\star \sqrt{SAK}\overline{\iota} + \sqrt{SAK}\overline{\iota}^2 \right).$$

③ **Combining the regret contributions in the two regimes.**

The overall regret is bounded with probability at least $1 - \delta$ by

$$R_K = R_{1 \to \kappa_\star} + R_{\kappa_\star \to K} \leq O\left( B_\star \sqrt{SAK}\overline{\iota} + \sqrt{SAK}\overline{\iota}^2 + B_\star^3 S^3 A\overline{\iota} + B_\star^2 S^2 A\overline{\iota}^2 + B_\star S^2 A\overline{\iota}^3 \right).$$

There remains to plug in the definition of $\overline{\iota}$. Denote by $T$ the cumulative time within the $K$ episodes and by $R_K^\star$ the regret after $K$ episodes of EB-SSP in the case of known $B_\star$ (i.e., the bound of Thm. 3 with $B = B_\star$). Then with probability at least $1 - \delta$ the regret of parameter-free EB-SSP can be bounded as

$$R_K = O\left( R_K^\star + \sqrt{SAK} \log^2\left(\frac{B_\star SAT}{\delta}\right) + B_\star^3 S^3 A \log^3\left(\frac{B_\star SAT}{\delta}\right) \right)$$

$$= O\left( R_K^\star \log\left(\frac{B_\star SAT}{\delta}\right) + B_\star^3 S^3 A \log^3\left(\frac{B_\star SAT}{\delta}\right) \right).$$

This concludes the proof of Thm. 9.

**Remark 4.** At a high level, our analysis to circumvent the knowledge of $B_\star$ boils down to the following argument: if the estimate is too small, we bound the regret by the cumulative cost; otherwise if it is large enough, we recover the regret bound under a known upper bound on $B_\star$. Interestingly, this somewhat resembles the reasoning behind the schemes for unknown SSP-diameter $D$ in the adversarial SSP algorithms of Rosenberg and Mansour [2021, App. I] and Chen and Luo [2021, App. E] (recall that $D := \max_{s \in \mathcal{S}} \min_{\pi \in \Pi_{\text{proper}}} T^\pi(s)$ and that $B_\star \leq D \leq T_\star$). Note however that these schemes change their algorithms' structure: whenever the agent is in a state that is insufficiently visited, it executes the Bernstein-SSP algorithm of Rosenberg et al. [2020] with unit costs until the goal is reached. In other words, these schemes first learn to reach the goal (regardless of the costs) and then focus on minimizing the costs to goal. In contrast, our scheme for unknown $B_\star$ targets the original SSP objective from the start and it does *not* fundamentally alter our algorithm EB-SSP with known $B_\star$. Indeed, the only addition of parameter-free EB-SSP is a *dual tracking* of the cumulative costs and VISGO ranges, and a *careful increment* of the estimate $\widetilde{B}$ in the bonus. Finally, our scheme only adds "horizon-free" lower-order terms (i.e., $B_\star, S, A$) as shown in Thm. 9, as opposed to the aforementioned schemes that introduce a lower-order dependence on the SSP-diameter $D$, which may be much larger than $B_\star$.

---

**Algorithm 2:** Algorithm for unknown $B_\star$: Parameter-free EB-SSP

---

1  **Input:** $\mathcal{S}$, $s_0 \in \mathcal{S}$, $g \notin \mathcal{S}$, $\mathcal{A}$, $\delta$.
2  **Optional input:** cost perturbation $\eta \in [0,1]$.
3  Set up **global constants:** $\mathcal{S}$, $\mathcal{A}$, $s_0 \in \mathcal{S}$, $g \notin \mathcal{S}$, $\eta$.
4  Set up **global variables:** $t$, $j$, $N()$, $n()$, $\widehat{P}$, $\theta()$, $\widehat{c}()$, $Q()$, $V()$.
5  Set estimate $\widetilde{B} \leftarrow 1$.
6  Set current starting state $s_{\text{start}} \leftarrow s_0$.
7  Set $t \leftarrow 1$, $k \leftarrow 1$, $j \leftarrow 0$.
8  For $(s,a,s') \in \mathcal{S} \times \mathcal{A} \times \mathcal{S}'$, set $N(s,a) \leftarrow 0$; $n(s,a) \leftarrow 0$; $N(s,a,s') \leftarrow 0$; $\widehat{P}_{s,a,s'} \leftarrow 0$; $\theta(s,a) \leftarrow$
    $0$; $\widehat{c}(s,a) \leftarrow 0$; $Q(s,a) \leftarrow 0$; $V(s) \leftarrow 0$.
9  Set phase counter $\phi \leftarrow 1$.
10  **while** True **do**
11     Set $s_{\text{cur}}$, $\widetilde{B}_{\text{cur}}$, $k_{\text{cur}} \leftarrow$ PHASE $(s_{\text{start}}, \widetilde{B}, k)$  (Alg. 3).
12     \\ PHASE *halts because of $B_\star$ underestimation, entering a new **phase***
13     Set $s_{\text{start}} \leftarrow s_{\text{cur}}$, $k \leftarrow k_{\text{cur}}$, $\widetilde{B} \leftarrow 2\widetilde{B}_{\text{cur}}$, and increment phase index $\phi \leftarrow \phi + 1$.

---

**Algorithm 3:** Subroutine PHASE

---

1  **Input:** $s_{\text{start}} \in \mathcal{S}$, $\widetilde{B}$, $k$.
2  **Global constants:** $\mathcal{S}$, $\mathcal{A}$, $s_0 \in \mathcal{S}$, $g \notin \mathcal{S}$, $\eta$.
3  **Global variables:** $t$, $j$, $N()$, $n()$, $\widehat{P}$, $\theta()$, $\widehat{c}()$, $Q()$, $V()$.
4  **Specify:** Trigger set $\mathcal{N} \leftarrow \{2^{j-1} : j = 1, 2, \ldots\}$. Constants $c_1 = 6$, $c_2 = 36$, $c_3 = 2\sqrt{2}$, $c_4 = 2\sqrt{2}$.
    Large enough absolute constant $x > 0$ (so that Eq. 15 holds, see App. H.3).
5  Set $C \leftarrow 0$. \\ *Reinitialize cumulative cost tracker*
6  **for** episode $k_{\text{cur}} = k, k+1, \ldots$ **do**
7     **if** $\sqrt{k_{\text{cur}}}/(S^{3/2}A^{1/2}) > \widetilde{B}$ **then**
8         Set $\widetilde{B} \leftarrow \sqrt{k_{\text{cur}}}/(S^{3/2}A^{1/2})$, and set $j \leftarrow j+1$, $\epsilon_{\text{VI}} \leftarrow 2^{-j}/(SA)$.
9         Info, $Q$, $V \leftarrow$ VISGO $(\widetilde{B}, \epsilon_{\text{VI}})$.
10         **if** Info = Fail **then**
11             \\ *Second halting condition: VISGO range exceeds threshold*
12             **return** $s_t$, $\widetilde{B}$, $k_{\text{cur}}$.
13     Set $s_t \leftarrow \begin{cases} s_{\text{start}}, & k_{\text{cur}} = k, \\ s_0, & \text{otherwise.} \end{cases}$
14     **while** $s_t \neq g$ **do**
15         Take action $a_t = \arg\min_{a \in \mathcal{A}} Q(s_t, a)$, incur cost $c_t$ and observe next state $s_{t+1} \sim P(\cdot|s_t, a_t)$.
16         Set $(s, a, s', c) \leftarrow (s_t, a_t, s_{t+1}, \max\{c_t, \eta\})$ and $t \leftarrow t+1$.
17         Set $N(s,a) \leftarrow N(s,a) + 1$, $\theta(s,a) \leftarrow \theta(s,a) + c$, $C \leftarrow C + c$, $N(s,a,s') \leftarrow N(s,a,s') + 1$,
        and set

$$C_{\text{bound}} \leftarrow k_{\text{cur}}\widetilde{B} + 3x\left(\widetilde{B}\sqrt{SAk_{\text{cur}}}\log_2\left(\frac{2\widetilde{B}tSA}{\delta}\right) + \widetilde{B}S^2A\log_2^2\left(\frac{2\widetilde{B}tSA}{\delta}\right)\right). \quad (17)$$

18         **if** $C > C_{\text{bound}}$ **then**
19             \\ *First halting condition: cumulative cost exceeds threshold*
20             **return** $s_t$, $\widetilde{B}$, $k_{\text{cur}}$.
21         **if** $N(s,a) \in \mathcal{N}$ **then**
22             Set $\widehat{c}(s,a) \leftarrow \mathbb{I}[N(s,a) \geq 2]\frac{2\theta(s,a)}{N(s,a)} + \mathbb{I}[N(s,a) = 1]\theta(s,a)$ and $\theta(s,a) \leftarrow 0$.
23             For all $s' \in \mathcal{S}$, set $\widehat{P}_{s,a,s'} \leftarrow N(s,a,s')/N(s,a)$, $n(s,a) \leftarrow N(s,a)$, and set
            $j \leftarrow j+1$, $\epsilon_{\text{VI}} \leftarrow 2^{-j}/(SA)$.
24             Info, $Q$, $V \leftarrow$ VISGO $(\widetilde{B}, \epsilon_{\text{VI}})$.
25             **if** Info = Fail **then**
26                 \\ *Second halting condition: VISGO range exceeds threshold*
27                 **return** $s_t$, $\widetilde{B}$, $k_{\text{cur}}$.

**Algorithm 4:** Subroutine VISGO

1 **Inputs:** $\widetilde{B}$, $\epsilon_{\text{VI}}$.
2 **Global constants:** $\mathcal{S}$, $\mathcal{A}$, $s_0 \in \mathcal{S}$, $g \notin \mathcal{S}$, $\eta$.
3 **Global variables:** $t$, $j$, $N()$, $n()$, $\widehat{P}$, $\theta()$, $\widehat{c}()$, $Q()$, $V()$.
4 For all $(s, a, s') \in \mathcal{S} \times \mathcal{A} \times \mathcal{S}'$, set

$$\widetilde{P}_{s,a,s'} \leftarrow \frac{n(s,a)}{n(s,a)+1}\widehat{P}_{s,a,s'} + \frac{\mathbb{I}[s' = g]}{n(s,a)+1}.$$

5 For all $(s, a) \in \mathcal{S} \times \mathcal{A}$, set $n^+(s,a) \leftarrow \max\{n(s,a), 1\}$, $\iota_{s,a} \leftarrow \ln\left(\frac{12SAS'[n^+(s,a)]^2}{\delta}\right)$.
6 Set $i \leftarrow 0$, $V^{(0)} \leftarrow 0$, $V^{(-1)} \leftarrow +\infty$.
7 **while** $\|V^{(i)} - V^{(i-1)}\|_\infty > \epsilon_{\text{VI}}$ **do**
8 | For all $(s, a) \in \mathcal{S} \times \mathcal{A}$, set

$$b^{(i+1)}(s,a) \leftarrow \max\left\{c_1\sqrt{\frac{\mathbb{V}(\widetilde{P}_{s,a}, V^{(i)})\iota_{s,a}}{n^+(s,a)}}, c_2\frac{\widetilde{B}\iota_{s,a}}{n^+(s,a)}\right\} + c_3\sqrt{\frac{\widehat{c}(s,a)\iota_{s,a}}{n^+(s,a)}} + c_4\frac{\widetilde{B}\sqrt{S'\iota_{s,a}}}{n^+(s,a)},$$
(18)

$$Q^{(i+1)}(s,a) \leftarrow \max\left\{\widehat{c}(s,a) + \widetilde{P}_{s,a}V^{(i)} - b^{(i+1)}(s,a), 0\right\},$$
(19)

$$V^{(i+1)}(s) \leftarrow \min_a Q^{(i+1)}(s,a).$$
(20)

9 | Set $V^{(i+1)}(g) \leftarrow 0$ and $i \leftarrow i + 1$.
10 | **if** $\|V^{(i)}\|_\infty > \widetilde{B}$ **then**
11 | | \\ *Second halting condition:* VISGO *range exceeds threshold*
12 | | **return** Fail, $Q^{(i)}$, $V^{(i)}$.

13 **return** Success, $Q^{(i)}$, $V^{(i)}$.