# OpenReview forum: "Stochastic Shortest Path: Minimax, Parameter-Free and Towards Horizon-Free Regret"
_NeurIPS.cc/2021/Conference — NeurIPS 2021 Spotlight_

### Official Review · Reviewer_ZDHq · 2021-07-09

**Rating:** 7
**Confidence:** 4

**Summary:**

This paper studies regret minimization in stochastic shortest paths. It presents an algorithm with regret that is: minimax optimal, parameter-free and horizon-free (in various cases). Here, horizon-free refers to logarithmic dependence in the expected time of the optimal policy (that replaces the horizon in finite-horizon problems).

**Limitations And Societal Impact:**

Yes.

**Main Review:**

For:

The results of this paper are very impressive.
It gives the first parameter-free algorithm that is able to achieve minimax optimal regret, compared to Rosenberg et al. (2020) which is parameter-free but is not minimax optimal (extra $\sqrt{B^* S} $)  and compared to Cohen et al. (2021) that is minimax optimal but needs to know $T^*$ and $B^*$.
Moreover, the algorithm enjoys graceful dependence on $T^*$ (i.e., logarithmic in some cases) compared to polynomial dependence in previous work.

Against:

My main concern with this paper is that it is not written in a way that lets readers understand the theoretical analysis of the algorithm.
I think that this paper does a great job presenting the problem, related work, algorithm (including algorithmic novelties) and main results. However, the analysis and the proof techniques are almost non-existent in the main paper and do not focus on the new technical contributions of the paper. Therefore, I think that the chance that a reader could learn these contributions and that they will affect future work is very limited. Moreover, it was very hard for me to follow the analysis in the appendix without the intuition that proof sketches are supposed to give (this is the reason why I cannot say with absolute certainty whether the proofs are correct). In my opinion this paper could have much more impact on the theoretical community (this is clearly a paper for this community),  if the authors clearly conveyed the main ideas behind the proofs and the intuition behind the analysis.

Specifically, I found the proof sketch of theorem 3 impossible to understand. It lists some facts (that  I could not combine into a proof) without giving a high level idea about what goes on in this proof. This especially goes for the techniques of Zhang et al. (2020a) - the authors do not explain what are those techniques, how they are used and what changes are needed so they work in SSP (and not finite-horizon).

To me it looks like some of the analysis sketch should have focused on properties (1) and (2) of the VISCO procedure. As the authors state, VISCO is the main contribution of this paper (and definitely the most interesting one) and these two properties are the essential components that make the analysis work. However, there is no sketch of this analysis and it is very hard to understand how these properties are proved and how they affect the analysis.

Instead of the analysis, I think the authors put too much focus on the exact form of the regret bound in various cases, which I did not find useful. This goes especially for the "General Costs and Order-Accurate Estimate of $T^*$ Available" which diminishes from the importance of the algorithm's regret bound in my opinion, as it (kind of) eliminates the need to be parameter-free since there is an estimate of $T^*$, although the algorithm does not require this knowledge really. I think that the authors should put less emphasis on those cases, state the main theorem in corollary 6 and say how it is stronger if costs are positive or an estimate of $T^*$ is available.

A smaller concern is the computational complexity of the algorithm. While it is polynomial and this is clearly not the focus of the paper, I noticed that $\epsilon_{VI} $ grows linearly with the number of timesteps (right?) and that the VISCO operator is $(1 - \nu^2) $-contracting for $\nu$ which is the minimal transition probability to the goal of the skewed transition function. Doesn't this mean that each computation of the optimistic policy takes time polynomial in $K$ (since $\nu $ might be $1/K$)? This is quite problematic (and also appears in some optimistic algorithms for linear finite-horizon MDPs).
I wonder if the algorithm can be made more efficient at the expense of worse regret, by skewing the transitions by $1/\sqrt{n}$ or maybe something else. Do you think this is possible? If so, looks like a good discussion to add.
In general, I think that a discussion about the computational complexity is important so readers can understand that maybe improvements can be made there. This is kind of hidden right now and appears deep in the appendix.

Post rebuttal: I thank the authors for these clarifications, and hope that they will be incorporated into the camera-ready version. I am keeping my original positive score.

**Time Spent Reviewing:**

4

---

> ### Author Response · Authors · 2021-08-10
> **Response to Reviewer ZDHq**
>
> Thank you for the positive comments and for the relevant suggestions to refactor and improve the presentation of the paper.
>
> **Proof details:**
> As suggested, we will push some content of Section 4.1 to the appendix so as to incorporate the following in the main paper:
>
> - Proof details on the two VISGO properties:
>
> We will explain that the key idea here is to study an SSP-specific auxiliary function $f$ (which represents the application of the optimal Bellman operator with iterate-dependent bonuses). We prove that $f$ is not only monotonous (similar to the one of [Zhang et al., 2020a] in finite-horizon), but also contractive. This allows us to prove optimism and convergence (properties 1 and 2), respectively.
>
> - A more insightful proof sketch of Thm. 3:
>
> We will go through the proof of Thm. 3 (App. D), focusing on the meaning of each lemma. Lem. 13 presents a set of concentration bounds on learned statistics ($V$-values, costs and transitions). The slightly skewed transition $\widetilde{P}$ guarantees that all policies calculated from it are proper (Lem. 15) and ensures finite convergence of VISGO (Lem. 16 (.3), Lem. 19), while introducing only very small error of $P$-estimation (Lem. 14). Optimism (Lem. 17) is closely related to the first two properties in Lem. 16.
>
> Next, we decompose the regret into three parts: $X_1$ (the error on learned $V^m$-values), $X_2$ (Bellman error) and $X_3$ (cost estimation error), and among them the major part is $X_2$.  Later, $X_1$ and $X_2$ introduce intermediate quantities $X_4$ (variance of learned $V^m$) and $X_5$ (variance of the difference $V^{\star} - V^m$), which are bounded using the recursion technique generalized from [Zhang et al., 2020a]. At a high-level, the key idea is to calculate errors of different orders, $F(1), F(2), \ldots, F(d), \ldots$ (refer to Lem. 24), and recursively bound $F(i)$'s variance by a sub-linear function of $F(i + 1)$.
>
> Throughout the proof we bound quantities by solving inequalities that contain the unknown quantities on both sides, such as $X_3 < \widetilde{O}(\sqrt{X_3 + C_{K}})$ or $X_2 < \widetilde{O}(\sqrt{X_2 + C_{K}})$, where $C_K$ denotes the cumulative cost over the $K$ episodes.
> Indeed, the analysis at each time step $t$ brings out the instantaneous cost $c_t$ and it is important to combine them so that we can make $C_K$ appear explicitly. We can then circumvent the dependence on $C_K$ (and replace it by a dependence on $K$) by using the definition of regret in SSP and its relation to $C_K$ (see App. D.5.4).
>
>
>
> **Computational complexity:** By the stopping condition $\vert \vert V^{(i+1)} - V^{(i)} \vert \vert_{\infty} \leq \epsilon_{\textrm{\tiny{VI}}}$ and the $\rho$-contraction of the operator (Lem. 19), any VISGO procedure is guaranteed to stop at an iteration $i \leq \log(\max(B_{\star},1)/ \epsilon_{\textrm{\tiny{VI}}}) / (1-\rho)$. The $\epsilon_{\textrm{\tiny{VI}}}$ precision is set to $2^{-j}/SA$ at the $j$-th VISGO planning procedure, hence its impact on the computational complexity is only logarithmic in $T$ since $\log(1/\epsilon_{\textrm{\tiny{VI}}}) = O(j) = O(SA \log(T))$ (the last equation is by the ''doubling'' definition of the trigger set line 4, Alg. 1). The complexity of a VISGO procedure also depends on the contraction factor $\rho$ (which we have improved from $1-\nu^2$ to $1-\nu$ since the submission deadline), which implies $O(K)$ complexity and thus $\widetilde{O}(K)$ total complexity. It is an interesting direction to improve it, which we believe is tied to a more general open question in SSP.
>
> Indeed, all existing parameter-free algorithms for online SSP have similar complexity. This is because the alternative quantity to our $\nu$ that can appear in computational complexity analysis is $c_{\min}^{-1}$: however, when the cost perturbation trick is applied, $c_{\min}^{-1}$ is set to be of order $K$, hence this leads to a computational complexity that is at least $\Omega(K)$. On the other hand, if $T_{\star}$ prior knowledge is available, $T_{\star}$ can be used to tackle successive finite-horizon problems as in [Cohen et al., 2021] which allows to achieve $O(\log(K))$ complexity. This highlights a trade-off in the current online SSP literature between parameter knowledge and computational complexity: it is an interesting open question whether it is possible to have $\log(K)$ complexity while staying parameter-free. We will incorporate this discussion on computational complexity in the main paper.

---

### Official Review · Reviewer_qhwx · 2021-07-16

**Rating:** 7
**Confidence:** 4

**Summary:**

This paper develop an algorithm that achieves minimax optimal regret for SSP with stochastic costs. Notably, the algorithm is completely parameter-free, and also horizon-free (logarithmic dependency on $T_{\star}$ even in the lower order term).

**Ethical Concerns:**

This work is mainly theoretically and has no ethical concerns.

**Limitations And Societal Impact:**

This work is mainly theoretically and has no foreseeable societal impact.

**Main Review:**

This paper resolves an important problems in SSP with stochastic costs: removing the extra $\sqrt{S}$ factor in the leading term. What's even more exciting is that they push $1/c_{\min}$ dependency into log terms. This is a very strong result, as $1/c_{\min}$ can be arbitrarily large in SSP and can only be handled by perturbation, in which case most of algorithms end up with sub-optimal regret bound. Thanks to the strong result, the upper bound $B$ can be as large as $O(\sqrt{K})$, making it easy to achieve parameter-free.
The paper is also well written and easy to follow. I appreciate author's effort on carefully discussing each scenario and the consequent regret bound.

The only short shortcoming is that the horizon-free analysis seems to be directly adopted from Zhang et al. [2020a], thus has limited novelty. However, the result of this paper is quite exciting and complete, and I suggest accepting this paper.

Questions:
1. Is the perturbation on transition really necessary? It seems to be an artifact of analysis.
2. Can the algorithm be extended to linear function approximation?

**Time Spent Reviewing:**

3

---

> ### Author Response · Authors · 2021-08-10
> **Response to Reviewer qhwx**
>
> Thank you for the valuable comments and for appreciating our work. Please find our response to the questions below.
>
> - “Is the perturbation on transition really necessary?”
>
> The perturbation on transition is required to guarantee that the SSP problem on which we plan is well-posed (i.e., admits a solution), in other words that the planning procedure (in our case, bonus-based value iteration for SSP) does not diverge. Indeed, in SSP it is mandatory to have the assumption of at least one proper policy to guarantee that value iteration converges [Bertsekas, 1995; Bonet, 2007]. In the online SSP setting, performing planning requires finding an SSP model for which there exists at least one proper policy. In fact, the empirical model may not verify this condition, hence the slight perturbation. Interestingly, the latter ensures the stronger property that all policies are proper, while only contributing to a lower-order term in the regret. Finally, we note that the perturbation would not be required under different schemes to compute policies (that do not rely on value iteration for SSP), e.g., via a finite-horizon reduction [Cohen et al., 2021] or via one-step planning à la Q-learning.
>
> - “Can the algorithm be extended to linear function approximation?”
>
> Extending SSP to linear function approximation (LFA) is a challenging open question. The recent work of [Vial et al., 2021] proposes a first approach under the strong assumption that all policies are proper in the true MDP. As they explain, the main challenge of SSP with LFA is the computational aspect, i.e., how to compute SSP policies in an efficient way. In this vein, we note that our skewing procedure (which adds a non-zero probability to reach the goal from every $(s,a)$) is intuitively reducing the problem to a discounted MDP problem (see Remark 1), but it is also still an open problem to devise computationally efficient algorithms for LFA in discounted MDPs.
>
> [Vial et al., 2021]: Regret Bounds for Stochastic Shortest Path Problems with Linear Function Approximation. Daniel Vial, Advait Parulekar, Sanjay Shakkottai, R. Srikant. arXiv preprint arXiv:2105.01593, 2021.

---

### Official Review · Reviewer_rqFE · 2021-07-16

**Rating:** 8
**Confidence:** 4

**Summary:**

This work studies the problem of learning stochastic shortest path (SSP) with unknown transition. The authors propose a model-based parameter-free algorithm EB-SSP which achieves the minimax regret rate O(B sqrt(SAK)), where B is the expected cumulative cost of the optimal policy, without prior knowledge of B or T (the expected time-to-goal of the optimal policy). The authors further consider several cases where the regret only has a logarithmic dependence on T.

**Limitations And Societal Impact:**

This work is pure theoretical and does not have any potential negative societal impact.

**Main Review:**

1. Contribution

To the best of my knowledge, EB-SSP is the first parameter-free algorithm that achieves O(B sqrt(SAK)) regret. This algorithm is designed upon recent advances in episodic MDPs (e.g. Azar et al., 2017; Jin et al., 2018; Zanette and Brunskill, 2019) and specifically the Bernstein-type exploration bonus. Importantly, the author adopts the idea of the MVP algorithm of Zhang et al. (2020a) to preserve the monotonicity property of Bellman update, which helps achieve tighter regret bounds. Besides, by slightly biasing the empirical transitions, the algorithm ensures that all induced polices are proper with contributing to a lower order extra term. Such technique is widely used in many tasks.

On the other hand, the discussion of various cases is meaningful and inspiring.

2. Weaknesses

Despite the algorithm efficiency, I haven't found any specific weak points of this paper.

**Time Spent Reviewing:**

48

---

> ### Author Response · Authors · 2021-08-10
> **Response to Reviewer rqFE**
>
> Thank you for the positive review! We welcome all further comments.

---

### Official Review · Reviewer_P4gA · 2021-07-19

**Rating:** 8
**Confidence:** 3

**Summary:**

This paper investigates the problem of regret minimization of the stochastic shortest path, which is actively studied in the recent two years. The main contribution is a novel algorithm that enjoys nearly minimax optimal regret, matching the existing lower bound up to some log factors. Furthermore, the algorithm, as well as its variant, enjoy other desired properties, including parameter-free (does not require the knowledge of $B_\star$ or $T_\star$) and horizon-free (does not depend on $T_\star$)

**Main Review:**

post rebuttal: I am fine with the authors' rebuttal and keep my score as it is.

===================

This paper investigates an important problem in modern RL theory -- regret minimization for the stochastic shortest path. The problem itself is actively studied in the recent two years. The recent work [Rosenberg et al., 2020] gives a nearly optimal regret bound for this problem, yet the algorithm requires an unknown parameter $B_\star$ and also suffers from a $\sqrt{S}$ suboptimality gap. The paper resolves the two issues by proposing some interesting ideas. From my point of view, the paper is of high quality and the overall presentation is good to follow. I would be happy to recommend the acceptance. Several comments are listed below for the authors' further clarification.

(1) The idea of skewed transition probability is very interesting. As stated in the paper (Line 208), such a skewing guarantees that all policies are proper in the policy class. It would be appreciated if the authors can provide more specific explanations for this argument. In addition, I found the trick related to the recent work on adversarial SSP [Chen et al. 2021], in which the authors use a skewed occupancy measure space. It would be interesting to have some discussions.

(2) The parameter-free part is another important contribution of this paper. However, I have to say that the presentation in the main paper is too thin to understand... (due to page constraints). I remember that there will be an extra page in the camera-ready stage, so the authors are encouraged to add more details, especially about the two-phase adaptation, once got accepted. Furthermore, I am curious that whether a two-layer method could be useful for eliminating the unknown $B_\star$? The paper [Chen et al. 2021] uses such a mechanism (with multi-scale experts) to remove the prior knowledge of $T_\star$. I would be happy to see some discussions on this point.

(3) The "horizon-free" claim, in my opinion, is not quite strong. In particular, as stated in the table, when achieving minimax optimal and parameter-free, there is actually a factual dependence on $T_\star$. I would suggest less emphasis in this regard.

**Time Spent Reviewing:**

15 hours

---

> ### Author Response · Authors · 2021-08-10
> **Response to Reviewer P4gA**
>
> Thank you for the valuable comments and for appreciating our work. Please find our response to the comments below.
>
> **Skewing argument:** The skewed transition probabilities ensure that there exists a non-zero probability of reaching the goal state from every state-action pair. This implies that the expected goal-reaching time of any policy is finite and thus that all policies are proper in this skewed model. While only contributing to a lower order term in the regret, our skewing ensures that the SSP problem (on $\widetilde{P}$) admits a solution (this may not be the case without it, since the empirical SSP problem on $\widehat{P}$ may not admit any proper policy and thus value iteration may diverge).
>
> In adversarial SSP [Chen et al., 2021], the skewing on the occupancy measures is not performed for well-posedness/computational reasons, but rather to improve the regret guarantee by reducing the variance of the learner. Interestingly, their skewed occupancy measure can be viewed as adding positive bias to the costs, thus increasing the “total cost” to reach the goal, while our positive bias on the goal-reaching probability has the opposite effect of decreasing the “total cost” to reach the goal.
>
> **Parameter-free scheme:** We need to ensure that the proxy $B$ will not remain below $B^{\star}$ for too long, since in this case, the regret may keep growing linearly. The main challenge is then to find an appropriate scheme to increase the value of $B$ over time. First, we increase $B$ as a function of $k$ whenever episode $k$ ends: this ensures that $B \geq B^{\star}$ for large enough episodes. Yet this is not enough: indeed notice that when $B < B^{\star}$, the agent may never reach the goal, so we cannot exclusively rely on the end of episode as a trigger for increasing $B$. We thus also increase $B$ whenever the cumulative cost exceeds a carefully defined threshold. Since the regret is upper bounded by the cumulative cost, this second condition prevents the learner from accumulating too large regret when $B < B^{\star}$. (We also need to introduce a third condition to ensure the computational efficiency, since VISGO may diverge when $B < B^{\star}$.) At a high-level, the analysis of the scheme proceeds as follows: bound the regret by the cumulative cost when $B < B^{\star}$ (first regime), and by the regret bound of Thm. 3 when $B \geq B^{\star}$ (second regime). Note that this two-regime decomposition is only implicit (i.e., at the level of analysis), since the agent is unable to know in which regime it is (since $B^{\star}$ is unknown). We will use the extra page in case of acceptance to provide more details on the scheme.
>
> The paper of [Chen et al., 2021] uses a multi-scale expert mechanism to circumvent the knowledge of $T_{\star}$ for adversarial SSP with known transitions. However, as explained in their follow-up paper [Chen and Luo, 2021] for unknown transitions, “Chen et al. (2020) were able to resolve this for expected regret bounds, but extending their techniques to high-probability bounds is related to deriving a high-probability bound for the so-called multi-scale expert problem, which is also still open.”
>
> **Horizon-free property:** The dependence on $T_{\star}$ or $1/c_{\min}$ (which may be arbitrarily large quantities) was polynomial in the additive terms in previous state-of-the-art regret bounds. EB-SSP makes a significant step in improving it, making it at worse $T_{\star}/\textrm{poly}(K)$ in the general case, and in particular only logarithmic in some cases (e.g., positive costs, or loose estimate of $T_{\star}$ available), which mirrors (and generalizes) the horizon-free properties recently uncovered in finite-horizon MDPs. It is an exciting open question whether fully parameter-free and horizon-free regret is simultaneously possible in SSP. Our intuition is that this would require an altogether different road map for the analysis that does not rely on the cost perturbation trick.

---

### Decision · Program_Chairs · 2021-09-27

**Decision:**

Accept (Spotlight)

**Comment:**

This paper makes solid contribution to pushing the frontier of the stochastic shortest path problem.
All reviewers support accept strongly.